# TTS-Hub: Leveraging Modular LoRAs and Arithmetic Composition for Controllable Text-to-Speech

## Abstract

Controllable text-to-speech (TTS) aims to generate speech from text while allowing control over prosodic and speaker-related attributes such as pitch, age, and accent. Existing controllable TTS methods primarily rely on natural language prompts to guide the synthesis process or utilize reference audio cloning to achieve control. However, prompt-based approaches often struggle with the cross-modal semantic gap between textual descriptions and intended speech attributes, leading to imprecise and coarse-grained control. Conversely, cloning methods depend heavily on reference audio samples and struggle to generalize beyond the characteristics seen in those samples, resulting in limited flexibility. To overcome these challenges, this paper proposes TTS-Hub, a novel controllable TTS framework that employs modular Low-Rank Adaptation (LoRA) components and their arithmetic-based composition to achieve fine-grained and flexible controllable TTS. Specifically, we construct a comprehensive Data Hub, which covers 6 high-level attribute categories and 32 fine-grained speech attributes. Leveraging this attribute-specific data, we fine-tune two mainstream TTS frameworks to obtain a corresponding LoRA Hub, where each modular LoRA is specialized for a specific speech attribute. At inference time, TTS-Hub selects the required LoRA modules and combines them through simple arithmetic composition to produce a fused LoRA that simultaneously encodes multiple attribute representations, enabling flexible and extensible multi-attribute control without retraining the backbone. Extensive experiments show that individual LoRAs provide precise single-attribute control, while arithmetic composition yields flexible and interpretable multi-attribute speech and consistently outperforms prompt-based baselines. Code and data are available in supplementary materials.

## 1 Introduction

Text-to-speech (TTS) aims to convert textual input into natural-sounding speech and has rapidly advanced from neural network-based architectures (e.g., Tacotron 2 (Shen et al., 2018), FastSpeech (Ren et al., 2019), NeuralSpeech (Tan et al., 2024), and CosyVoice (Du et al., 2024a)) to large language model (LLM)-based synthesis systems such as GPT-4o-mini-TTS (OpenAI, 2025), Gemini-2.5-TTS (Comanici et al., 2025), and Qwen-TTS (Qwen, 2025). To meet the growing demand for personalized synthesis, controllable TTS systems (Ji et al., 2025; Wang et al., 2025; Huang et al., 2024c;b) have become increasingly essential. These systems enable users to precisely specify speech attributes, including accent, age, and emotion, thus facilitating diverse applications such as customized audiobooks, personalized voice assistants, and virtual game character creation.

Current Controllable TTS approaches are dominated by two paradigms, each with significant limitations. Natural-language prompting (Guo et al., 2023; Yang et al., 2024; Ji et al., 2024a) guides speech synthesis through intuitive instructions (e.g., "read this like a sad teenager with a Scottish accent"), but the inherent cross-modal semantic gap requires the model to infer distinctive acoustic features from vague textual descriptions. These features are often only weakly expressed or partially realized in the generated speech, resulting in coarse and imprecise control over individual attributes (Xie et al., 2024). Reference-audio cloning (Hao et al., 2025; Ju et al., 2024; Jiang et al., 2024) alleviates this ambiguity by conditioning on a provided reference clip. However, it requires

a precisely matched reference for every desired attribute combination, rendering it impractical for diverse or composite requests such as "elderly, cheerful, Mandarin accent" (Barakat et al., 2024). Consequently, there is an urgent need for a fine-grained and flexible controllable TTS solution.

A promising solution emerges from Low-Rank Adaptation (LoRA) (Hu et al., 2022), a parameter-efficient fine-tuning technique that keeps the pretrained backbone frozen while inserting compact low-rank adapter matrices that are learned to adapt to specific tasks and capabilities. The modular architecture of LoRA has sparked further research on the synergistic composition of multiple trained LoRAs, showing that composing several pretrained LoRAs can improve cross-task generalization (Huang et al., 2024a; Wu et al., 2024). This composable modularity motivates us to investagate LoRAs and their composition for controllable TTS.

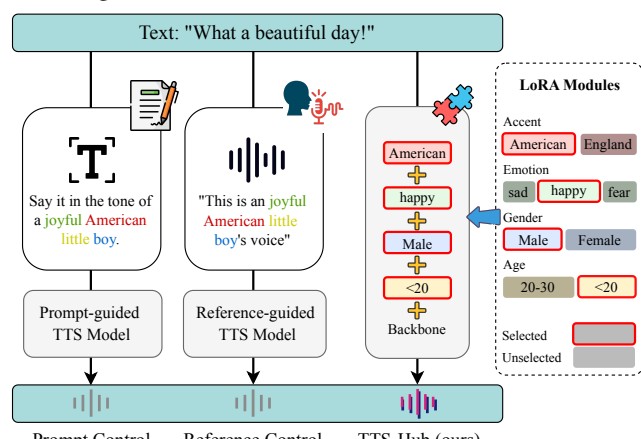

Figure 1: Overview of two mainstream Controllable TTS Paradigms and Our Proposed TTS-Hub Framework. Existing methods rely on natural-language prompts (coarse and imprecise due to cross-modal semantic gaps) or reference audio cloning (inflexible and hard to generalize). In contrast, TTS-Hub achieves fine-grained, flexible, and extensible control via composing attribute-specific LoRA modules.

To this end, we propose TTS-Hub, which treats each LoRA as an attribute specialist and achieves fine-grained and flexible control by selecting and composing the relevant LoRAs at inference. Specifically, we first collect four publicly available TTS datasets and reconstruct the samples into a comprehensive data hub, covering 6 high-level attribute groups (accent, age, emotion, gender, energy, pitch) and 32 fine-grained speech attributes. Leveraging these attribute-specific subsets, we build a LoRA hub by fine-tuning one adapter in a specific attribute. During inference, TTS-Hub selects the adapters that match a user's request, combines them into a single composite LoRA when multiple attributes are needed, and inserts this adapter into the frozen backbone to generate speech. Extensive experiments demonstrate that: (i) individual adapters can precisely control their target attributes; and (ii) arithmetic combination allows intuitive mixing, attenuation, or amplification of attributes, providing a flexible and interpretable approach to multi-attribute control. In addition, TTS-Hub is inherently extensible: new attribute adapters can be added to the hub, thereby continuously expanding the system's expressive capabilities. To the best of our knowledge, we are the first to explore LoRAs and their composition for controllable TTS.

Our main contributions are as follows.

- We present TTS-Hub, the first framework that formulates controllable TTS as arithmetic-based composition of LoRA adapters, enabling fine-grained and flexible control.
- We construct and release a comprehensive and easily-extensible hub of attribute-specialist adapters for two mainstream backbones, enabling reproducible research and practical deployment.
- Extensive experiments demonstrate that individual adapters deliver precise single-attribute control, whereas arithmetic composition yields interpretable multi-attribute synthesis.

## 2 RELATED WORKS

**Controllable TTS.** Recent advances in controllable text-to-speech (TTS) systems, such as VALL-E R (Han et al., 2024), CosyVoice (Du et al., 2024b), and GLM-4-Voice (Zeng et al., 2024), have made significant strides in generating high-fidelity speech with some degree of style control, driven by large-scale models and extensive training data. Systems like PromptTTS (Guo et al., 2023) use natural-language prompts for acoustic attribute control (e.g., pitch, timbre), while (Liu et al., 2024) supports accent control and (Bott et al., 2024) emotion control. However, challenges remain in achieving fine-grained controllability. Current models struggle to synthesize specific voices, such

as a 60-year-old Irish speaker, or combine nuanced attributes flexibly. Moreover, reference-audio cloning in systems like (Ruggiero et al., 2021) captures speaker traits but is impractical for complex attribute combinations, particularly in real-time scenarios. Consequently, most systems are limited to replicating example styles or offering only a narrow set of attributes.

**LoRA-based Fine-tuning for TTS.** Low-Rank Adaptation (LoRA) (Hu et al., 2022) provides a parameter-efficient method for fine-tuning large pretrained models by learning low-rank weight updates while freezing the backbone parameters. This approach has been widely used in speech synthesis for efficient feature learning. For example, (Qi et al., 2024) modeled emotional features with LoRA modules, (Bondaruk & Kubiak, 2025) enhanced multilingual expressiveness, and (Lou et al., 2024) improved speaker style modeling. These studies demonstrate LoRA's effectiveness in capturing expressive speech features. However, its compositional potential remains underexplored, especially in combining LoRA modules to capture diverse, fine-grained speech attributes. To address this, we introduce TTS-Hub, a modular framework that trains LoRA modules for individual speech attributes and combines them at inference time, enabling flexible, interpretable user-controlled speech generation.

## 3 PROPOSED METHOD

In this section we present TTS-Hub, a novel framework for speech synthesis that enables fine-grained and flexible control through the composition of modular LoRA adapters. Our framework consists of three stages: (A) **Hub Construction** builds a data hub with labeled attributes and trains attribute-specific LoRA modules; (B) **LoRA Matching** selects and composes modules based on customized user intent; (C) **Inference with LoRA Adapters** injects the composite adapter into a frozen backbone to synthesize speech that aligns with the target features.

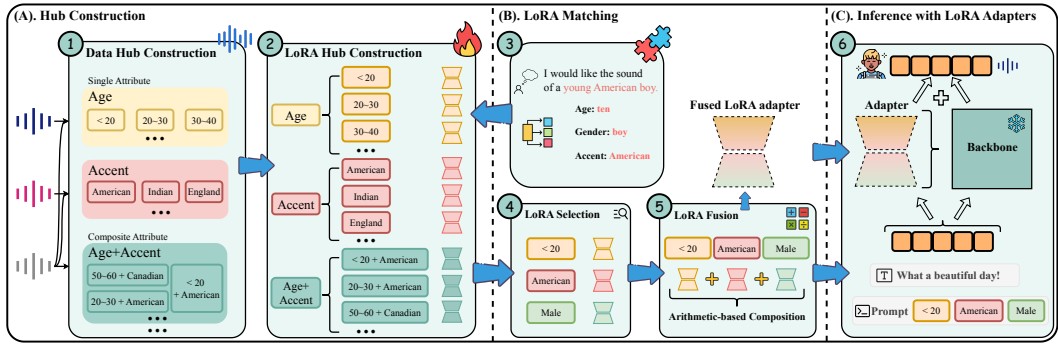

Figure 2: Overview of Our TTS-Hub.

### 3.1 HUB CONSTRUCTION

We first build the Data Hub by collecting and reorganizing samples from multiple public TTS datasets to ensure diverse speech attribute coverage. Then, we use attribute-specific subsets to train modular LoRA adapters, forming the LoRA Hub.

**Data Hub.** We collect multiple publicly available TTS datasets and organize a unified attribute taxonomy $A = A_a \cup A_c$, where:

- Atomic Attributes $A_a$. Each atomic attribute $a_j \in A_a, j = 1, \ldots, J$ represents an indivisible speaker characteristic, such as *"British accent"* or *"over 60 years old"*. $J$ denotes the total number of atomic attributes. These attributes are grouped into attribute categories like *Accent*, *Age*, etc., enabling control over specific aspects of speech.

- Composite Attributes $A_c$. Each composite attribute (denoted as $\hat{a}_k \in A_c, k = 1, \ldots, K$) is a combination of two or more atomic attributes drawn from different categories. Formally, we write $\hat{a}_k = a_{k_1} + a_{k_2} \cdots + a_{k_n}$, where $a_{k_i} \in A_a$, $n \geq 2$ is the number of atomic attributes involved in $\hat{a}_k$, and $K$ is the number of composite attributes. For example, the

composite attribute *"age: 20+accent: American"* merges the atomic attributes *"20 years old"* and *"American accent"*, forming a richer and more specific speaker profile.

For each atomic and composite attribute $a \in A$, we collect an attribute-specific subset $\mathcal{D}_a = \{x_i, y_i, a\}_{i=1}^{n_a}$ from the overall corpus, where $x_i$ is the input text, $y_i$ is the corresponding speech waveform, and $a$ is the associated attribute annotation. The number of samples in $\mathcal{D}_a$ is denoted by $n_a$. The full Data Hub is therefore represented as:

$$\mathcal{D} = \{\mathcal{D}_a \mid a \in A\} \tag{1}$$

**LoRA Hub**. Given the data hub $\mathcal{D}$, we train a set of attribute-specialized LoRA adapters, collectively forming the LoRA Hub $\mathcal{H}$. Each adapter in $\mathcal{H}$ is trained to capture the effect of a specific attribute $a$, while keeping the backbone model $\mathcal{M}$ frozen. Let $W \in \mathbb{R}^{d_{\text{out}} \times d_{\text{in}}}$ denote a trainable weight matrix in $\mathcal{M}$ (e.g., a linear projection in the self-attention block), LoRA introduces a learnable low-rank update $\Delta W_a$, applied additively to the frozen weight:

$$W_{\text{adapted}} = W + \Delta W_a, \quad \Delta W_a = \mathbf{B}_a \mathbf{A}_a \tag{2}$$

where $\mathbf{A}_a \in \mathbb{R}^{r \times d_{\text{in}}}$, $\mathbf{B}_a \in \mathbb{R}^{d_{\text{out}} \times r}$, and $r \ll \min(d_{\text{in}}, d_{\text{out}})$ is the bottleneck rank that controls the parameter efficiency. Only $\mathbf{A}$ and $\mathbf{B}$ are trainable, while the original backbone parameters $W$ remain frozen. Since LoRA may be applied to multiple layers within the backbone model, we represent the complete adapter for attribute $a$ as $\mathbf{\Delta}_a = \left\{ \Delta W_a^{(l)} = \mathbf{B}_a^{(l)} \mathbf{A}_a^{(l)} \right\}_{l \in \mathcal{M}}$. The optimization objective is defined as:

$$\min_{\mathbf{\Delta}_a} \sum_{(x_i, y_i) \in \mathcal{D}_a} \mathcal{L}\left(\mathcal{M}_{\mathbf{\Delta}_a}(x_i), \ y_i\right) \tag{3}$$

where $\mathcal{L}$ is the training loss adopted by the backbone model $\mathcal{M}$. Each attribute $a \in A$ is associated with a subset $\mathcal{D}_a \subset \mathcal{D}$, which is used to train its corresponding adapter independently. Finally, we obtain the LoRA hub:

$$\mathcal{H} = \{\mathbf{\Delta}_a \mid a \in A\} \tag{4}$$

### 3.2 LoRA Matching

Once the LoRA Hub $\mathcal{H}$ is constructed, it enables flexible control over speech synthesis through modular selection and composition.

1) Single-attribute control. A specific attribute $a \in A$ can be controlled by directly injecting its corresponding LoRA adapter $\mathbf{\Delta}_a$ into the backbone model.

2) Multi-attribute control. Beyond individual attributes, more complex attribute behaviors can be expressed by composing multiple LoRA adapters. For instance, additive composition $\mathbf{\Delta}_{a_1} + \mathbf{\Delta}_{a_2} + \cdots + \mathbf{\Delta}_{a_n}$ allows simultaneous control over multiple attributes. Alternatively, subtractive operations can be used to remove a specific attribute from a composite adapter: $\mathbf{\Delta}_{a_{k_1} + a_{k_2} \cdots + a_{k_n}} - \mathbf{\Delta}_{a_{k_j}}$, where $\mathbf{\Delta}_{a_{k_1} + a_{k_2} \cdots + a_{k_n}}$ is a LoRA adapter for composite attribute $a_{k_1} + a_{k_2} \cdots + a_{k_n}$, and $a_{k_j}$ is the attribute we would like to remove from the speech.

3) Attribute scaling. Furthermore, the strength of a single attribute can be adjusted by scaling the adapter with a weight $w$, as in $w \cdot \mathbf{\Delta}_a$, allowing gradual amplification or attenuation of the target attribute.

### 3.3 Inference with LoRA Adapters

The selected or composite LoRA adapter $\mathbf{\Delta}$ is integrated into the frozen backbone model $\mathcal{M}$. Given an input text $x$, the final output is synthesized as:

$$y' = \mathcal{M}_{\mathbf{\Delta}}(x) \tag{5}$$

In this way, our method eliminates the need for retraining the model for each attribute combination and enables flexible and interpretable control over speech attributes via arithmetic composition.

### 3.4 FRAMEWORK ANALYSIS

***Take-away*1. The framework's modular design provides unparalleled flexibility and scalability.**
The use of modular LoRA adapters enables flexible composition of various attributes, allowing users
to create novel voice styles without retraining. Additionally, new attributes can be easily integrated
into the system by fine-tuning and adding new LoRA modules, ensuring seamless expansion without
altering the backbone model.

***Take-away*2. The framework combines transparency with user control for more informed customization.** Unlike conventional black-box approaches, our framework offers interpretable control
by explicitly composing LoRA modules, each dedicated to a specific attribute. This modular design
enhances transparency, enabling users to easily trace and influence each attribute's contribution to
the synthesized speech, while providing fine-grained customization options.

## 4 EXPERIMENTAL SETTINGS

### 4.1 TTS-HUB CONSTRUCTION

**Data Hub Assembly.** To support fine-grained controllable text-to-speech (TTS) using attribute-specific LoRA module composition, we reconstruct a large-scale speech dataset through a label-sharing strategy, integrating data from four publicly available corpora: Common Voice (Ardila et al., 2020), L2-ARCTIC (Zhao et al., 2018), Indic TTS (Baby et al., 2016), and TextrolSpeech (Ji et al., 2024b). Each utterance in the dataset is annotated with one or more fine-grained attribute labels. Specifically, the dataset covers six high-level at-

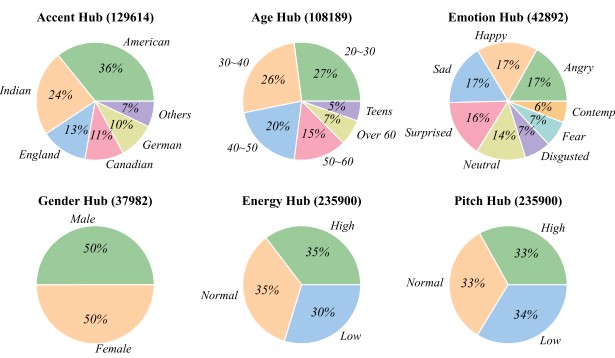

Figure 3: Attribute distributions in the Data Hub.

tribute categories: Accent, Age, Emotion, Gender, Energy, and Pitch, comprising a total of 32
fine-grained subclasses. These include 10 accent types (e.g., American, Indian, British, Scottish), 6
age groups (ranging from under 20 years old to over 60 years old), 8 emotional states (e.g., Angry,
Happy, Neutral), 2 gender classes (Male and Female), and 3 levels each for energy and pitch (High,
Normal, and Low). In addition, based on label co-occurrence, we construct 9 sets of composite
attribute data (e.g., American + over 60 years old) to support the subtractive validation experiments.
We split the entire dataset into training, validation, and test sets using an 8:1:1 ratio. This carefully
curated and diverse dataset provides a robust foundation for evaluating the fine-grained attribute
controllability of TTS systems. The detailed distribution of the dataset is shown in Figure 3.

**LoRA Hub Training.** We adopt two backbone models: Parler-TTS (Lacombe et al., 2024) and
VoiceLDM (Lee et al., 2024). We report results on Parler-TTS in the main paper, and leave the
results on VoiceLDM to the supplementary materials due to space constraints. The former is based
on a Transformer architecture, while the latter employs a diffusion-based architecture. Both frameworks support controllable speech synthesis purely through textual prompts. All experiments are
conducted on a Linux system equipped with eight NVIDIA A40 GPUs, each featuring 48 GB memory. During training, we set the batch size to 8 and train for 10 epochs to ensure stable model
convergence. Details of additional hyperparameter configurations are provided in the supplementary materials. We perform LoRA fine-tuning using all 32 single-attribute datasets. For the two key
LoRA hyperparameters, namely LoRA Rank ($r$) and LoRA Alpha ($\alpha$), we follow the configuration
described in (Hu et al., 2022), and further validate the effectiveness of our chosen hyperparameters
through additional experiments presented in the supplementary materials. Therefore, in all subsequent experiments, we consistently set $r$ to 16 and $\alpha$ to 32 throughout the entire model training
process. The process leading to this choice is elaborated in Appendix 6.

## 4.2 EVALUATION METRICS

In order to comprehensively evaluate the performance of the synthesized speech, we first employ a human evaluation metric the Attribute Mean Opinion Score (**AMOS**), which assesses the perceptual expressiveness of LoRA modules on specific attributes from a human auditory perception perspective. The AMOS evaluation is conducted by a panel of 30 expert raters, all with research backgrounds in speech synthesis. Each participant is instructed to rate the quality of the synthesized audio on a scale from 1 to 5. Raters are compensated based on the number of evaluations they complete.

Moreover, we introduce two large language model-based metrics: **Win-Attr**, which quantifies the win rate in terms of attribute expressiveness, and **Win-Qual**, which assesses the win rate for overall speech quality. These metrics are derived through pairwise comparisons conducted with GPT-4o-Audio-Preview, which accepts audio inputs and returns final judgments in textual form. We employ two carefully designed prompts tailored to guide the model in comparing LoRA-based synthesis with natural language prompting, determining which method produces speech with superior quality or more salient attribute expression. In cases where the model finds the two audio samples comparable and unable to decisively differentiate them, it is allowed to declare a tie. The prompt details used in these evaluations are provided in Appendix 6.

In addition, three automatic objective metrics are adopted: Mel-Cepstral Distortion (**MCD**, (Kubichek, 1993)), F0 Root Mean Square Error (**F0-RMSE**, (Toda et al., 2007)), and Structural Similarity Index (**SSIM**, (Wang et al., 2004)), which measure the similarity between synthesized speech and ground truth.

## 4.3 EVALUATION RATIONALE

**Data Selection and Evaluation Based on Attribute Co-occurrence** Given the scale of the training data, we operate under the hypothesis that data samples sharing a specific attribute will cluster together in the feature space, thereby sufficiently capturing the characteristic patterns of that attribute. To validate the model's capability in modeling individual attributes or their combinations, our experimental methodology is based on the principle of attribute label co-occurrence for both training and evaluation. Specifically, when conducting experiments targeting a particular attribute (e.g., "age=30"), we construct the dataset by selecting all samples annotated with the target attribute label. While these samples naturally exhibit variation in other attributes (such as emotion or accent), their shared target attribute is presumed to reflect the common feature characteristics relevant to it. This approach is necessitated by the practical constraint that real-world data rarely, if ever, contains samples that are identical across all attributes.

For the quantitative evaluation, we assess the performance of the synthesized speech by measuring its similarity to genuine samples possessing the target attribute. From the test set containing the attribute of interest, 100 samples are randomly selected to serve as the Ground Truth (GT). The similarity between each synthesized speech sample and this set of 100 GT samples is computed using relevant acoustic similarity metrics. The average of these scores is reported as the final measure of how well the synthesized speech captures the target attribute relative to the natural data. This evaluation protocol ensures a robust assessment focused on the attribute-specific (common characteristics), while accommodating the inherent diversity of other attributes in real-world data.

## 5 RESULTS AND DISCUSSION

To evaluate the effectiveness of Modular LoRAs and their arithmetic composition for controlling speech synthesis, we first adopt individual LoRA adapters to examine whether single adapter can successfully model attribute-specific speech characteristics. We then explore three composition strategies: *addition*, *subtraction*, and *scaling*, which allow us to flexibly combine or modify attribute effects without retraining. These composition mechanisms facilitate modular and interpretable control over synthesized speech. An overview of the evaluation settings is presented in Table 1.

Table 1: Overview of experimental designs.

| Control Type | Composition Strategy | Results | Control Type | Composition Strategy | Results |
|---|---|---|---|---|---|
| Single Attribute | $\mathbf{\Delta}_a$ | Section 5.1 | Addition | $w_1 \cdot \mathbf{\Delta}_{a_1} + w_2 \cdot \mathbf{\Delta}_{a_2}$ | Section 5.2 |
| Subtraction | $\mathbf{\Delta}_{a_1+a_2} - \mathbf{\Delta}_{a_1}$ | Section 5.3 | Scaling | $w \cdot \mathbf{\Delta}_a$ | Section 5.4 |

Table 2: Comparison of our TTS-Hub with ground truth (GT) and a natural-language prompting baseline (Prompt) under the single-attribute control setting. ↑ indicates metrics where higher values are better, and ↓ indicates metrics where lower values are preferred. Values in parentheses denote variances. *Note:* All results are reported based on the Parler-TTS model. Results on the alternative backbone VoiceLDM are included in the appendix and show consistent conclusions.

| Attribute | | | AMOS↑ | Win-Attr↑ | Win-Qual↑ | MCD↓ | F0-RMSE↓ | SSIM↑ |
|---|---|---|---|---|---|---|---|---|
| **Accent** | Indian | GT | 3.890(±0.0230) | - | - | - | - | - |
| | | Prompt | 2.920(±0.0433) | 0.230(±0.0010) | 0.368(±0.0030) | 7.060 | 66.798 | 0.277 |
| | | Ours | 3.750(±0.0050) | 0.638(±0.0028) | 0.550(±0.0020) | 6.086 | 57.935 | 0.374 |
| | US | GT | 3.776(±0.0041) | - | - | - | - | - |
| | | Prompt | 3.732(±0.0029) | 0.282(±0.0026) | 0.452(±0.0024) | 7.522 | 55.722 | 0.334 |
| | | Ours | 3.736(±0.0077) | 0.664(±0.0014) | 0.446(±0.0005) | 7.282 | 51.677 | 0.362 |
| | England | GT | 3.720(±0.0022) | - | - | - | - | - |
| | | Prompt | 3.510(±0.0182) | 0.282(±0.0029) | 0.354(±0.0049) | 7.643 | 65.963 | 0.329 |
| | | Ours | 3.770(±0.0108) | 0.704(±0.0020) | 0.456(±0.0056) | 7.252 | 53.823 | 0.377 |
| | Canadian | GT | 3.842(±0.0026) | - | - | - | - | - |
| | | Prompt | 3.090(±0.0380) | 0.262(±0.0049) | 0.338(±0.0017) | 7.590 | 66.140 | 0.341 |
| | | Ours | 3.614(±0.0080) | 0.620(±0.0037) | 0.458(±0.0045) | 7.138 | 51.251 | 0.407 |
| **Age** | <20 | GT | 3.872(±0.0068) | - | - | - | - | - |
| | | Prompt | 2.974(±0.0109) | 0.148(±0.0016) | 0.378(±0.0052) | 8.328 | 87.401 | 0.311 |
| | | Ours | 3.754(±0.0098) | 0.666(±0.0032) | 0.378(±0.0032) | 7.484 | 77.262 | 0.348 |
| | 20-29 | GT | 3.580(±0.0133) | - | - | - | - | - |
| | | Prompt | 3.120(±0.0158) | 0.270(±0.0072) | 0.488(±0.0020) | 8.408 | 80.402 | 0.310 |
| | | Ours | 3.390(±0.0080) | 0.658(±0.0051) | 0.438(±0.0023) | 7.389 | 73.073 | 0.338 |
| | 30-39 | GT | 3.500(±0.0063) | - | - | - | - | - |
| | | Prompt | 3.400(±0.0213) | 0.260(±0.0017) | 0.596(±0.0039) | 7.714 | 62.773 | 0.328 |
| | | Ours | 3.430(±0.0095) | 0.626(±0.0060) | 0.386(±0.0019) | 7.412 | 50.999 | 0.374 |
| | >60 | GT | 3.906(±0.0107) | - | - | - | - | - |
| | | Prompt | 2.990(±0.0180) | 0.240(±0.0034) | 0.340(±0.0015) | 7.250 | 66.951 | 0.325 |
| | | Ours | 3.744(±0.0049) | 0.698(±0.0021) | 0.536(±0.0029) | 7.585 | 58.022 | 0.373 |
| **Overall** | | GT | 3.790±(0.0102) | - | - | - | - | - |
| | | Prompt | 3.344±(0.0196) | 0.254(±0.0038) | 0.376(±0.0061) | 7.330 | 66.429 | 0.314 |
| | | Ours | 3.661±(0.0102) | 0.600(±0.0051) | 0.409(±0.0079) | 6.953 | 57.800 | 0.363 |

## 5.1 EVALUATION UNDER SINGLE-ATTRIBUTE CONTROL

**LoRA-based control significantly outperforms Prompt-only methods**. We implement single-attribute controllable synthesis by prompting the backbone model equipped with attribute-specific LoRA modules. Due to the large number of attributes, we sample 16 audio clips per attribute for human evaluation. All other metrics are computed over the entire test set. To ensure the stability of the Large Language Model-based metrics, each pairwise comparison is evaluated five times. As shown in Table 2, the LoRA-based approach consistently outperforms prompt-based control across multiple evaluation metrics for the vast majority of attributes. This advantage is particularly evident in two metrics specifically designed to assess attribute expressiveness. For instance, for the highly distinguishable attribute "age over 60", our method significantly surpasses the prompt-based approach, improving AMOS from 2.990 to 3.744 and Win-Attr from 0.240 to 0.698. Although in some cases, such as the "US accent" attribute, our approach shows a slight drop in Win-Qual compared to the prompt-based method, it still demonstrates a clear advantage in attribute expressiveness, improving Win-Attr from 0.282 to 0.664. These results underscore the effectiveness of LoRA modules in capturing fine-grained acoustic characteristics. Furthermore, to mitigate potential bias stemming from the distinctiveness of individual attributes, we construct an attribute-invariant test set by randomly sampling 128 utterances from all single-attribute generations. As shown in the last three rows

Table 3: AMOS scores with varying weights between two LoRA modules. AMOS-$a_1$ and AMOS-$a_2$ reflect the perceived strength of attributes $a_1$ and $a_2$, with higher scores indicating stronger expression of the target attribute.

| Composition | AMOS-$a_1$ ↑ | AMOS-$a_2$ ↑ | Composition | AMOS-$a_1$ ↑ | AMOS-$a_2$ ↑ |
|---|---|---|---|---|---|
| $0.0 \cdot \boldsymbol{\Delta}_{a_1} + 1.0 \cdot \boldsymbol{\Delta}_{a_2}$ | 1.803(±0.0577) | 3.717(±0.0084) | $0.1 \cdot \boldsymbol{\Delta}_{a_1} + 0.9 \cdot \boldsymbol{\Delta}_{a_2}$ | 1.761(±0.0124) | 3.644(±0.0163) |
| $0.2 \cdot \boldsymbol{\Delta}_{a_1} + 0.8 \cdot \boldsymbol{\Delta}_{a_2}$ | 1.834(±0.0263) | 3.549(±0.0312) | $0.3 \cdot \boldsymbol{\Delta}_{a_1} + 0.7 \cdot \boldsymbol{\Delta}_{a_2}$ | 2.409(±0.0875) | 3.456(±0.0177) |
| $0.4 \cdot \boldsymbol{\Delta}_{a_1} + 0.6 \cdot \boldsymbol{\Delta}_{a_2}$ | 2.419(±0.0484) | 3.126(±0.0434) | $0.6 \cdot \boldsymbol{\Delta}_{a_1} + 0.4 \cdot \boldsymbol{\Delta}_{a_2}$ | 3.185(±0.0291) | 2.761(±0.0462) |
| $0.7 \cdot \boldsymbol{\Delta}_{a_1} + 0.3 \cdot \boldsymbol{\Delta}_{a_2}$ | 3.387(±0.0182) | 2.337(±0.0507) | $0.8 \cdot \boldsymbol{\Delta}_{a_1} + 0.2 \cdot \boldsymbol{\Delta}_{a_2}$ | 3.565(±0.0292) | 2.323(±0.0374) |
| $0.9 \cdot \boldsymbol{\Delta}_{a_1} + 0.1 \cdot \boldsymbol{\Delta}_{a_2}$ | 3.619(±0.0292) | 1.874(±0.0646) | $1.0 \cdot \boldsymbol{\Delta}_{a_1} + 0.0 \cdot \boldsymbol{\Delta}_{a_2}$ | 3.832(±0.0186) | 1.383(±0.0607) |

Table 4: Performance of multi-attribute control. #Attr denotes the number of composite attributes.

| Method | #Attr | AMOS↑ | Win-Attr↑ | Win-Qual↑ | #Attr | AMOS↑ | Win-Attr↑ | Win-Qual↑ |
|---|---|---|---|---|---|---|---|---|
| Prompt | 1 | 3.344 (± 0.0196) | 0.254 (± 0.0038) | 0.376 (± 0.0061) | 2 | 2.937 (± 0.0452) | 0.214 (± 0.0013) | 0.367 (± 0.0080) |
| Ours | | 3.661 (± 0.0102) | 0.600 (± 0.0051) | 0.409 (± 0.0079) | | 3.625 (± 0.0077) | 0.594 (± 0.0118) | 0.408 (± 0.0074) |
| Prompt | 3 | 2.656 (± 0.0529) | 0.217 (± 0.0018) | 0.333 (± 0.0051) | 4 | 2.196 (± 0.0425) | 0.296 (± 0.0030) | 0.337 (± 0.0020) |
| Ours | | 3.281 (± 0.0322) | 0.604 (± 0.0152) | 0.379 (± 0.0048) | | 2.525 (± 0.0625) | 0.397 (± 0.0039) | 0.362 (± 0.0016) |

of Table 2, the overall performance demonstrates that our approach consistently achieves superior attribute expressiveness.

## 5.2 Evaluation under Multiple-Attribute Control

**Two-Attribute Control.** We investigate the compositional behavior of fine-grained attributes under LoRA-based control by combining two specific attributes: accent ($a_1$) and ($a_2$). In total, we conduct four groups of fine-grained attribute combination experiments. The corresponding LoRA modules are fused through weighted summation, with weights ranging from 0.1 to 0.9 in increments of 0.2, to examine the impact of fine-grained weighting on synthesis quality. We randomly sample 128 audio clips from the combined set of all four groups for evaluation. As shown in Table 3, the results reveal a clear trade-off between the two attributes: increasing the weight of $a_1$ consistently improves the AMOS-$a_1$ score, while AMOS-$a_2$ gradually declines. This trend highlights the controllability and flexibility of our approach, enabling smooth and continuous transitions between multiple attribute representations. Moreover, the disentangled behavior suggests that LoRA modules effectively preserve attribute-specific information, thereby better supporting compositional control across features.

**Multi-Attribute Control.** In order to explore the upper limit of attribute composition using LoRA modules, we conducted experiments with an increased number of attributes, focusing specifically on combinations involving three and four features. Similarly, we mixed all multi-attribute combinations and randomly sampled 128 clips for evaluation. The same test set sampling strategy is applied for the subsequent subtractive and scaling experiments. The overall performance across different configurations is summarized in Table 4. The results indicate that composing three attributes with LoRA modules still yields strong synthesis performance, particularly in metrics closely related to attribute expressiveness such as AMOS and Win-Attr. However, when the number of attributes increases to four, we observe a notable degradation in synthesis quality. This finding suggests that introducing too many fused LoRA modules may impair the performance of the backbone model, which is consistent with the conclusions drawn in (Hu et al., 2022). Despite this limitation, the three-feature combination already covers a broad range of personalized synthesis scenarios.

## 5.3 Exploring Attribute Subtraction

To investigate whether specific attributes can be selectively attenuated from a composite LoRA representation, we conducted arithmetic subtraction experiments. Specifically, we subtracted an attribute-specific LoRA module (e.g., $\boldsymbol{\Delta}a_2$) from a composite module trained on a combined attribute set (e.g., $\boldsymbol{\Delta}a_1 + a_2$). This setup aims to assess whether subtraction can suppress the influence of the removed attribute while retaining the representation of the remaining one.

As shown in Table 5, subtracting $\boldsymbol{\Delta}_{a_2}$ from $\boldsymbol{\Delta}_{a_1+a_2}$ results in a noticeable decrease in AMOS for attribute $a_2$, dropping from 3.548 to 2.889, which indicates successful attenuation of the $a_2$-related representation. Meanwhile, we observe a degradation in AMOS for attribute $a_1$, decreasing from

3.615 to 3.317. This suggests that subtractive composition between LoRA modules may partially impair the original attribute expressiveness within the composite embedding. Additionally, a decline in overall Win-Qual scores is observed compared to the unmodified composite $\boldsymbol{\Delta}_{a_1+a_2}$, implying that such arithmetic operations may compromise the general synthesis quality. This trade-off highlights the inherent limitations of arithmetic operations within the LoRA parameter space to some extent.

Table 5: Evaluation of composition between two LoRA modules. AMOS-$a_1$ and AMOS-$a_2$ assess the presence of attributes $a_1$ and $a_2$, respectively. Win-Attr-$a_1$ and Win-Attr-$a_2$ indicate the pairwise win rates for correctly expressing $a_1$ and $a_2$. Win-Qual measures overall audio quality preference.

| Subtraction | AMOS-$a_1$ ↑ | AMOS-$a_2$ ↑ | Win-Attr-$a_1$ ↑ | Win-Attr-$a_2$ ↑ | Win-Qual↑ |
|---|---|---|---|---|---|
| GT | 3.891(±0.0180) | 3.813(±0.0206) | - | - | - |
| Prompt | 3.270(±0.0114) | 3.066(±0.0280) | 0.276(±0.0036) | 0.213(±0.0026) | 0.390(±0.0052) |
| $\boldsymbol{\Delta}_{a_1+a_2}$ | 3.615(±0.0137) | 3.548(±0.0249) | 0.341(±0.0031) | 0.342(±0.0050) | 0.382(±0.0019) |
| $\boldsymbol{\Delta}_{a_1}$ | 3.450(±0.0220) | 2.989(±0.1300) | 0.338(±0.0030) | 0.268(±0.0026) | 0.364(±0.0032) |
| $\boldsymbol{\Delta}_{a_2}$ | 2.817(±0.0455) | 3.478(±0.0352) | 0.241(±0.0029) | 0.375(±0.0025) | 0.366(±0.0021) |
| $\boldsymbol{\Delta}_{a_1+a_2} - \boldsymbol{\Delta}_{a_2}$ | 3.317(±0.0215) | 2.889(±0.1549) | 0.315(±0.0019) | 0.278(±0.0022) | 0.335(±0.0018) |
| $\boldsymbol{\Delta}_{a_1+a_2} - \boldsymbol{\Delta}_{a_1}$ | 2.984(±0.0477) | 3.494(±0.0417) | 0.259(±0.0030) | 0.353(±0.0030) | 0.347(±0.0038) |

## 5.4 SCALING EXPERIMENTS ON INDIVIDUAL ATTRIBUTES

To further investigate whether LoRA modules exhibit scalable control over individual attributes, we conducted a series of scaling experiments using LoRA modules fine-tuned on single-attribute datasets. Module scaling involves applying a scalar coefficient to modulate the influence of an attribute-specific LoRA module on the backbone model.

Experimental results in Figure 4 reveal that when the scaling factor approaches zero, the target attribute's expressiveness is substantially attenuated. As the coefficient increases, the attribute becomes progressively more salient. However, beyond a certain threshold, further amplification yields diminishing perceptual gains, indicating a saturation effect in controllability. In additional, we examined the impact of negative scaling values. While negative coefficients

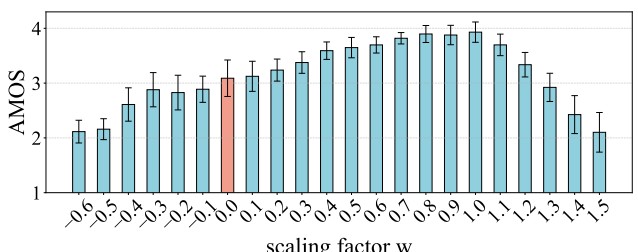

Figure 4: AMOS results under different scaling factors, where red denotes the absence of any attribute-specific LoRA modules.

reduced the prominence of the attribute to some extent, they did not induce an inverse or antagonistic acoustic effect. This suggests that the scaling mechanism based on LoRA modules indeed enables continuous and degree-controllable modulation of attribute strength, but lacks the ability to reverse the semantic polarity of attribute expression.

## 6 CONCLUSION

This paper introduces TTS-Hub, a novel framework for controllable text-to-speech synthesis that leverages modular LoRA adapters and their arithmetic composition to enable fine-grained, flexible, and extensible control over speech attributes. By reconstructing an attribute-specific Data Hub and fine-tuning LoRA modules accordingly, we build a versatile LoRA Hub in which each adapter is specialized for a distinct speech attribute. During inference, TTS-Hub composes selected adapters through simple arithmetic operations, allowing intuitive multi-attribute control without retraining the backbone model. Experimental results show that individual adapters achieve precise control over single attributes, while their compositions support interpretable and robust multi-attribute synthesis. Furthermore, the modular architecture enables seamless integration of new attributes, highlighting the potential of LoRA-based modularity as a powerful and scalable paradigm for developing controllable TTS systems.

ETHICS STATEMENT

Our research utilizes only open-source audio datasets for experimental purposes. All data have been thoroughly anonymized by their original providers, ensuring that no personal identifiers are present. As such, our use of these datasets does not involve any infringement of personal privacy.

Furthermore, the speech synthesis framework proposed in this study is intended solely for academic and scientific research. Any application of this technology must comply with legal and ethical standards. We explicitly prohibit its use for unlawful activities or any purpose that may cause harm to individuals or society.

REPRODUCIBILITY STATEMENT

This study is committed to ensuring the reproducibility of its findings. To this end, we have ensured full transparency regarding data, methodology, and implementation. All experiments are based on publicly accessible datasets, which are detailed in Section 4.1. A complete description of the model architectures, evaluation metrics, and experimental procedures is provided in Section 4.1, 4.2 and 4.3. Furthermore, the complete source code, including configurations for training and inference, is available in the supplementary materials to facilitate the replication of our results.

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

## THE USE OF LARGE LANGUAGE MODELS

**Using an LLM to help with paper writing**    During the preparation of this work, the authors utilized Large Language Models for language polishing, improving the structural clarity of the manuscript, and refining the formal expression of individual sentences. The use of Large Language Models did not influence the substantive content of the study and served solely as a writing aid.

## OVERALL OF THE APPENDIX

The supplementary material is organized into the following sections:

- Section A: Single-Attribute Control.
- Section B: Multiple-Attribute Control
- Section C: Fine-Grained Attributes Control
- Section D: Exploring Attribute Subtraction
- Section E: Exploring Attribute Scaling
- Section F: Prompts Utilized in the Evaluation Process and Hyperparameter Description
- Section J: Failure Modes Analysis
- Section H: Cosine Similarity Between all Adapter
- Section I: Visualization of AMOS Scores from Table 3, showing the impact of varying weights between two LoRA modules

## A: SINGLE-ATTRIBUTE CONTROL

### SINGLE-ATTRIBUTE CONTROL WITH PARLER-TTS

This section reports the complete experimental results for the single-attribute control setting, where we assess the effectiveness of LoRA-based control using the Parler-TTS model. We compare our proposed framework (TTS-Hub) against the ground truth (GT) and a natural language prompting baseline (Prompt). Metrics marked with ↑ indicate that higher values are better, while those marked with ↓ indicate that lower values are preferred. Values in parentheses denote the variance across samples. For detailed results, please refer to Table 7 and Table 8.

The experimental results demonstrate that our TTS-Hub exhibits significant advantages in fine-grained attribute control, providing strong evidence for the effectiveness of LoRA modules in learning and representing speech characteristics.

### SINGLE-ATTRIBUTE CONTROL WITH VOICELDM

In this set of experiments, we adopt the diffusion-based VoiceLDM as the backbone model to evaluate single-attribute control. For detailed results, please refer to Table 9 and Table 10. The results demonstrate that our proposed TTS-Hub exhibits strong architectural robustness and can be effectively integrated with mainstream TTS frameworks.

## B: MULTIPLE-ATTRIBUTE CONTROL

### TWO-ATTRIBUTE CONTROL

To evaluate the arithmetic compositionality of LoRA modules, we conduct experiments by combining multiple pairs of LoRA modules, each corresponding to a distinct speech attribute. Similar to the single-attribute setting, we perform evaluations on both the Parler-TTS and VoiceLDM backbones. We report the performance of various attribute pair combinations. As shown in Table 11 and Table 12, combining two LoRA modules enables coherent and expressive speech synthesis in most cases, while maintaining effective control over each individual attribute. These results underscore the flexibility of modular attribute composition in LoRA-based TTS systems.

### TRIPLE-ATTRIBUTE CONTROL

We investigate whether LoRA-based controllable generation can retain strong attribute expressiveness as the number of combined attributes increases by conducting triple-attribute control experiments on both backbone architectures. The results in Table 13 and Table 14 demonstrate that, while triple-attribute fusion generally preserves attribute expressiveness and intelligibility, a moderate degradation in overall synthesis quality is observed compared to the single- and two-attribute settings. This indicates that increasing the number of fused attributes may slightly compromise the synthesis quality of the backbone, though the performance remains within an acceptable range. These findings underscore the potential of LoRA modularity for controllable TTS synthesis across diverse and complex attribute configurations.

### QUADRUPLE-ATTRIBUTE CONTROL

To further explore the upper limits of attribute composition under LoRA-based controllable synthesis, this set of experiments involves the fusion of four independently trained LoRA modules for simultaneous multi-attribute control in both Parler-TTS and VoiceLDM. Each configuration combines four distinct attributes to evaluate the scalability of modular LoRA-based generation.

As shown in Table 15 and Table 16, overall synthesis quality declines more noticeably compared to the fusion of one, two, or three attributes. While basic attribute expressiveness is preserved in some configurations, increased interference among modules often leads to reduced naturalness and stability. These findings reveal the limitations of naïve LoRA stacking at larger scales and underscore the need for more sophisticated fusion mechanisms or advanced composition strategies. Despite the observed degradation, this setting demonstrates the scalability potential of LoRA for fine-grained TTS control and motivates future research on controllable synthesis in more complex attribute spaces.

## C: FINE-GRAINED ATTRIBUTES CONTROL

To investigate the impact of varying fusion weights assigned to different attributes during LoRA module composition, we conduct experiments using a finer-grained set of fusion weights in the two-attribute control setting. The results are presented in Tables 17a, 17b, 18a and 18b.

The findings show that increasing the weight of a specific attribute leads to a more pronounced manifestation of that attribute in the synthesized speech. Conversely, reducing the corresponding weight diminishes its expressiveness. These results confirm that LoRA-based fusion maintains strong controllability even at a fine-grained level, demonstrating its potential as a promising solution for precise and flexible voice editing tasks.

## D: EXPLORING ATTRIBUTE SUBTRACTION

To explore whether specific speech attributes can be attenuated in the synthesized output through subtractive operations between LoRA modules, we conduct experiments on four attribute combinations. The goal is to assess whether such subtractive composition can support more flexible and targeted speech editing. Please refer to Tables 19, 20, 21, 22for detailed results.

The results indicate that subtractive operations are indeed effective in weakening the influence of targeted attributes in the output speech. However, this manipulation also introduces collateral effects, occasionally diminishing the expressiveness of remaining desired attributes or affecting overall speech naturalness. These findings suggest that while attribute subtraction via LoRA modules provides a viable pathway for fine-grained semantic control, it also reveals the inherent entanglement between attributes in the learned representation. Please see Table 23.

## E: EXPLORING ATTRIBUTE SCALING

In this set of experiments, we investigate whether scaling operations applied to individual LoRA modules can achieve smooth and continuous modulation of specific speech attributes. To this end, we select four prominent attributes and conduct controlled scaling experiments on their corresponding LoRA modules. The results are presented in Fig. 5a, 5b, 5c, and 5d.

The findings reveal three key observations. First, attribute expressiveness can be modulated in a relatively smooth and continuous manner by scaling the weight coefficients of the corresponding LoRA modules. Second, when the scaling factor exceeds a certain threshold, the overall speech quality deteriorates significantly. Third, applying negative scaling coefficients does not result in an inversion of the target attribute, contrary to initial expectations.

These results highlight the strong potential of LoRA-based control for fine-grained speech editing, while also emphasizing the critical importance of carefully selecting appropriate scaling factors to ensure stable and interpretable attribute manipulation.

## F: PROMPTS UTILIZED IN THE EVALUATION PROCESS AND HYPERPARAMETER DESCRIPTION

When utilizing the GPT-4o-Audio API for speech evaluation, we design two distinct prompting strategies tailored to different evaluation dimensions.

### FOR WIN-ATTR

---

**LLM-Win-Qual Prompt:**
You are a speech synthesis expert with extensive experience in evaluating audio quality from multiple perspectives.
We have developed two different text-to-speech (TTS) systems, and each system has generated one audio sample.
The order of the audio samples has been randomized (referred to as Audio Sample 1 and Audio Sample 2).
Please listen carefully to both samples and evaluate their overall audio quality.
Then, based on your analysis, indicate which sample (Audio Sample 1 or Audio Sample 2 or tie) you believe demonstrates higher overall quality.
If both samples are of comparable quality and you find it difficult to determine a clear winner, respond with"tie" instead of guessing.
Remember, only return me your final choice (Audio Sample 1 or Audio Sample 2 or tie)!

---

FOR WIN-QUAL

---

**LLM-Win-Attr Prompt:**

Assume you're now a speech expert with extensive experience in evaluating audio based on specific linguistic features, such as attribute.

You are presented with two audio clips, each generated by a different Text-to-Speech (TTS) system. The order of the clips has been randomized(referred to as Audio Sample 1 and Audio Sample 2), so you do not know which system produced which clip.

Your task is to carefully analyze both audio clips and assess how well each one conforms to the specified feature: attribute.

Focus on aspects such as pronunciation, intonation, and rhythm that are characteristic of the feature. After your analysis, determine which audio clip (Audio Sample 1 or Audio Sample 2 or tie) better matches the specified feature and provide your conclusion.

If both samples are of comparable quality and you find it difficult to determine a clear winner, respond with"tie" instead of guessing.

Remember, only return me your final choice (Audio Sample 1 or Audio Sample 2 or tie)!

---

**Parler-TTS Model**  We integrated LoRA modules into the decoder component of the Parler-TTS model. The ParlerTTSDecoder architecture comprises three main components: an `embed_tokens` layer, an `embed_positions` layer, and a 24-layer ParlerTTSDecoderLayer structure. Each ParlerTTSDecoderLayer contains self-attention (`self_attn`), encoder-attention (`encoder_attn`), activation function, and a two-layer feed-forward network (FC).

Specifically, LoRA was applied to the following components within each ParlerTTSDecoderLayer:

- Self-attention projections: `k_proj`, `q_proj`, `v_proj`, and `out_proj`
- Both fully connected layers in the feed-forward network

This integration enhances the trainable capacity of both self-attention mechanisms and feed-forward components.

**VoiceLDM Model**  For the VoiceLDM model, LoRA was incorporated into the UNet module. The integration focuses on three critical components:

- Downsampling blocks: `CrossAttnDownBlock2D`
- Upsampling blocks: `CrossAttnUpBlock2D`
- Intermediate block: `UNetMidBlock2DCrossAttn`

Within these blocks, LoRA was applied to:

- Transformer query, key, and value projections
- Attention output projections
- Input and output convolutional layers
- Linear layers in FeedForward networks

**Hyperparameter Configuration**  The key LoRA hyperparameters, rank ($r$) and scaling factor ($\alpha$), were determined through systematic ablation studies as shown in Table 6. The optimal combination that yielded superior performance was selected for final model training.

**LoRA Integration Specifications**

Table 6: WER results of LoRA fine-tuned models with different $(r, \alpha)$ configurations compared to the base model.

| Model / $(r,\alpha)$ | (4,8) | (8,16) | (16,32) | (32,64) | (64,128) | (128,256) |
|---|---|---|---|---|---|---|
| LoRA (Indian Accent) WER | 17.55 | 25.02 | 13.50 | 20.27 | 13.13 | 13.06 |
| LoRA (20 years old) WER | 18.26 | 13.82 | 9.80 | 10.23 | 9.58 | 9.60 |
| Base model WER | 10.17 | 10.17 | 10.17 | 10.17 | 10.17 | 10.17 |

Table 7: Single LoRA Parler-tts table1

| Metrics | | | AMOS↑ | Win-Attr↑ | Win-Qual↑ | MCD↓ | F0-RMSE↓ | SSIM↑ |
|---|---|---|---|---|---|---|---|---|
| **Accent** | India | GT | 3.890(±0.0230) | - | - | - | - | - |
| | | Prompt | 2.920(±0.0433) | 0.2 30(±0.0010) | 0.368(±0.0030) | 7.060 | 66.798 | 0.277 |
| | | Ours | 3.750(±0.0050) | 0.638(±0.0028) | 0.550(±0.0020) | 6.086 | 57.935 | 0.374 |
| | US | GT | 3.776(±0.0041) | - | - | - | - | - |
| | | Prompt | 3.732(±0.0029) | 0.282(±0.0026) | 0.452(±0.0024) | 7.522 | 55.722 | 0.334 |
| | | Ours | 3.736(±0.0077) | 0.664(±0.0014) | 0.446(±0.0005) | 7.282 | 51.677 | 0.362 |
| | England | GT | 3.720(±0.0022) | - | - | - | - | - |
| | | Prompt | 3.510(±0.0182) | 0.282(±0.0029) | 0.354(±0.0049) | 7.643 | 65.963 | 0.329 |
| | | Ours | 3.770(±0.0108) | 0.704(±0.0020) | 0.456(±0.0056) | 7.252 | 53.823 | 0.377 |
| | Canadian | GT | 3.842(±0.0026) | - | - | - | - | - |
| | | Prompt | 3.090(±0.0380) | 0.262(±0.0049) | 0.338(±0.0017) | 7.590 | 66.140 | 0.341 |
| | | Ours | 3.614(±0.0080) | 0.620(±0.0037) | 0.458(±0.0045) | 7.138 | 51.251 | 0.407 |
| | African | GT | 3.842(±0.0026) | - | - | - | - | - |
| | | Prompt | 3.286(±0.0165) | 0.250(±0.0010) | 0.428(±0.0035) | 8.537 | 74.751 | 0.301 |
| | | Ours | 3.664(±0.0097) | 0.660(±0.0004) | 0.418(±0.0216) | 7.850 | 65.504 | 0.328 |
| | Australian | GT | 3.842(±0.0026) | - | - | - | - | - |
| | | Prompt | 3.252(±0.0058) | 0.222(±0.0020) | 0.298(±0.0051) | 8.008 | 74.914 | 0.335 |
| | | Ours | 3.684(±0.0175) | 0.640(±0.0015) | 0.474(±0.0258) | 7.454 | 69.470 | 0.376 |
| | German | GT | 3.842(±0.0026) | - | - | - | - | - |
| | | Prompt | 3.180(±0.0233) | 0.250(±0.0067) | 0.452(±0.0134) | 7.931 | 76.689 | 0.284 |
| | | Ours | 3.230(±0.0245) | 0.570(±0.0050) | 0.458(±0.0296) | 7.303 | 71.360 | 0.454 |
| | Irish | GT | 3.842(±0.0026) | - | - | - | - | - |
| | | Prompt | 3.250(±0.0138) | 0.308(±0.0031) | 0.372(±0.0120) | 7.002 | 59.462 | 0.360 |
| | | Ours | 3.660(±0.0168) | 0.610(±0.0067) | 0.436(±0.0197) | 5.954 | 23.537 | 0.519 |
| | New Zealand | GT | 3.842(±0.0026) | - | - | - | - | - |
| | | Prompt | 3.240(±0.0268) | 0.182(±0.0012) | 0.316(±0.0117) | 7.814 | 75.104 | 0.321 |
| | | Ours | 3.770(±0.0113) | 0.650(±0.0017) | 0.380(±0.0036) | 7.107 | 70.368 | 0.381 |
| | Scottish | GT | 3.842(±0.0026) | - | - | - | - | - |
| | | Prompt | 2.990(±0.0305) | 0.208(±0.0029) | 0.464(±0.0043) | 7.344 | 68.867 | 0.327 |
| | | Ours | 3.632(±0.0083) | 0.578(±0.0060) | 0.414(±0.0081) | 7.512 | 60.210 | 0.334 |
| **Age** | <20 | GT | 3.872(±0.0068) | - | - | - | - | - |
| | | Prompt | 2.974(±0.0109) | 0.148(±0.0016) | 0.378(±0.0052) | 8.328 | 87.401 | 0.311 |
| | | Ours | 3.754(±0.0098) | 0.666(±0.0032) | 0.378(±0.0032) | 7.484 | 77.262 | 0.348 |
| | 20-29 | GT | 3.580(±0.0133) | - | - | - | - | - |
| | | Prompt | 3.120(±0.0158) | 0.270(±0.0072) | 0.488(±0.0020) | 8.408 | 80.402 | 0.310 |
| | | Ours | 3.390(±0.0080) | 0.658(±0.0051) | 0.438(±0.0023) | 7.389 | 73.073 | 0.338 |
| | 30-39 | GT | 3.500(±0.0063) | - | - | - | - | - |
| | | Prompt | 3.400(±0.0213) | 0.260(±0.0017) | 0.596(±0.0039) | 7.714 | 62.773 | 0.328 |
| | | Ours | 3.430(±0.0095) | 0.626(±0.0060) | 0.386(±0.0019) | 7.412 | 50.999 | 0.374 |
| | 40-49 | GT | 3.500(±0.0063) | - | - | - | - | - |
| | | Prompt | 3.240(±0.0230) | 0.228(±0.0028) | 0.454(±0.0158) | 7.792 | 77.185 | 0.306 |
| | | Ours | 3.140(±0.0093) | 0.552(±0.0071) | 0.544(±0.0146) | 7.533 | 59.221 | 0.378 |
| | 50-59 | GT | 3.500(±0.0063) | - | - | - | - | - |
| | | Prompt | 3.130(±0.0195) | 0.244(±0.0040) | 0.398(±0.0361) | 7.330 | 70.847 | 0.313 |
| | | Ours | 3.496(±0.0115) | 0.618(±0.0091) | 0.396(±0.0182) | 7.368 | 63.911 | 0.379 |
| | >60 | GT | 3.906(±0.0107) | - | - | - | - | - |
| | | Prompt | 2.990(±0.0180) | 0.240(±0.0034) | 0.340(±0.0015) | 7.250 | 66.951 | 0.325 |
| | | Ours | 3.744(±0.0049) | 0.698(±0.0021) | 0.536(±0.0029) | 7.585 | 58.022 | 0.373 |

Table 8: Single LoRA Parler-tts table2

| Metrics | | | AMOS↑ | Win-Attr↑ | Win-Qual↑ | MCD↓ | F0-RMSE↓ | SSIM↑ |
|---|---|---|---|---|---|---|---|---|
| **Emotion** | Angry | GT | 3.872(±0.0068) | - | - | - | - | - |
| | | Prompt | 2.880±(0.0570) | 0.212(±0.0026) | 0.314(±0.0017) | 6.793 | 66.863 | 0.321 |
| | | Ours | 3.306(±0.0057) | 0.632(±0.0014) | 0.362(±0.0051) | 6.685 | 56.836 | 0.342 |
| | Contempt | GT | 3.580(±0.0133) | - | - | - | - | - |
| | | Prompt | 3.140±(0.0193) | 0.236(±0.0051) | 0.400(±0.0057) | 7.055 | 55.141 | 0.276 |
| | | Ours | 3.170(±0.0172) | 0.586(±0.0054) | 0.382(±0.0029) | 6.399 | 47.880 | 0.308 |
| | Disgusted | GT | 3.500(±0.0063) | - | - | - | - | - |
| | | Prompt | 2.880±(0.0533) | 0.222(±0.0028) | 0.226(±0.0027) | 7.547 | 73.899 | 0.209 |
| | | Ours | 3.050(±0.0138) | 0.566(±0.0097) | 0.242(±0.0019) | 7.027 | 51.770 | 0.275 |
| | Fear | GT | 3.500(±0.0063) | - | - | - | - | - |
| | | Prompt | 3.000±(0.0313) | 0.430(±0.0016) | 0.292(±0.0070) | 7.301 | 73.298 | 0.209 |
| | | Ours | 3.350(±0.0125) | 0.400(±0.0062) | 0.316(±0.0019) | 7.041 | 72.044 | 0.261 |
| | Happy | GT | 3.500(±0.0063) | - | - | - | - | - |
| | | Prompt | 2.900±(0.0113) | 0.234(±0.0023) | 0.238(±0.0075) | 6.976 | 79.335 | 0.293 |
| | | Ours | 3.586(±0.0082) | 0.648(±0.0018) | 0.306(±0.0037) | 6.564 | 71.381 | 0.384 |
| | Neutral | GT | 3.500(±0.0063) | - | - | - | - | - |
| | | Prompt | 3.700±(0.0015) | 0.254(±0.0026) | 0.306(±0.0074) | 6.555 | 44.267 | 0.328 |
| | | Ours | 3.698(±0.0060) | 0.612(±0.0036) | 0.312(±0.0025) | 6.521 | 49.650 | 0.330 |
| | Sad | GT | 3.500(±0.0063) | - | - | - | - | - |
| | | Prompt | 2.670±(0.0145) | 0.230(±0.0026) | 0.316(±0.0056) | 6.576 | 48.579 | 0.313 |
| | | Ours | 3.728(±0.0035) | 0.392(±0.0076) | 0.386(±0.0065) | 6.540 | 41.201 | 0.353 |
| | Surprised | GT | 3.906(±0.0107) | - | - | - | - | - |
| | | Prompt | 2.674±(0.0110) | 0.200(±0.0007) | 0.252(±0.0039) | 6.843 | 96.548 | 0.292 |
| | | Ours | 3.750(±0.0027) | 0.444(±0.0041) | 0.282(±0.0045) | 6.709 | 89.640 | 0.346 |
| **Gender** | Female | GT | 3.872(±0.0068) | - | - | - | - | - |
| | | Prompt | 4.900±(0.0500) | 0.340(±0.0118) | 0.394(±0.0058) | 7.992 | 58.487 | 0.330 |
| | | Ours | 5.000(±0.0000) | 0.590(±0.0134) | 0.406(±0.0203) | 7.288 | 53.530 | 0.387 |
| | Male | GT | 3.872(±0.0068) | - | - | - | - | - |
| | | Prompt | 5.000±(0.0000) | 0.306(±0.0033) | 0.440(±0.0025) | 7.668 | 58.527 | 0.344 |
| | | Ours | 4.900(±0.0500) | 0.668(±0.0017) | 0.464(±0.0024) | 7.193 | 46.722 | 0.385 |
| **Energy** | High | GT | 3.872(±0.0068) | - | - | - | - | - |
| | | Prompt | 3.600±(0.0063) | 0.264(±0.0068) | 0.320(±0.0031) | 6.930 | 67.648 | 0.341 |
| | | Ours | 3.746(±0.0033) | 0.570(±0.0097) | 0.340(±0.0040) | 6.871 | 50.762 | 0.356 |
| | Normal | GT | 3.872(±0.0068) | - | - | - | - | - |
| | | Prompt | 3.550±(0.0145) | 0.284(±0.0062) | 0.442(±0.0037) | 6.577 | 57.822 | 0.349 |
| | | Ours | 3.622(±0.0030) | 0.574(±0.0022) | 0.426(±0.0064) | 6.512 | 54.277 | 0.354 |
| | Low | GT | 3.872(±0.0068) | - | - | - | - | - |
| | | Prompt | 3.702±(0.0019) | 0.314(±0.0199) | 0.394(±0.0057) | 6.433 | 52.847 | 0.326 |
| | | Ours | 3.676(±0.0031) | 0.538(±0.0182) | 0.418(±0.0006) | 6.118 | 50.705 | 0.345 |
| **Pitch** | High | GT | 3.872(±0.0068) | - | - | - | - | - |
| | | Prompt | 3.806±(0.0172) | 0.264(±0.0007) | 0.486(±0.0013) | 6.694 | 70.196 | 0.325 |
| | | Ours | 3.786(±0.0017) | 0.576(±0.0050) | 0.448(±0.0047) | 6.245 | 69.723 | 0.364 |
| | Normal | GT | 3.872(±0.0068) | - | - | - | - | - |
| | | Prompt | 3.530±(0.0057) | 0.264(±0.0027) | 0.424(±0.0020) | 6.671 | 53.482 | 0.334 |
| | | Ours | 3.580(±0.0180) | 0.634(±0.0068) | 0.412(±0.0068) | 6.482 | 46.770 | 0.362 |
| | Low | GT | 3.872(±0.0068) | - | - | - | - | - |
| | | Prompt | 3.762±(0.0039) | 0.202(±0.0012) | 0.308(±0.0021) | 6.666 | 38.810 | 0.366 |
| | | Ours | 3.734(±0.0047) | 0.618(±0.0018) | 0.408(±0.0046) | 6.604 | 39.073 | 0.375 |
| **Overall** | | GT | 3.790(±0.0196) | - | - | - | - | - |
| | | Prompt | 3.344±(0.0196) | 0.254(±0.0038) | 0.377(±0.0061) | 7.330 | 66.429 | 0.314 |
| | | Ours | 3.661(±0.0102) | 0.600(±0.0051) | 0.409(±0.0076) | 6.953 | 57.800 | 0.363 |

Table 9: Single LoRA VoiceLDM table1

| Metrics | | | AMOS↑ | Win-Attr↑ | Win-Qual↑ | MCD↓ | F0-RMSE↓ | SSIM↑ |
|---|---|---|---|---|---|---|---|---|
| **Accent** | India | GT | 3.890(±0.0230) | - | - | - | - | - |
| | | Prompt | 2.979(±0.0525) | 0.216±(0.0014) | 0.350±(0.0061) | 9.666 | 65.030 | 0.264 |
| | | Ours | 3.391(±0.0102) | 0.638±(0.0018) | 0.566±(0.0020) | 8.598 | 59.025 | 0.343 |
| | US | GT | 3.776(±0.0041) | - | - | - | - | - |
| | | Prompt | 3.625(±0.0119) | 0.286±(0.0016) | 0.424±(0.0041) | 7.412 | 70.939 | 0.305 |
| | | Ours | 3.663(±0.0035) | 0.572±(0.0007) | 0.440±(0.0006) | 7.206 | 68.877 | 0.334 |
| | England | GT | 3.720(±0.0022) | - | - | - | - | - |
| | | Prompt | 3.192(±0.0327) | 0.312±(0.0026) | 0.336±(0.0073) | 7.682 | 70.294 | 0.341 |
| | | Ours | 3.734(±0.0297) | 0.606±(0.0028) | 0.422±(0.0036) | 7.598 | 68.787 | 0.362 |
| | Canadian | GT | 3.842(±0.0026) | - | - | - | - | - |
| | | Prompt | 3.206(±0.0626) | 0.268±(0.0013) | 0.322±(0.0031) | 7.608 | 101.237 | 0.349 |
| | | Ours | 3.572(±0.0175) | 0.500±(0.0084) | 0.466±(0.0028) | 7.250 | 80.957 | 0.391 |
| | African | GT | 3.842(±0.0026) | - | - | - | - | - |
| | | Prompt | 3.206(±0.0109) | 0.302±(0.0033) | 0.406±(0.0023) | 8.039 | 81.730 | 0.251 |
| | | Ours | 3.518(±0.0053) | 0.556±(0.0019) | 0.496±(0.0077) | 7.335 | 74.216 | 0.309 |
| | Australian | GT | 3.842(±0.0026) | - | - | - | - | - |
| | | Prompt | 3.065(±0.0426) | 0.270±(0.0068) | 0.282±(0.0058) | 7.406 | 56.671 | 0.358 |
| | | Ours | 3.474(±0.0250) | 0.556±(0.0019) | 0.482±(0.0203) | 7.035 | 50.224 | 0.393 |
| | German | GT | 3.842(±0.0026) | - | - | - | - | - |
| | | Prompt | 2.766(±0.0058) | 0.272±(0.0063) | 0.452±(0.0157) | 7.828 | 61.770 | 0.303 |
| | | Ours | 3.466(±0.0143) | 0.502±(0.0054) | 0.442±(0.0271) | 7.412 | 56.454 | 0.366 |
| | Irish | GT | 3.842(±0.0026) | - | - | - | - | - |
| | | Prompt | 3.185(±0.0448) | 0.330±(0.0049) | 0.342±(0.0116) | 7.872 | 112.735 | 0.320 |
| | | Ours | 3.573(±0.0200) | 0.550±(0.0053) | 0.434±(0.0088) | 7.103 | 87.870 | 0.465 |
| | New Zealand | GT | 3.842(±0.0026) | - | - | - | - | - |
| | | Prompt | 3.283(±0.0429) | 0.234±(0.0073) | 0.302±(0.0129) | 7.111 | 84.131 | 0.261 |
| | | Ours | 3.505(±0.0142) | 0.610±(0.0055) | 0.396±(0.0054) | 6.729 | 75.663 | 0.311 |
| | Scottish | GT | 3.842(±0.0026) | - | - | - | - | - |
| | | Prompt | 3.248(±0.0556) | 0.216±(0.0028) | 0.444±(0.0091) | 7.313 | 87.023 | 0.363 |
| | | Ours | 3.286(±0.0226) | 0.530±(0.0070) | 0.432±(0.0119) | 7.433 | 81.028 | 0.386 |
| **Age** | <20 | GT | 3.872(±0.0068) | - | - | - | - | - |
| | | Prompt | 2.293(±0.0364) | 0.148±(0.0016) | 0.324±(0.0023) | 8.526 | 91.606 | 0.210 |
| | | Ours | 3.330(±0.0183) | 0.604±(0.0013) | 0.340±(0.0013) | 7.812 | 82.244 | 0.285 |
| | 20-29 | GT | 3.580(±0.0133) | - | - | - | - | - |
| | | Prompt | 3.125(±0.0149) | 0.270±(0.0055) | 0.308±(0.0020) | 7.773 | 80.275 | 0.250 |
| | | Ours | 3.403(±0.0019) | 0.580±(0.0024) | 0.508±(0.0017) | 7.393 | 65.687 | 0.264 |
| | 30-39 | GT | 3.500(±0.0063) | - | - | - | - | - |
| | | Prompt | 3.362(±0.0221) | 0.304±(0.0028) | 0.408±(0.0020) | 7.959 | 55.910 | 0.269 |
| | | Ours | 3.412(±0.0295) | 0.510±(0.0051) | 0.424±(0.0043) | 7.659 | 50.485 | 0.313 |
| | 40-49 | GT | 3.500(±0.0063) | - | - | - | - | - |
| | | Prompt | 2.772(±0.0202) | 0.258±(0.0029) | 0.332±(0.0018) | 7.778 | 62.638 | 0.260 |
| | | Ours | 2.880(±0.0195) | 0.494±(0.0041) | 0.428±(0.0066) | 7.635 | 59.207 | 0.283 |
| | 50-59 | GT | 3.500(±0.0063) | - | - | - | - | - |
| | | Prompt | 2.519(±0.0097) | 0.274±(0.0036) | 0.374±(0.0187) | 7.780 | 58.911 | 0.274 |
| | | Ours | 3.290(±0.0193) | 0.558±(0.0061) | 0.470±(0.0210) | 7.601 | 53.792 | 0.297 |
| | >60 | GT | 3.906(±0.0107) | - | - | - | - | - |
| | | Prompt | 2.338(±0.0268) | 0.244±(0.0016) | 0.280±(0.0005) | 7.575 | 57.913 | 0.317 |
| | | Ours | 3.273(±0.0201) | 0.582±(0.0027) | 0.504±(0.0082) | 7.126 | 52.973 | 0.364 |

Table 10: Single LoRA VoiceLDM table2

| Metrics | | | AMOS↑ | Win-Attr↑ | Win-Qual↑ | MCD↓ | F0-RMSE↓ | SSIM↑ |
|---|---|---|---|---|---|---|---|---|
| **Emotion** | Angry | GT | 3.872(±0.0068) | - | - | - | - | - |
| | | Prompt | 2.320±(0.0057) | 0.216±(0.0023) | 0.228±(0.0028) | 8.744 | 80.619 | 0.294 |
| | | Ours | 2.960±(0.0093) | 0.542±(0.0011) | 0.360±(0.0077) | 8.041 | 74.315 | 0.339 |
| | Contempt | GT | 3.580(±0.0133) | - | - | - | - | - |
| | | Prompt | 3.110±(0.0055) | 0.320±(0.0303) | 0.380±(0.0337) | 9.411 | 88.837 | 0.216 |
| | | Ours | 3.090±(0.0068) | 0.438±(0.0237) | 0.398±(0.0174) | 7.882 | 75.575 | 0.243 |
| | Disgusted | GT | 3.500(±0.0063) | - | - | - | - | - |
| | | Prompt | 2.460±(0.0730) | 0.250±(0.0009) | 0.378±(0.0095) | 8.250 | 86.964 | 0.241 |
| | | Ours | 2.780±(0.0220) | 0.496±(0.0071) | 0.388±(0.0193) | 8.320 | 77.363 | 0.263 |
| | Fear | GT | 3.500(±0.0063) | - | - | - | - | - |
| | | Prompt | 2.860±(0.0168) | 0.302±(0.0159) | 0.416±(0.0297) | 8.740 | 82.451 | 0.218 |
| | | Ours | 3.410±(0.0155) | 0.544±(0.0124) | 0.440±(0.0105) | 8.036 | 65.608 | 0.239 |
| | Happy | GT | 3.500(±0.0063) | - | - | - | - | - |
| | | Prompt | 2.370±(0.0395) | 0.264±(0.0017) | 0.360±(0.0122) | 7.917 | 98.318 | 0.246 |
| | | Ours | 3.170±(0.0545) | 0.574±(0.0031) | 0.464±(0.0102) | 7.383 | 72.511 | 0.317 |
| | Neutral | GT | 3.500(±0.0063) | - | - | - | - | - |
| | | Prompt | 3.630±(0.0370) | 0.298±(0.0029) | 0.400±(0.0115) | 6.690 | 59.095 | 0.286 |
| | | Ours | 3.760±(0.0230) | 0.486±(0.0072) | 0.400±(0.0053) | 6.770 | 65.419 | 0.282 |
| | Sad | GT | 3.500(±0.0063) | - | - | - | - | - |
| | | Prompt | 2.150±(0.0125) | 0.312±(0.0277) | 0.426±(0.0109) | 7.772 | 74.204 | 0.224 |
| | | Ours | 3.660±(0.0530) | 0.506±(0.0152) | 0.460±(0.0096) | 7.562 | 75.800 | 0.253 |
| | Surprised | GT | 3.906(±0.0107) | - | - | - | - | - |
| | | Prompt | 2.180±(0.0470) | 0.228±(0.0037) | 0.284±(0.0016) | 7.172 | 99.517 | 0.279 |
| | | Ours | 3.480±(0.0220) | 0.400±(0.0046) | 0.346±(0.0013) | 7.219 | 93.524 | 0.302 |
| **Gender** | Female | GT | 3.872(±0.0068) | - | - | - | - | - |
| | | Prompt | 5.000±(0.0000) | 0.370±(0.0067) | 0.400±(0.0055) | 7.509 | 95.107 | 0.156 |
| | | Ours | 5.000±(0.0000) | 0.514±(0.0052) | 0.408±(0.0042) | 7.465 | 88.820 | 0.196 |
| | Male | GT | 3.872(±0.0068) | - | - | - | - | - |
| | | Prompt | 4.900±(0.0500) | 0.316±(0.0027) | 0.436±(0.0042) | 7.984 | 66.466 | 0.183 |
| | | Ours | 5.000±(0.0000) | 0.570±(0.0009) | 0.414±(0.0018) | 6.708 | 50.520 | 0.287 |
| **Energy** | High | GT | 3.872(±0.0068) | - | - | - | - | - |
| | | Prompt | 3.530±(0.0070) | 0.288±(0.0067) | 0.364±(0.0046) | 9.110 | 75.222 | 0.244 |
| | | Ours | 3.760±(0.0092) | 0.476±(0.0158) | 0.426±(0.0211) | 8.674 | 71.325 | 0.297 |
| | Normal | GT | 3.872(±0.0068) | - | - | - | - | - |
| | | Prompt | 3.620±(0.0058) | 0.292±(0.0049) | 0.352±(0.0025) | 7.256 | 68.623 | 0.296 |
| | | Ours | 3.620±(0.0033) | 0.528±(0.0010) | 0.402±(0.0041) | 6.841 | 64.300 | 0.309 |
| | Low | GT | 3.872(±0.0068) | - | - | - | - | - |
| | | Prompt | 3.660±(0.0430) | 0.324±(0.0183) | 0.376±(0.0204) | 9.483 | 86.203 | 0.265 |
| | | Ours | 3.670±(0.0495) | 0.460±(0.0088) | 0.334±(0.0039) | 9.127 | 69.790 | 0.286 |
| **Pitch** | High | GT | 3.872(±0.0068) | - | - | - | - | - |
| | | Prompt | 3.731±(0.0136) | 0.270±(0.0015) | 0.408±(0.0029) | 7.392 | 80.564 | 0.140 |
| | | Ours | 3.822±(0.0007) | 0.520±(0.0008) | 0.422±(0.0086) | 6.657 | 73.766 | 0.195 |
| | Normal | GT | 3.872(±0.0068) | - | - | - | - | - |
| | | Prompt | 3.629±(0.0055) | 0.280±(0.0037) | 0.450±(0.0094) | 5.954 | 54.105 | 0.205 |
| | | Ours | 3.604±(0.0083) | 0.558±(0.0029) | 0.458±(0.0028) | 5.492 | 52.942 | 0.239 |
| | Low | GT | 3.872(±0.0068) | - | - | - | - | - |
| | | Prompt | 3.611±(0.0016) | 0.232±(0.0021) | 0.292±(0.0017) | 7.211 | 47.296 | 0.199 |
| | | Ours | 3.657±(0.0068) | 0.512±(0.0014) | 0.338±(0.0047) | 7.000 | 45.805 | 0.222 |
| **Overall** | | GT | 3.795±(0.0145) | - | - | - | - | - |
| | | Prompt | 3.154±(0.0267) | 0.274±(0.0059) | 0.364±(0.0084) | 7.873 | 76.325 | 0.262 |
| | | Ours | 3.538±(0.0173) | 0.534±(0.0054) | 0.428±(0.0083) | 7.441 | 68.277 | 0.304 |

Table 11: Dual LoRA Parler-tts table

| Metrics | | | AMOS↑ | Win-Attr↑ | Win-Qual↑ |
|---|---|---|---|---|---|
| **2 Attributes** | UK+20 | Prompt | 2.900(±0.0613) | 0.220(±0.0030) | 0.296(±0.0048) |
| | | Ours | 3.730(±0.0108) | 0.592(±0.0184) | 0.322(±0.0030) |
| | UK+60 | Prompt | 2.880(±0.0133) | 0.220(±0.0035) | 0.382(±0.0033) |
| | | Ours | 3.668(±0.0124) | 0.634(±0.0029) | 0.430(±0.0245) |
| | UK+Happy | Prompt | 2.848(±0.1334) | 0.162(±0.0008) | 0.322(±0.0071) |
| | | Ours | 3.620(±0.0066) | 0.636(±0.0012) | 0.378(±0.0028) |
| | UK+Sad | Prompt | 2.742(±0.0391) | 0.266(±0.0010) | 0.398(±0.0036) |
| | | Ours | 3.720(±0.0060) | 0.592(±0.0045) | 0.370(±0.0017) |
| | UK+Surprised | Prompt | 2.772(±0.0412) | 0.194(±0.0002) | 0.400(±0.0011) |
| | | Ours | 3.420(±0.0105) | 0.638(±0.0010) | 0.464(±0.0013) |
| | UK+teens | Prompt | 2.792(±0.0196) | 0.240(±0.0011) | 0.414(±0.0088) |
| | | Ours | 3.550(±0.0062) | 0.560(±0.0234) | 0.462(±0.0027) |
| | US+20 | Prompt | 3.320(±0.0408) | 0.212(±0.0008) | 0.294(±0.0222) |
| | | Ours | 3.650(±0.0025) | 0.622(±0.0016) | 0.354(±0.0156) |
| | US+60 | Prompt | 3.030(±0.0208) | 0.208(±0.0011) | 0.412(±0.0053) |
| | | Ours | 3.620(±0.0110) | 0.538(±0.0228) | 0.448(±0.0020) |
| | US+teens | Prompt | 3.150(±0.0375) | 0.204(±0.0006) | 0.382(±0.0158) |
| | | Ours | 3.648(±0.0037) | 0.536(±0.0307) | 0.448(±0.0133) |
| **Overall** | | Prompt | 2.937(±0.0452) | 0.214(±0.0013) | 0.367(±0.0080) |
| | | Ours | 3.625(±0.0077) | 0.594(±0.0118) | 0.408(±0.0074) |

Table 12: Dual LoRA VoiceLDM table

| Metrics | | | AMOS↑ | Win-Attr↑ | Win-Qual↑ |
|---|---|---|---|---|---|
| **2 Attributes** | UK+20 | Prompt | 2.780±(0.1145) | 0.258±(0.0030) | 0.258±(0.0053) |
| | | Ours | 3.570±(0.0208) | 0.454±(0.0016) | 0.358±(0.0025) |
| | UK+60 | Prompt | 2.660±(0.0168) | 0.234±(0.0070) | 0.368±(0.0091) |
| | | Ours | 3.740±(0.0668) | 0.426±(0.0094) | 0.416±(0.0055) |
| | UK+Happy | Prompt | 2.918±(0.1915) | 0.206±(0.0007) | 0.308±(0.0103) |
| | | Ours | 3.350±(0.0195) | 0.470±(0.0177) | 0.368±(0.0015) |
| | UK+Sad | Prompt | 2.820±(0.1407) | 0.284±(0.0018) | 0.380±(0.0049) |
| | | Ours | 3.608±(0.0494) | 0.456±(0.0031) | 0.376±(0.0010) |
| | UK+Surprised | Prompt | 2.542±(0.0788) | 0.204±(0.0010) | 0.376±(0.0028) |
| | | Ours | 3.730±(0.1286) | 0.430±(0.0183) | 0.438±(0.0010) |
| | UK+teens | Prompt | 3.010±(0.0330) | 0.292±(0.0033) | 0.394±(0.0088) |
| | | Ours | 3.624±(0.0261) | 0.432±(0.0020) | 0.472±(0.0067) |
| | US+20 | Prompt | 3.056±(0.0093) | 0.224±(0.0011) | 0.304±(0.0208) |
| | | Ours | 3.780±(0.0261) | 0.388±(0.0056) | 0.366±(0.0122) |
| | US+60 | Prompt | 2.760±(0.0318) | 0.250±(0.0045) | 0.410±(0.0022) |
| | | Ours | 3.255±(0.0146) | 0.416±(0.0032) | 0.428±(0.0032) |
| | US+teens | Prompt | 2.895±(0.0240) | 0.258±(0.0050) | 0.366±(0.0220) |
| | | Ours | 3.554±(0.0161) | 0.376±(0.0031) | 0.412±(0.0024) |
| **Overall** | | Prompt | 2.827±(0.0711) | 0.246±(0.0030) | 0.352±(0.0096) |
| | | Ours | 3.579±(0.0409) | 0.428±(0.0071) | 0.404±(0.0040) |

Table 13: Triple LoRA Parler-tts table

| Metrics | | | AMOS↑ | Win-Attr↑ | Win-Qual↑ |
|---|---|---|---|---|---|
| **3 Attributes** | High Energy+Happy+US | Prompt | 2.660(±0.0430) | 0.224(±0.0024) | 0.365(±0.0086) |
| | | Ours | 3.250(±0.0250) | 0.594(±0.0128) | 0.382(±0.0010) |
| | Low Pitch+Happy+UK | Prompt | 2.520(±0.0970) | 0.178(±0.0007) | 0.376(±0.0099) |
| | | Ours | 3.220(±0.0333) | 0.664(±0.0001) | 0.434(±0.0102) |
| | 60s+Happy+UK | Prompt | 2.650(±0.0052) | 0.236(±0.0007) | 0.278(±0.0037) |
| | | Ours | 3.180(±0.0283) | 0.652(±0.0081) | 0.348(±0.0085) |
| | Teens+Happy+US | Prompt | 2.870(±0.0108) | 0.236(±0.0009) | 0.302(±0.0016) |
| | | Ours | 3.354(±0.0081) | 0.594(±0.0195) | 0.404(±0.0011) |
| | Teens+Sad+UK | Prompt | 2.618(±0.1514) | 0.212(±0.0032) | 0.310(±0.0047) |
| | | Ours | 3.352(±0.0490) | 0.580(±0.0182) | 0.322(±0.0054) |
| | Teens+Sad+US | Prompt | 2.618(±0.0101) | 0.214(±0.0028) | 0.366(±0.0019) |
| | | Ours | 3.330(±0.0497) | 0.538(±0.0325) | 0.384(±0.0027) |
| **Overall** | | Prompt | 2.656(±0.0529) | 0.217(±0.0018) | 0.333(±0.0051) |
| | | Ours | 3.281(±0.0322) | 0.604(±0.0152) | 0.379(±0.0048) |

Table 14: Triple LoRA VoiceLDM table

| Metrics | | | AMOS↑ | Win-Attr↑ | Win-Qual↑ |
|---|---|---|---|---|---|
| **3 Attributes** | High Energy+Happy+US | Prompt | 2.722±(0.0324) | 0.294±(0.0301) | 0.345±(0.0077) |
| | | Ours | 3.115±(0.0374) | 0.474±(0.0199) | 0.354±(0.0033) |
| | Low Pitch+Happy+UK | Prompt | 2.350±(0.0413) | 0.204±(0.0015) | 0.366±(0.0055) |
| | | Ours | 2.881±(0.0117) | 0.462±(0.0057) | 0.422±(0.0010) |
| | 60s+Happy+UK | Prompt | 2.537±(0.0258) | 0.258±(0.0006) | 0.252±(0.0088) |
| | | Ours | 3.018±(0.1326) | 0.516±(0.0007) | 0.338±(0.0009) |
| | Teens+Happy+US | Prompt | 3.052±(0.0074) | 0.266±(0.0013) | 0.310±(0.0008) |
| | | Ours | 3.201±(0.0048) | 0.472±(0.0032) | 0.376±(0.0004) |
| | Teens+Sad+UK | Prompt | 2.653±(0.0755) | 0.256±(0.0013) | 0.304±(0.0074) |
| | | Ours | 3.008±(0.0089) | 0.444±(0.0067) | 0.384±(0.0030) |
| | Teens+Sad+US | Prompt | 2.805±(0.0133) | 0.238±(0.0013) | 0.342±(0.0020) |
| | | Ours | 3.412±(0.0344) | 0.396±(0.0025) | 0.364±(0.0008) |
| **Overall** | | Prompt | 2.687±(0.0326) | 0.253±(0.0060) | 0.320±(0.0054) |
| | | Ours | 3.106±(0.0383) | 0.461±(0.0065) | 0.373±(0.0016) |

Table 15: Quadra LoRA Parler-tts table

| Metrics | | | AMOS↑ | Win-Attr↑ | Win-Qual↑ |
|---|---|---|---|---|---|
| **4 Attributes** | Teens+Happy+Indian+High Pitch | Prompt | 2.200(±0.0288) | 0.276(±0.0018) | 0.374(±0.0012) |
| | | Ours | 2.424(±0.1037) | 0.476(±0.0058) | 0.356(±0.0016) |
| | Teens+Happy+UK+Low Pitch | Prompt | 2.228(±0.0257) | 0.322(±0.0058) | 0.288(±0.0027) |
| | | Ours | 2.252(±0.0253) | 0.414(±0.0047) | 0.362(±0.0018) |
| | 60s+Surprised+US+High Energy | Prompt | 2.160(±0.0730) | 0.290(±0.0016) | 0.350(±0.0021) |
| | | Ours | 2.898(±0.0585) | 0.300(±0.0013) | 0.368(±0.0014) |
| **Overall** | | Prompt | 2.196(±0.0425) | 0.296(±0.0030) | 0.337(±0.0020) |
| | | Ours | 2.525(±0.0625) | 0.397(±0.0039) | 0.362(±0.0016) |

Table 16: Quadra LoRA VoiceLDM table

| Metrics | | | AMOS↑ | Win-Attr↑ | Win-Qual↑ |
|---|---|---|---|---|---|
| **4 Attributes** | Teens+Happy+Indian+High Pitch | Prompt | 2.070(±0.0145) | 0.324(±0.0013) | 0.316(±0.0023) |
| | | Ours | 2.190(±0.0753) | 0.384(±0.0015) | 0.410(±0.0116) |
| | Teens+Happy+UK+Low Pitch | Prompt | 2.142(±0.0145) | 0.332(±0.0088) | 0.272(±0.0025) |
| | | Ours | 2.222(±0.0119) | 0.360(±0.0018) | 0.368(±0.0011) |
| | 60s+Surprised+US+High Energy | Prompt | 2.166(±0.0773) | 0.282(±0.0032) | 0.250(±0.0014) |
| | | Ours | 2.746(±0.0397) | 0.330(±0.0017) | 0.338(±0.0011) |
| **Overall** | | Prompt | 2.126(±0.0354) | 0.313(±0.0044) | 0.279(±0.0021) |
| | | Ours | 2.386(±0.0423) | 0.358(±0.0017) | 0.372(±0.0046) |

Table 17: Weighted experiences across different attribute pairs.

| Combination | AMOS-$a_1$ ↑ | AMOS-$a_2$ ↑ |
|---|---|---|
| $0.0 \cdot \mathbf{\Delta}_{a_1} + 1.0 \cdot \mathbf{\Delta}_{a_2}$ | 1.560(±0.1330) | 3.740(±0.0090) |
| $0.1 \cdot \mathbf{\Delta}_{a_1} + 0.9 \cdot \mathbf{\Delta}_{a_2}$ | 1.226(±0.0234) | 3.720(±0.0285) |
| $0.2 \cdot \mathbf{\Delta}_{a_1} + 0.8 \cdot \mathbf{\Delta}_{a_2}$ | 1.254(±0.0441) | 3.684(±0.0117) |
| $0.3 \cdot \mathbf{\Delta}_{a_1} + 0.7 \cdot \mathbf{\Delta}_{a_2}$ | 2.120(±0.2570) | 3.574(±0.0113) |
| $0.4 \cdot \mathbf{\Delta}_{a_1} + 0.6 \cdot \mathbf{\Delta}_{a_2}$ | 1.480(±0.0782) | 3.292(±0.1106) |
| $0.5 \cdot \mathbf{\Delta}_{a_1} + 0.5 \cdot \mathbf{\Delta}_{a_2}$ | 3.028(±0.0465) | 3.006(±0.0341) |
| $0.6 \cdot \mathbf{\Delta}_{a_1} + 0.4 \cdot \mathbf{\Delta}_{a_2}$ | 3.136(±0.0462) | 2.988(±0.0342) |
| $0.7 \cdot \mathbf{\Delta}_{a_1} + 0.3 \cdot \mathbf{\Delta}_{a_2}$ | 3.290(±0.0092) | 2.402(±0.0738) |
| $0.8 \cdot \mathbf{\Delta}_{a_1} + 0.2 \cdot \mathbf{\Delta}_{a_2}$ | 3.396(±0.0113) | 2.276(±0.0513) |
| $0.9 \cdot \mathbf{\Delta}_{a_1} + 0.1 \cdot \mathbf{\Delta}_{a_2}$ | 3.636(±0.0212) | 2.100(±0.0700) |
| $1.0 \cdot \mathbf{\Delta}_{a_1} + 0.0 \cdot \mathbf{\Delta}_{a_2}$ | 3.684(±0.0063) | 1.310(±0.1330) |

(a) Weighted experience: $\mathbf{\Delta}_{a_1}$: Indian, $\mathbf{\Delta}_{a_2}$: Happy

| Combination | AMOS-$a_1$ ↑ | AMOS-$a_2$ ↑ |
|---|---|---|
| $0.0 \cdot \mathbf{\Delta}_{a_1} + 1.0 \cdot \mathbf{\Delta}_{a_2}$ | 1.190(±0.0430) | 3.520(±0.0108) |
| $0.1 \cdot \mathbf{\Delta}_{a_1} + 0.9 \cdot \mathbf{\Delta}_{a_2}$ | 1.100(±0.0100) | 3.402(±0.0099) |
| $0.2 \cdot \mathbf{\Delta}_{a_1} + 0.8 \cdot \mathbf{\Delta}_{a_2}$ | 1.250(±0.0325) | 3.470(±0.0322) |
| $0.3 \cdot \mathbf{\Delta}_{a_1} + 0.7 \cdot \mathbf{\Delta}_{a_2}$ | 2.040(±0.0680) | 3.344(±0.0409) |
| $0.4 \cdot \mathbf{\Delta}_{a_1} + 0.6 \cdot \mathbf{\Delta}_{a_2}$ | 2.350(±0.0550) | 2.870(±0.0545) |
| $0.5 \cdot \mathbf{\Delta}_{a_1} + 0.5 \cdot \mathbf{\Delta}_{a_2}$ | 2.550(±0.1375) | 2.550(±0.0213) |
| $0.6 \cdot \mathbf{\Delta}_{a_1} + 0.4 \cdot \mathbf{\Delta}_{a_2}$ | 2.790(±0.0330) | 2.494(±0.0967) |
| $0.7 \cdot \mathbf{\Delta}_{a_1} + 0.3 \cdot \mathbf{\Delta}_{a_2}$ | 3.240(±0.0330) | 2.380(±0.0522) |
| $0.8 \cdot \mathbf{\Delta}_{a_1} + 0.2 \cdot \mathbf{\Delta}_{a_2}$ | 3.596(±0.0295) | 1.940(±0.0432) |
| $0.9 \cdot \mathbf{\Delta}_{a_1} + 0.1 \cdot \mathbf{\Delta}_{a_2}$ | 3.510(±0.0630) | 1.620(±0.0082) |
| $1.0 \cdot \mathbf{\Delta}_{a_1} + 0.0 \cdot \mathbf{\Delta}_{a_2}$ | 3.716(±0.0035) | 1.290(±0.0397) |

(b) Weighted experience: $\mathbf{\Delta}_{a_1}$: UK, $\mathbf{\Delta}_{a_2}$: Surprised

Table 18: Weighted experiences across different attribute pairs.

| Combination | AMOS-$a_1$ ↑ | AMOS-$a_2$ ↑ |
|---|---|---|
| $0.0 \cdot \mathbf{\Delta}_{a_1} + 1.0 \cdot \mathbf{\Delta}_{a_2}$ | 1.270(±0.0345) | 4.010(±0.0118) |
| $0.1 \cdot \mathbf{\Delta}_{a_1} + 0.9 \cdot \mathbf{\Delta}_{a_2}$ | 1.508(±0.0121) | 3.930(±0.0145) |
| $0.2 \cdot \mathbf{\Delta}_{a_1} + 0.8 \cdot \mathbf{\Delta}_{a_2}$ | 1.444(±0.0089) | 3.762(±0.0290) |
| $0.3 \cdot \mathbf{\Delta}_{a_1} + 0.7 \cdot \mathbf{\Delta}_{a_2}$ | 2.048(±0.0161) | 3.748(±0.0038) |
| $0.4 \cdot \mathbf{\Delta}_{a_1} + 0.6 \cdot \mathbf{\Delta}_{a_2}$ | 2.300(±0.0513) | 3.406(±0.0057) |
| $0.5 \cdot \mathbf{\Delta}_{a_1} + 0.5 \cdot \mathbf{\Delta}_{a_2}$ | 2.562(±0.0080) | 3.220(±0.0158) |
| $0.6 \cdot \mathbf{\Delta}_{a_1} + 0.4 \cdot \mathbf{\Delta}_{a_2}$ | 3.176(±0.0339) | 2.890(±0.0468) |
| $0.7 \cdot \mathbf{\Delta}_{a_1} + 0.3 \cdot \mathbf{\Delta}_{a_2}$ | 3.306(±0.0164) | 2.276(±0.0596) |
| $0.8 \cdot \mathbf{\Delta}_{a_1} + 0.2 \cdot \mathbf{\Delta}_{a_2}$ | 3.474(±0.0514) | 2.640(±0.0430) |
| $0.9 \cdot \mathbf{\Delta}_{a_1} + 0.1 \cdot \mathbf{\Delta}_{a_2}$ | 3.520(±0.0133) | 1.890(±0.0705) |
| $1.0 \cdot \mathbf{\Delta}_{a_1} + 0.0 \cdot \mathbf{\Delta}_{a_2}$ | 3.812(±0.0157) | 1.420(±0.0270) |

(a) Weighted experience: $\mathbf{\Delta}_{a_1}$: UK, $\mathbf{\Delta}_{a_2}$: >60

| Combination | AMOS-$a_1$ ↑ | AMOS-$a_2$ ↑ |
|---|---|---|
| $0.0 \cdot \mathbf{\Delta}_{a_1} + 1.0 \cdot \mathbf{\Delta}_{a_2}$ | 3.190(±0.0205) | 3.598(±0.0023) |
| $0.1 \cdot \mathbf{\Delta}_{a_1} + 0.9 \cdot \mathbf{\Delta}_{a_2}$ | 3.210(±0.0043) | 3.524(±0.0121) |
| $0.2 \cdot \mathbf{\Delta}_{a_1} + 0.8 \cdot \mathbf{\Delta}_{a_2}$ | 3.386(±0.0197) | 3.280(±0.0520) |
| $0.3 \cdot \mathbf{\Delta}_{a_1} + 0.7 \cdot \mathbf{\Delta}_{a_2}$ | 3.426(±0.0088) | 3.156(±0.0149) |
| $0.4 \cdot \mathbf{\Delta}_{a_1} + 0.6 \cdot \mathbf{\Delta}_{a_2}$ | 3.544(±0.0089) | 2.934(±0.0027) |
| $0.5 \cdot \mathbf{\Delta}_{a_1} + 0.5 \cdot \mathbf{\Delta}_{a_2}$ | 3.548(±0.0315) | 2.730(±0.0320) |
| $0.6 \cdot \mathbf{\Delta}_{a_1} + 0.4 \cdot \mathbf{\Delta}_{a_2}$ | 3.638(±0.0032) | 2.670(±0.0073) |
| $0.7 \cdot \mathbf{\Delta}_{a_1} + 0.3 \cdot \mathbf{\Delta}_{a_2}$ | 3.710(±0.0142) | 2.290(±0.0173) |
| $0.8 \cdot \mathbf{\Delta}_{a_1} + 0.2 \cdot \mathbf{\Delta}_{a_2}$ | 3.794(±0.0247) | 2.434(±0.0121) |
| $0.9 \cdot \mathbf{\Delta}_{a_1} + 0.1 \cdot \mathbf{\Delta}_{a_2}$ | 3.810(±0.0193) | 1.886(±0.1095) |
| $1.0 \cdot \mathbf{\Delta}_{a_1} + 0.0 \cdot \mathbf{\Delta}_{a_2}$ | 4.114(±0.0489) | 1.510(±0.0430) |

(b) Weighted experience: $\mathbf{\Delta}_{a_1}$: US, $\mathbf{\Delta}_{a_2}$: 20s

Table 19: $\boldsymbol{\Delta}_{a_1}$: US, $\boldsymbol{\Delta}_{a_2}$: Teens

| Subtraction | AMOS-$a_1$ ↑ | AMOS-$a_2$ ↑ | Win-Attr-$a_1$ ↑ | Win-Attr-$a_2$ ↑ | Win-Qual↑ |
|---|---|---|---|---|---|
| GT | 3.918(±0.0221) | 3.540(±0.0193) | - | - | - |
| Prompt | 3.290(±0.0055) | 2.790(±0.0143) | 0.314(±0.0011) | 0.198(±0.0047) | 0.364(±0.0082) |
| $\boldsymbol{\Delta}_{a_1+a_2}$ | 3.484(±0.0165) | 3.476(±0.0156) | 0.336(±0.0016) | 0.320(±0.0048) | 0.394(±0.0027) |
| $\boldsymbol{\Delta}_{a_1}$ | 3.702(±0.0250) | 2.780(±0.0420) | 0.410±(0.0012) | 0.142(±0.0024) | 0.356(±0.0059) |
| $\boldsymbol{\Delta}_{a_2}$ | 2.960(±0.0655) | 3.420(±0.0308) | 0.274(±0.0042) | 0.292(±0.0020) | 0.384(±0.0033) |
| $\boldsymbol{\Delta}_{a_1+a_2} - \boldsymbol{\Delta}_{a_2}$ | 3.624(±0.0015) | 2.608(±0.0222) | 0.372(±0.0024) | 0.216(±0.0024) | 0.294(±0.0023) |
| $\boldsymbol{\Delta}_{a_1+a_2} - \boldsymbol{\Delta}_{a_1}$ | 3.090(±0.0780) | 3.350(±0.0500) | 0.292(±0.0036) | 0.338(±0.0021) | 0.308(±0.0038) |

Table 20: $\boldsymbol{\Delta}_{a_1}$: US, $\boldsymbol{\Delta}_{a_2}$: Over 60

| Subtraction | AMOS-$a_1$ ↑ | AMOS-$a_2$ ↑ | Win-Attr-$a_1$ ↑ | Win-Attr-$a_2$ ↑ | Win-Qual↑ |
|---|---|---|---|---|---|
| GT | 3.828(±0.0152) | 4.020(±0.0258) | - | - | - |
| Prompt | 3.270(±0.0158) | 2.596(±0.0265) | 0.290(±0.0044) | 0.120(±0.0024) | 0.372(±0.0044) |
| $\boldsymbol{\Delta}_{a_1+a_2}$ | 3.538(±0.0228) | 3.402(±0.0315) | 0.296(±0.0029) | 0.358(±0.0087) | 0.352(±0.0016) |
| $\boldsymbol{\Delta}_{a_1}$ | 3.798(±0.0200) | 2.332(±0.0149) | 0.426(±0.0017) | 0.234(±0.0021) | 0.356(±0.0019) |
| $\boldsymbol{\Delta}_{a_2}$ | 2.894(±0.0357) | 3.528(±0.0127) | 0.220(±0.0033) | 0.488(±0.0038) | 0.348(±0.0007) |
| $\boldsymbol{\Delta}_{a_1+a_2} - \boldsymbol{\Delta}_{a_2}$ | 3.702(±0.0088) | 2.240(±0.0518) | 0.388(±0.0010) | 0.244(±0.0026) | 0.330(±0.0017) |
| $\boldsymbol{\Delta}_{a_1+a_2} - \boldsymbol{\Delta}_{a_1}$ | 3.270(±0.0395) | 3.690(±0.0455) | 0.198(±0.0044) | 0.372(±0.0051) | 0.378(±0.0047) |

Table 21: $\boldsymbol{\Delta}_{a_1}$: 20, $\boldsymbol{\Delta}_{a_2}$: US

| Subtraction | AMOS-$a_1$ ↑ | AMOS-$a_2$ ↑ | Win-Attr-$a_1$ ↑ | Win-Attr-$a_2$ ↑ | Win-Qual↑ |
|---|---|---|---|---|---|
| GT | 3.816(±0.0160) | 3.838(±0.0315) | - | - | - |
| Prompt | 2.990(±0.0193) | 3.268(±0.0604) | 0.206(±0.0048) | 0.242(±0.0026) | 0.378(±0.0032) |
| $\boldsymbol{\Delta}_{a_1+a_2}$ | 3.614(±0.0092) | 3.544(±0.0424) | 0.356(±0.0047) | 0.354(±0.0039) | 0.362(±0.0027) |
| $\boldsymbol{\Delta}_{a_1}$ | 2.558(±0.0138) | 3.282(±0.4114) | 0.172(±0.0035) | 0.378(±0.0024) | 0.374(±0.0032) |
| $\boldsymbol{\Delta}_{a_2}$ | 3.454(±0.0115) | 3.260(±0.0843) | 0.316(±0.0023) | 0.316(±0.0019) | 0.348(±0.0021) |
| $\boldsymbol{\Delta}_{a_1+a_2} - \boldsymbol{\Delta}_{a_2}$ | 2.340(±0.0543) | 3.150(±0.5138) | 0.196(±0.0024) | 0.358(±0.0022) | 0.332(±0.0015) |
| $\boldsymbol{\Delta}_{a_1+a_2} - \boldsymbol{\Delta}_{a_1}$ | 3.410(±0.0130) | 3.320(±0.0220) | 0.340(±0.0013) | 0.322(±0.0018) | 0.346(±0.0020) |

Table 22: $\boldsymbol{\Delta}_{a_1}$: UK, $\boldsymbol{\Delta}_{a_2}$: 30

| Subtraction | AMOS-$a_1$ ↑ | AMOS-$a_2$ ↑ | Win-Attr-$a_1$ ↑ | Win-Attr-$a_2$ ↑ | Win-Qual↑ |
|---|---|---|---|---|---|
| GT | 4.000(±0.0188) | 3.854(±0.0060) | - | - | - |
| Prompt | 3.528(±0.0053) | 3.610(±0.0109) | 0.292(±0.0041) | 0.292(±0.0010) | 0.444(±0.0051) |
| $\boldsymbol{\Delta}_{a_1+a_2}$ | 3.824(±0.0061) | 3.768(±0.0100) | 0.376(±0.0030) | 0.334(±0.0024) | 0.420(±0.0005) |
| $\boldsymbol{\Delta}_{a_1}$ | 3.740(±0.0293) | 3.560(±0.0517) | 0.342(±0.0058) | 0.316(±0.0036) | 0.368(±0.0019) |
| $\boldsymbol{\Delta}_{a_2}$ | 1.960(±0.0692) | 3.703(±0.0131) | 0.152(±0.0019) | 0.402(±0.0023) | 0.384(±0.0022) |
| $\boldsymbol{\Delta}_{a_1+a_2} - \boldsymbol{\Delta}_{a_2}$ | 3.602(±0.0215) | 3.556(±0.0319) | 0.304(±0.0020) | 0.294(±0.0016) | 0.384(±0.0018) |
| $\boldsymbol{\Delta}_{a_1+a_2} - \boldsymbol{\Delta}_{a_1}$ | 2.166(±0.0603) | 3.614(±0.0495) | 0.204(±0.0026) | 0.380(±0.0029) | 0.356(±0.0047) |

Table 23: Overall of subtract experience

| Subtraction | AMOS-$a_1$ ↑ | AMOS-$a_2$ ↑ | Win-Attr-$a_1$ ↑ | Win-Attr-$a_2$ ↑ | Win-Qual↑ |
|---|---|---|---|---|---|
| GT | 3.891(±0.0180) | 3.813(±0.0206) | - | - | - |
| Prompt | 3.270(±0.0114) | 3.066(±0.0280) | 0.276(±0.0036) | 0.213(±0.0026) | 0.390(±0.0052) |
| $\boldsymbol{\Delta}_{a_1+a_2}$ | 3.615(±0.0137) | 3.548(±0.0249) | 0.341(±0.0031) | 0.342(±0.0050) | 0.382(±0.0019) |
| $\boldsymbol{\Delta}_{a_1}$ | 3.450(±0.0220) | 2.989(±0.1300) | 0.338(±0.0030) | 0.268(±0.0026) | 0.364(±0.0032) |
| $\boldsymbol{\Delta}_{a_2}$ | 2.817(±0.0455) | 3.478(±0.0352) | 0.241(±0.0029) | 0.375(±0.0025) | 0.366(±0.0021) |
| $\boldsymbol{\Delta}_{a_1+a_2} - \boldsymbol{\Delta}_{a_2}$ | 3.317(±0.0215) | 2.889(±0.1549) | 0.315(±0.0019) | 0.278(±0.0022) | 0.335(±0.0018) |
| $\boldsymbol{\Delta}_{a_1+a_2} - \boldsymbol{\Delta}_{a_1}$ | 2.984(±0.0477) | 3.494(±0.0417) | 0.259(±0.0030) | 0.353(±0.0030) | 0.347(±0.0038) |

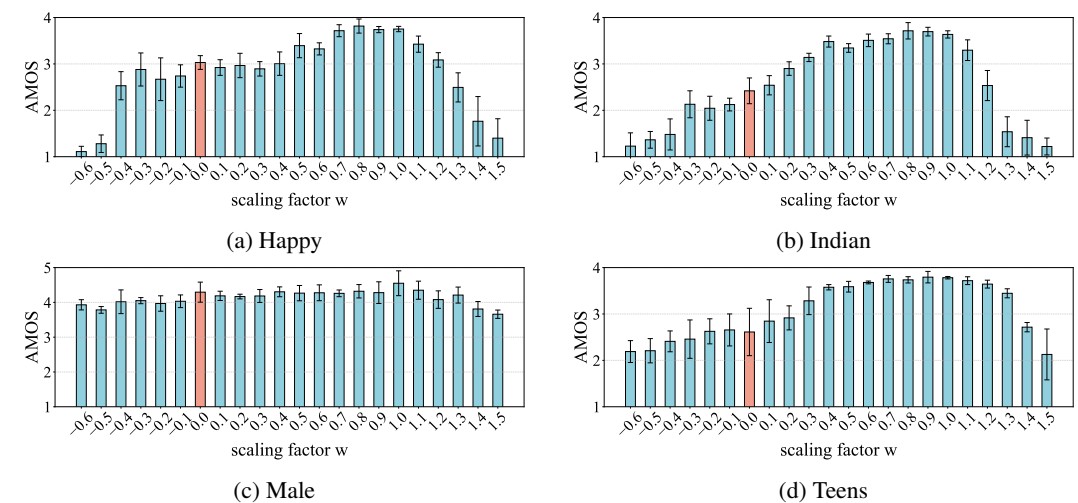

(a) Happy

(b) Indian

(c) Male

(d) Teens

Figure 5: MOS Results of the Scaling Experiment across different conditions.

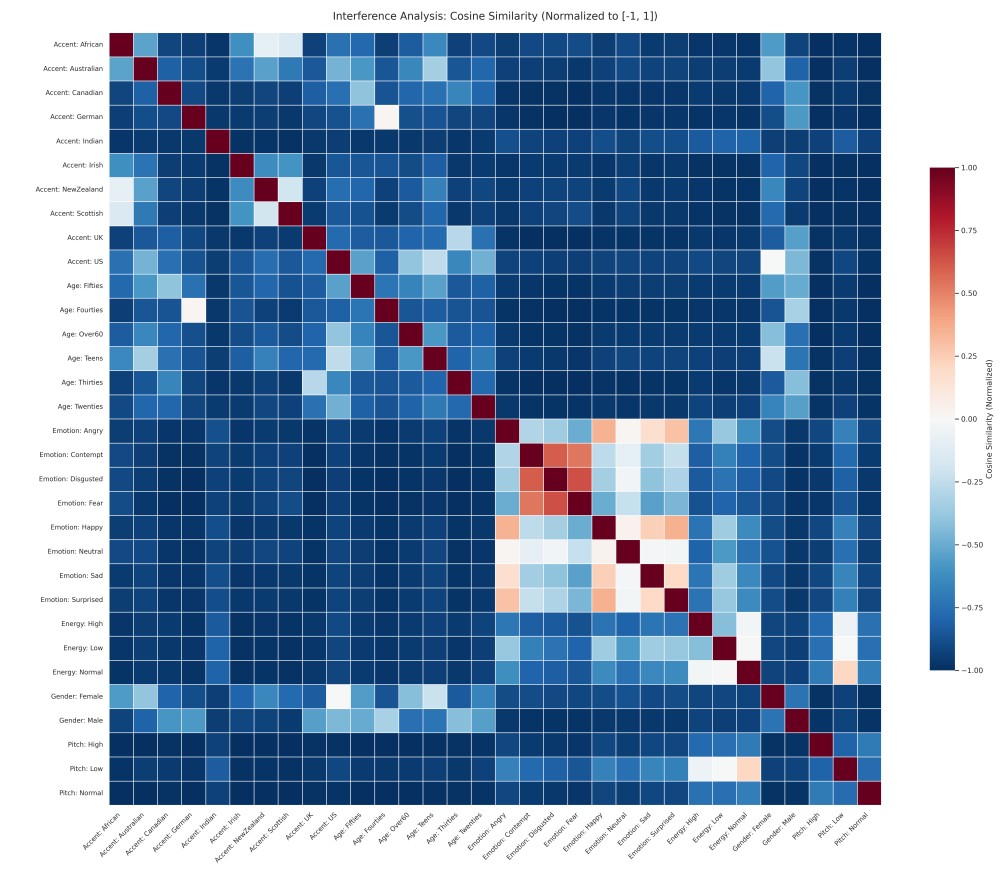

Figure 6: Heatmap of Model Performance Across All Attribute.

## H: COSINE SIMILARITY BETWEEN ALL ADAPTER

## I: VISUALIZATION OF AMOS SCORES FROM TABLE 3, SHOWING THE IMPACT OF VARYING WEIGHTS BETWEEN TWO LORA MODULES

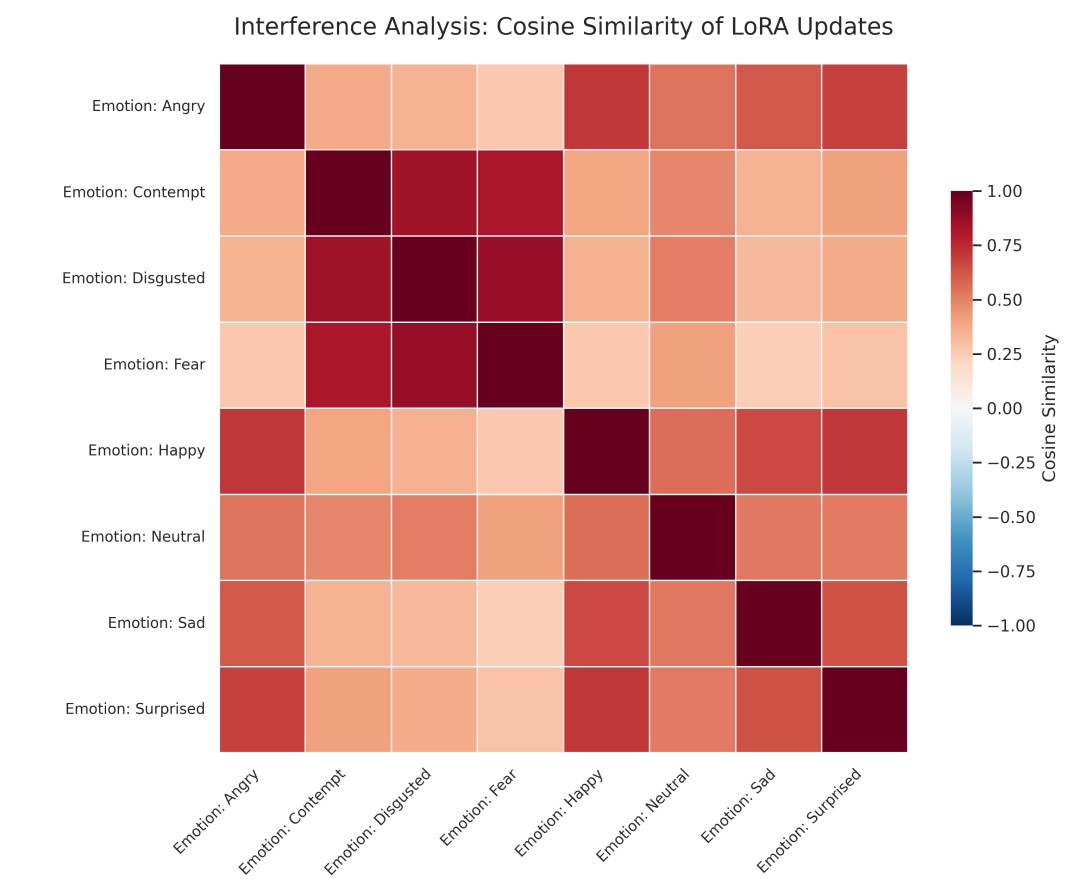

Figure 7: Heatmap visualization of all experimental results.

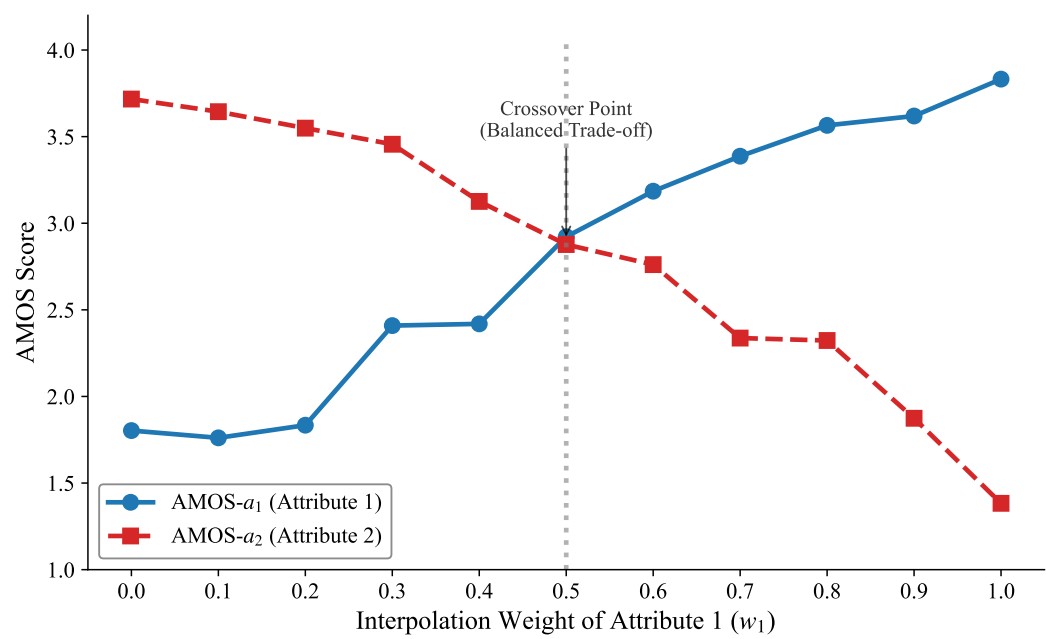

Figure 8: Visualization of AMOS Scores from Table 3, showing the impact of varying weights between two LoRA modules.

## J: FAILURE MODES ANALYSIS

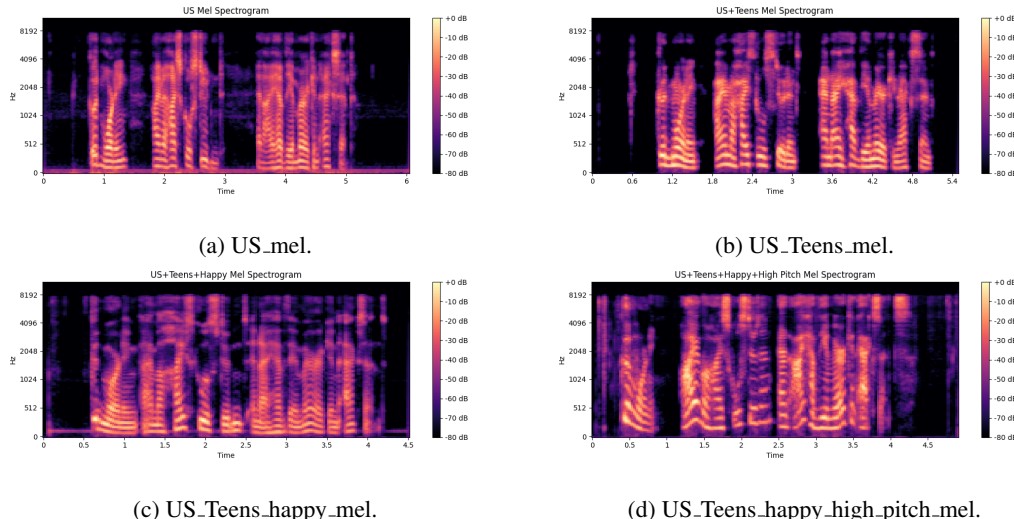

(a) US_mel.

(b) US_Teens_mel.

(c) US_Teens_happy_mel.

(d) US_Teens_happy_high_pitch_mel.

Figure 9: Visualization of Synthesis Failure Modes. The increasing blur and noise in the Mel spectrograms demonstrate the model's difficulty in coordinating multiple attributes, leading to a collapse in output fidelity.

As evidenced by the Mel spectrograms in Figure 9, the model's performance deteriorates with the number of attribute constraints. This decline manifests as a progressive blurring of the spectral structure, where distinct harmonic components begin to smear and unresolved noise becomes more prominent.

## K. STATISTICAL RELIABILITY OF EVALUATION METRICS

To ensure that our subjective metrics (AMOS) and LLM-based proxy metrics (Win-Attr/Win-Qual) are statistically robust and not driven by noise or evaluator bias, we conducted rigorous ground-truth validation studies.

**AMOS Inter-Rater Reliability** We clarify our evaluation protocol: the reported results are based on a Many-to-One MOS protocol. For every condition, 16 distinct synthesized clips were evaluated by 10 randomly assigned experts from a pool of 30, resulting in 160 independent observations per score. We measured the agreement among these 30 expert annotators using Krippendorff's Alpha (ordinal). As shown in Table 24, the Alpha score of 0.8362 indicates high inter-rater consistency.

**Validation of GPT-4o-Audio-Preview** To validate GPT-4o-Audio-Preview as an evaluation proxy, we conducted a Ground-Truth Alignment Study. We sampled 100 audio pairs and compared the LLM's decisions against a consensus established by the majority vote of 30 human experts. We measured agreement using Fleiss' Kappa. The scores of 0.8135 (Win-Attr) and 0.8417 (Win-Qual) fall within the range of "substantial agreement," confirming that the LLM serves as a reliable proxy for human perception in this task.

Table 24: Statistical Reliability Analysis of Evaluation Metrics.

| Metric | Validation Setup | Statistical Measure | Score |
|---|---|---|---|
| AMOS | 100 samples rated by 30 experts (1-5 scale) | Krippendorff's Alpha ($\alpha$) | 0.8362 |
| Win-Attr | 100 pairs (Human Majority vs. GPT-4o) | Fleiss' Kappa ($\kappa$) | 0.8135 |
| Win-Qual | 100 pairs (Human Majority vs. GPT-4o) | Fleiss' Kappa ($\kappa$) | 0.8417 |

## L. COMPARISON WITH REFERENCE-BASED SOTA MODELS

We compared TTS-Hub against state-of-the-art reference-based models, CosyVoice 2 and Style-Speech, to demonstrate the advantages of our modular approach. We evaluated two settings:

- **Matched Reference:** The baseline models are provided with a reference audio that explicitly contains the target attribute.
- **Blank Reference + Prompt:** The baseline models must generate the target attribute based solely on a style tag/prompt, simulating real-world scenarios where an ideal reference is unavailable.

TTS-Hub operates without any reference audio. Table 25 shows that while reference-based models perform well with matched references, their controllability degrades significantly in the Blank Reference setting. TTS-Hub matches the performance of reference-based models without requiring any reference audio, highlighting its superior flexibility.

Table 25: Comparison with SOTA Reference-Based Models. TTS-Hub achieves comparable performance to matched-reference baselines without requiring auxiliary audio inputs.

| Model | Condition | WER ↓ | AMOS ↑ | Win-Attr ↑ | Win-Qual ↑ |
|---|---|---|---|---|---|
| CosyVoice 2 | Matched Reference | 4.394 | 4.048 | 0.652 | 0.715 |
| CosyVoice 2 | Blank Ref + Prompt | 4.322 | 3.314 | 0.387 | 0.464 |
| StyleSpeech | Matched Reference | 4.713 | 3.623 | 0.609 | 0.651 |
| StyleSpeech | Blank Ref + Prompt | 4.604 | 3.207 | 0.362 | 0.389 |
| **TTS-Hub (Ours)** | **No Reference Needed** | 4.622 | 3.661 | 0.600 | 0.409 |

## M. EXTENDED ABLATION STUDIES

### LAYER-WISE LoRA INSERTION ANALYSIS

To identify the optimal insertion strategy, we partitioned the Parler-TTS decoder layers into Bottom (0–8), Mid (9–16), and Top (17–24) groups. As shown in Table 26, different attributes favor different components. High-level characteristics like *Age* are best modeled by Bottom layers, while linguistic features like *Accent* benefit most from FFN modification. However, the Full LoRA configuration provides the most robust performance across attributes.

Table 26: Layer-wise Ablation Study on Parler-TTS.

| Attribute | Configuration | AMOS ↑ | Win-Attr ↑ | Win-Qual ↑ | MCD ↓ | F0-RMSE ↓ | SSIM ↑ |
|---|---|---|---|---|---|---|---|
| Age (20s) | Baseline (Full LoRA) | 3.390 | 0.658 | 0.438 | 7.389 | 73.073 | 0.338 |
|  | Layers 0–8 (Bottom) | **3.582** | **0.712** | **0.489** | **7.056** | **68.442** | **0.355** |
|  | Layers 9–16 (Mid) | 3.356 | 0.642 | 0.425 | 7.421 | 74.115 | 0.334 |
|  | Layers 17–24 (Top) | 3.112 | 0.485 | 0.356 | 7.895 | 82.336 | 0.312 |
|  | FFN Only | 2.856 | 0.324 | 0.245 | 8.442 | 89.112 | 0.298 |
|  | Cross-Attention Only | 3.375 | 0.649 | 0.431 | 7.405 | 73.885 | 0.336 |
| Accent (US) | Baseline (Full LoRA) | 3.736 | 0.664 | 0.446 | 7.282 | 51.677 | 0.362 |
|  | Layers 0–8 (Bottom) | 3.412 | 0.512 | 0.388 | 7.654 | 55.123 | 0.335 |
|  | Layers 9–16 (Mid) | 3.455 | 0.534 | 0.392 | 7.598 | 54.882 | 0.341 |
|  | Layers 17–24 (Top) | 3.021 | 0.356 | 0.289 | 8.102 | 62.441 | 0.310 |
|  | FFN Only | **3.752** | **0.678** | **0.458** | **7.215** | **50.992** | **0.366** |
|  | Cross-Attention Only | 3.356 | 0.498 | 0.365 | 7.712 | 56.012 | 0.330 |

### CONFLICT AND INTERFERENCE ANALYSIS

We investigated the geometry of the LoRA parameter space by analyzing conflicting attributes. **Interference:** Table 27 demonstrates that adding contradictory attributes (e.g., High Pitch + Low

Pitch) results in destructive interference rather than a neutral output, confirming that LoRA vectors are not perfectly orthogonal. **Inverse Asymmetry:** Table 28 shows that subtracting an attribute is not equivalent to adding its opposite (i.e., $-\Delta_{\text{High}} \neq +\Delta_{\text{Low}}$), indicating that attributes affect different feature subspaces.

Table 27: Analysis of Conflicting Attributes (Destructive Interference).

| Combination Strategy | Target | AMOS ↑ | Win-Qual ↑ | MCD ↓ | F0-RMSE ↓ | SSIM ↑ |
|---|---|---|---|---|---|---|
| Base (No Adapter) | Neutral | 3.428 | — | 7.091 | 66.287 | 0.304 |
| $\Delta_{\text{High Pitch}} + \Delta_{\text{Low Pitch}}$ | Neutral | 2.811 | 0.441 | 8.672 | 89.314 | 0.132 |
| $\Delta_{\text{Age 20}} + \Delta_{\text{Age 60}}$ | Neutral | 2.739 | 0.512 | 9.349 | 80.737 | 0.155 |

Table 28: Subtraction vs. Addition of Opposites (Inverse Asymmetry).

| Operation | Target | AMOS ↑ | Win-Attr ↑ | Win-Qual ↑ | MCD ↓ | F0-RMSE ↓ | SSIM ↑ |
|---|---|---|---|---|---|---|---|
| $+\Delta_{\text{Low Pitch}}$ | Low Pitch | 3.657 | 0.512 | 0.338 | 7.000 | 45.805 | 0.222 |
| $-\Delta_{\text{High Pitch}}$ | Low Pitch | 2.395 | 0.410 | 0.375 | 9.191 | 78.834 | 0.121 |
| $+\Delta_{\text{Age 60}}$ | Elder | 3.273 | 0.582 | 0.504 | 7.126 | 52.973 | 0.364 |
| $-\Delta_{\text{Age 20}}$ | Elder | 1.972 | 0.363 | 0.438 | 8.773 | 90.194 | 0.136 |

## N. EFFICIENCY AND SCALABILITY ANALYSIS

### RUNTIME AND MEMORY OVERHEAD

We quantified the computational overhead of composing multiple LoRAs. As shown in Table 29, TTS-Hub exhibits negligible latency overhead (1.01x scaling) regardless of the number of adapters, as composition occurs in the weight space before the forward pass. Peak VRAM usage remains stable.

Table 29: Runtime and Memory Overhead on Parler-TTS.

| Config | Avg Latency (s) | Scale | Peak VRAM (MB) | Storage (MB) |
|---|---|---|---|---|
| **Base** | **7.10** | **1.00x** | **5827.27** | **-** |
| +1 LoRA | 7.16 | 1.01x | 5836.11 | +39.57 |
| +3 LoRA | 7.14 | 1.01x | 5831.92 | +118.70 |
| +6 LoRA | 7.15 | 1.01x | 5830.47 | +240.76 |

### FAIRNESS AND DATA IMBALANCE

We analyzed performance across high-resource and low-resource attributes (Table 30). Performance degrades on low-resource combinations, confirming that fairness limitations stem primarily from data scarcity rather than the compositional mechanism.

## O. ADVANCED FUSION STRATEGIES

To mitigate interference in complex combinations, we explored a **Mini-MoE Router**, a learnable mechanism that predicts optimal mixing weights. The router successfully separates conflicting attributes (e.g., assigning 1.0 to UK and 0.0 to US) and suppresses redundant experts (assigning 0.0 to "Sad" for non-emotional prompts). As shown in Table 31, the Mini-MoE strategy yields measurable gains over naive arithmetic addition, offering a high-quality alternative for scenarios where training overhead is acceptable.

Table 30: Performance comparison between High-Resource and Low-Resource attribute combinations.

| Attribute Combination | AMOS ↑ | Win-Attr ↑ | Win-Qual ↑ |
|---|---|---|---|
| *High-Res:* US Accent + Age 20-30 | 3.730 | 0.592 | 0.322 |
| *High-Res:* Indian Accent + Age 30-40 | 3.685 | 0.553 | 0.306 |
| *Low-Res:* African Accent + Age 20-30 | 3.237 | 0.359 | 0.228 |
| *Low-Res:* German Accent + Teens | 3.464 | 0.461 | 0.275 |

Table 31: Comparison of Naive Arithmetic Addition vs. Learned Mini-MoE Fusion.

| Fusion Strategy | AMOS ↑ | Win-Attr ↑ | Win-Qual ↑ |
|---|---|---|---|
| Naive Addition (Baseline) | 3.650 | 0.622 | 0.354 |
| Learned Mini-MoE (Ours) | **3.679** | **0.700** | **0.371** |
| *Improvement* | *+0.029* | *+0.078* | *+0.017* |

## P. METHODOLOGICAL COMPARISON

Table 32 contextualizes TTS-Hub within the broader landscape of LoRA-based adaptations. Unlike Dynamic Composition approaches (focused on cross-task transfer) or holistic TTS adaptations (which model style as an indivisible unit), TTS-Hub enables atomic decomposition and deterministic arithmetic control.

Table 32: Methodological Comparison of LoRA-based Approaches.

| Category | Works | Methodology | Insight for TTS-Hub |
|---|---|---|---|
| Dynamic Composition | LoRAHub, MoLE | Gradient-free search/routing | Motivates deterministic arithmetic for transparency. |
| LoRA for TTS (Holistic) | EELE, LoRP-TTS | Holistic style modeling | Motivates attribute decomposition for fine control. |
| **TTS-Hub (Ours)** | **TTS-Hub** | **Atomic LoRAs + Arithmetic** | **Enables zero-shot, disentangled attribute creation.** |

