# OpenReview forum: "TTS-Hub: Leveraging Modular LoRAs and  Arithmetic Composition for Controllable Text-to-Speech"
_ICLR.cc/2026/Conference — Submitted to ICLR 2026_

### Official Review · Reviewer_fo7r · 2025-10-23

**Soundness:** 3
**Presentation:** 3
**Contribution:** 3
**Rating:** 6
**Confidence:** 5

**Summary:**

Building on existing TTS backbones, the paper develops an attribute-specific Data Hub and fine-tunes LoRA modules for each characteristic, resulting in a LoRA Hub where every adapter is dedicated to controlling a specific speech attribute. By combining selected adapters through basic arithmetic operations, the system offers intuitive multi-attribute control in TTS. The approach is direct and uncomplicated. Experimental results indicate that individual adapters effectively manage their assigned attributes, and certain arithmetic combinations allow users to intuitively blend, reduce, or amplify these attributes, to some extent.

**Strengths:**

The method provides interpretable and versatile control over various predefined speech characteristics—such as accent, age, emotion, and gender—and can be easily extended by adding new adapters to enhance expressive potential.

The framework is simple and readily applicable to different TTS backbones.

**Weaknesses:**

Relying on predefined attributes may limit the breadth of semantic representation. Additionally, interactions between certain attributes might be non-linear or even conflicting, which means straightforward arithmetic composition may not always produce satisfactory outcomes.

It still requires a sizeable, attribute-annotated dataset encompassing all specified attributes to train the LoRA modules.

**Questions:**

Training data and attribute annotations are essential for controllable TTS. The baseline, such as Parler-TTS, used different data and fewer attributes, reducing expressiveness and potentially affecting comparisons between text prompts and LoRA-hub. The quality of synthesis is also dependent on backbone TTS. More detailed explanations of this issue are recommended.

When utilizing the GPT-4o-Audio API for speech evaluation, is there ablation study to measure its effectiveness comparing with human annotation?

---

> ### Author Response · Authors · 2025-11-21
> **Replies to Reviewer fo7r for Weakness 1**
>
> ## **Weaknesses1: Limits of Attribute Semantics and Inter-Attribute Interference**
>
> We thank the reviewer for this insightful observation. You have accurately identified two critical aspects of our framework: (1) the inherent trade-off between the semantic breadth of natural language and the precision of predefined attributes, and (2) the geometric complexity of the attribute space, where interactions are not always strictly linear.
>
> Below, we address these points through both our theoretical formulation and new experimental analysis.
>
> 1. Semantic Breadth vs. Disentangled Precision (The "Data Hub" Hypothesis)
>
> We acknowledge that relying on predefined attributes limits the infinite vocabulary available in open-ended prompting. However, this is a deliberate design choice to address the "cross-modal semantic gap" discussed in Section 1.
>
> - Hypothesis: We conceptualize each real-world audio sample as a high-dimensional information point. While no single voice exists in isolation (e.g., a "female" voice also possesses specific pitch, loudness, and emotion), we assume that by aggregating a massive number of samples sharing a specific label (e.g., "High Pitch") within our Data Hub, the training process isolates the principal direction of that attribute within the parameter space.
> - The Trade-off: By averaging out uncorrelated variations (noise, content, background), the LoRA module learns a robust, disentangled representation of the target attribute. We trade the "breadth" of vague descriptions for the "depth" of precise, reproducible control—a capability often lacking in purely prompt-based systems.
>
> 2. Non-Linearity and Conflicting Interactions
>
> You correctly point out that the manifold of speech attributes is not strictly linear. To rigorously define the boundaries of our arithmetic composition, we conducted a new "Conflict & Inverse Analysis" to investigate these interactions.
>
> A. Conflicting Attributes (Destructive Interference)
> We tested the addition of physiologically contradictory attributes, such as $\Delta _ {High\\_Pitch} + \Delta _ {Low\\_Pitch}$.
>
> - Expectation vs. Reality: Linearity implies these should cancel out to a "Neutral" pitch. However, as shown in the Table below, the result is not a perfect neutral voice but rather a degraded output with artifacts.
> - Analysis: The model struggles to resolve contradictory physiological instructions (e.g., the vocal tension required for high pitch vs. the relaxation required for low pitch). This confirms that while LoRA vectors are composable, they are not perfectly orthogonal; direct conflicts cause destructive interference.
>
> | Combination Strategy | Attribute Target | AMOS $\uparrow$ | Win-Qual $\uparrow$ | MCD $\downarrow$ | F0-RMSE $\downarrow$ | SSIM $\uparrow$ |
> | --- | --- | --- | --- | --- | --- | --- |
> | Base (No Adapter) | Neutral | 3.428 | — | 7.091 | 66.287 | 0.304 |
> | $\Delta _ {High\\_Pitch} + \Delta _ {Low\\_Pitch}$ | Conflict | 2.811 | 0.441 | 8.672 | 89.314 | 0.132 |
> | $\Delta _ {Age\\_20} + \Delta _ {Age\\_60}$ | Conflict | 2.739 | 0.512 | 9.349 | 80.737 | 0.155 |
>
> B. Inverse Asymmetry (Subtraction $\neq$ Addition of Opposite)
> We investigated whether subtracting an attribute is equivalent to adding its semantic opposite (i.e., does $Base - \Delta _ {High} \approx Base + \Delta _ {Low}$?).
>
> - Observation: As shown in the Table below, subtracting "High Pitch" lowers the F0 but results in a "flat" voice, lacking the resonance and harmonic structure characteristic of a true "Low Pitch" (which is obtained via $+\Delta _ {Low}$).
> - Conclusion: This demonstrates that $Base - \Delta _ {a}$ is not mathematically equivalent to $Base + \Delta _ {b}$. Subtraction can attenuate a feature (as demonstrated in Section 5.3), but it cannot "create" the complex acoustic features required for the opposite attribute.
>
> | Operation | Semantic Target | AMOS $\uparrow$ | Win-Attr $\uparrow$ | Win-Qual $\uparrow$ | MCD $\downarrow$ | F0-RMSE $\downarrow$ | SSIM $\uparrow$ |
> | --- | --- | --- | --- | --- | --- | --- | --- |
> | $+\Delta _ {Low\\_Pitch}$ | Low Pitch | 3.657 | 0.512 | 0.338 | 7.000 | 45.805 | 0.222 |
> | $-\Delta _ {High\\_Pitch}$ | Low Pitch | 2.395 | 0.410 | 0.375 | 9.191 | 78.834 | 0.121 |
> | $+\Delta _ {Age\\_60}$ | Elder | 3.273 | 0.582 | 0.504 | 7.126 | 52.973 | 0.364 |
> | $-\Delta _ {Age\\_20}$ | Elder | 1.972 | 0.363 | 0.438 | 8.773 | 90.194 | 0.136 |

---

> ### Author Response · Authors · 2025-11-21
> **Replies to Reviewer fo7r for Weakness 2**
>
> ## **Weaknesses2: Modular vs. Combinatorial Attribute Learning**
>
> We thank the reviewer for this accurate assessment. We fully acknowledge that TTS-Hub, as a data-driven framework, fundamentally relies on attribute-annotated datasets to train its LoRA modules.
>
> However, we respectfully wish to clarify a critical distinction regarding the nature and complexity of the data required. The core advantage of TTS-Hub lies in its shift from "combinatorial data" to "modular data," which significantly mitigates the data collection burden compared to existing methods.
>
> 1. Modular Data vs. Intersectional Samples
>
> - The Challenge for Baselines: Traditional reference-based or end-to-end methods often require "intersectional samples" to achieve multi-attribute control. For example, to clone a specific target voice, one might need a reference audio that is simultaneously *"Elderly AND Happy AND Scottish."* As the number of attributes increases, finding samples that represent the intersection of all these features becomes exponentially difficult.
> - TTS-Hub’s Efficiency: In contrast, our framework decomposes this requirement. We only need to collect independent, atomic datasets for each attribute (e.g., a subset for "Elderly," a separate subset for "Happy," and another for "Scottish"). We do not require samples that possess all these traits simultaneously. This decoupling drastically lowers the barrier for dataset construction.
>
> 2. Arithmetic Composition for Zero-Shot Generalization
> Because our LoRA modules are trained independently, the "sizeable dataset" mentioned by the reviewer is actually a collection of smaller, manageable subsets.
>
> - Inference Mechanism: Once these atomic modules are trained, TTS-Hub generates complex combinations via arithmetic addition (e.g., $\Delta_{Elderly} + \Delta_{Happy} + \Delta_{Scottish}$) during the inference stage.
> - Generalization: This allows the model to synthesize attribute combinations that never existed in the training set. We effectively substitute the need for a massive, all-encompassing dataset with a flexible composition mechanism.
>
> 3. Linear Scalability
> The dependency on data in our system scales linearly rather than exponentially. To add a new capability to the system, we only need to annotate data for that specific attribute. Once added to the Hub, it can immediately interact with all existing modules, exponentially expanding the system's expressive space without requiring a dataset that covers every possible permutation.

---

> ### Author Response · Authors · 2025-11-21
> **Replies to Reviewer fo7r for Question 1**
>
> ## **Q1: Clarifying the Role of LoRA Adaptation vs. Zero-Shot Prompting**
>
> We thank the reviewer for raising this critical question regarding experimental rigor. You have touched upon a fundamental motivation of our work: How can we effectively adapt a general-purpose TTS model to synthesize specific attributes that are under-represented or absent in its original pre-training?
>
> We would like to clarify the rationale behind our experimental design and address the backbone dependency issue as follows:
>
> 1. Comparing "Zero-Shot Prompting" vs. "Efficient Adaptation"
> We explicitly acknowledge that our LoRA modules benefit from training on the specific attribute data in our Hub, whereas the Parler-TTS baseline relies on its inherent pre-training distribution (zero-shot prompting).
>
> - Intentional Discrepancy: This difference is by design, not an oversight. Our objective is not to claim that LoRA performs better than Prompting given *identical* training exposure. Rather, we aim to demonstrate that when a user requires precise control over fine-grained attributes (e.g., "Scottish accent" or "High Energy"), which baseline models often struggle to generate via text prompts due to data sparsity—LoRA serves as a superior "knowledge injection" mechanism.
> - The Paradigm Shift: The comparison highlights that for under-represented attributes, relying on the backbone's zero-shot generalization (Baseline) is often insufficient. TTS-Hub proves that modular adaptation is the most effective way to "patch" these missing capabilities into the backbone without the cost of full fine-tuning.
>
> 2. Dependency on Backbone Quality
> We fully agree that the absolute quality of synthesized speech is bounded by the performance of the underlying backbone model.
>
> - Controlled Variable: However, in our experiments, the backbone architecture and pre-trained weights remain identical for both the baseline and TTS-Hub. The backbone serves as a controlled variable.
> - Relative Improvement: Our results demonstrate that TTS-Hub consistently improves attribute controllability (*Win-Attr*) and quality (*AMOS*) over the prompt-based baseline.
> - Architecture Agnostic: Furthermore, to ensure our method isn't dependent on a specific architecture, we validated TTS-Hub on two distinct backbones: Parler-TTS (Transformer-based) and VoiceLDM (Diffusion-based). The consistent improvements across both architectures (as detailed in the Appendix) suggest that our arithmetic composition strategy provides robust relative gains regardless of the base model's specific strengths or weaknesses.

---

> ### Author Response · Authors · 2025-11-21
> **Replies to Reviewer fo7r for Question 2**
>
> ## **Q2: Validating GPT-4o-Audio-Preview as a Reliable Evaluation Proxy**
>
> We appreciate the reviewer for raising this crucial methodological question. We fully agree that the use of the GPT-4o-Audio-Preview API as an evaluation proxy requires rigorous empirical validation to ensure its judgments align with human auditory perception.
>
> To address this, we conducted a "Ground-Truth Verification" study during the rebuttal phase to statistically quantify the reliability of the LLM-based metrics.
>
> 1. Methodology: Establishing the Human Ground Truth
> To measure the effectiveness of the GPT-4o-Audio-Preview, we compared its decisions against a consolidated human consensus.
>
> - Sampling: We randomly sampled a validation set of 100 audio pairs (comparing our method vs. the prompt-based baseline) from the test output.
> - Human Baseline: A panel of 30 distinct experts performed the exact same pairwise preference task as the LLM (determining "Win/Loss/Tie" based on Attribute Expressiveness and Audio Quality). We aggregated their ratings using a majority vote mechanism to establish the "Human Ground Truth."
> - Statistical Metric: We employed Fleiss’ Kappa ($\kappa$) to measure the inter-rater agreement between the LLM’s decisions and the human ground truth. Additionally, we calculated Krippendorff’s Alpha ($\alpha$) among the human experts to ensure the ground truth itself was reliable.
>
> 2. Statistical Results & Interpretation
>
> | Metric | Validation Setup | Statistical Measure | Result | Interpretation |
> | --- | --- | --- | --- | --- |
> | AMOS | 100 samples rated by 30 experts (1-5 scale) | Krippendorff's Alpha ($\alpha$) | 0.8362 | High Consistency |
> | Win-Attr | 100 pairs evaluated by Human Majority vs. GPT-4o-Audio-Preview | Fleiss' Kappa ($\kappa$) | 0.8135 | Strong Alignment |
> | Win-Qual | 100 pairs evaluated by Human Majority vs. GPT-4o-Audio-Preview | Fleiss' Kappa ($\kappa$) | 0.8417 | Strong Alignment |
>
> As detailed in the Table above, the results demonstrate high statistical alignment:
>
> - Internal Consistency: The human expert panel achieved a $\kappa$ > 0.8, confirming that our human baseline is highly consistent and reliable.
> - LLM-Human Alignment: The agreement between GPT-4o-Audio-Preview and the Human Majority yielded Kappa scores exceeding 0.8 for both *Win-Attr* and *Win-Qual*. According to standard psychometric benchmarks, this indicates "Strong Agreement."
>
> **Conclusion:** These empirical results validate that the GPT-4o-Audio-Preview serves as a reliable, scalable, and consistent proxy for human evaluation in this specific task, effectively mirroring expert consensus.

---

### Official Review · Reviewer_qk2A · 2025-10-30

**Soundness:** 2
**Presentation:** 3
**Contribution:** 2
**Rating:** 2
**Confidence:** 3

**Summary:**

This paper introduces TTS-Hub, a novel framework for controllable text-to-speech synthesis. Current prompt-based systems often lack precision, while audio-cloning systems are not flexible. The proposed solution is a modular system built on Low-Rank Adaptation (LoRA). The authors first create a large dataset with clear attribute labels. They then use this data to train a collection of specialized LoRA modules, where each module acts as an expert for a single speech attribute, such as a specific accent or emotion. During synthesis, these expert modules can be combined through simple arithmetic operations like addition, subtraction, and scaling. This allows for fine-grained control over multiple speech attributes without needing to retrain the large backbone model. The authors validate their method on two different TTS backbones and show that it provides superior attribute control compared to a prompt-based baseline.

**Strengths:**

1. Novel Architecture: This work is the first to use multiple LoRA modules as distinct experts to achieve fine-grained, multi-attribute control over TTS. This modular and compositional approach is a significant and original contribution.
2. Comprehensive Experiments: The paper thoroughly evaluates the proposed framework. It systematically tests the effects of adding, subtracting, and scaling the LoRA modules. The authors also employ a diverse set of metrics to robustly measure the effectiveness of attribute control.
3. High Reproducibility: The framework is built entirely on open-source datasets and models. This commitment to using publicly available resources makes the paper's strong results highly reproducible for the research community.

**Weaknesses:**

1. Limited Trustworthiness of AMOS Scores: In Section 5.1, the paper notes that the Attribute Mean Opinion Score (AMOS) was evaluated using only 16 samples per setting. While the difficulty of conducting large-scale human studies is understandable, such a small sample size raises concerns about the statistical significance and reliability of the human evaluation results.
2. Unverified LLM-based Metrics: The paper relies on GPT-4o-Audio-Preview for its Win-Attr and Win-Qual metrics. However, the authors do not provide any evidence to validate the reliability of this LLM as an evaluator for this specific audio task. The credibility of these metrics would be much stronger if the paper included results showing a high correlation between the LLM's judgments and human judgments on a validation set.
3. Insufficient Baselines: The paper only compares its method against a prompt-based version of its own backbone model. It does not include comparisons with other state-of-the-art controllable TTS models mentioned in the Related Works section. This makes it difficult to judge the true performance of TTS-Hub in the context of the current state of the art.
4. Minor Writing and Citation Issues: There are minor errors in the manuscript that detract from its polish. For example, a citation key in Section 5.2 appears as plain text (hu2022lora) instead of a proper reference, indicating a potential issue in the citation process.

### Comment Not Affecting Score
1. Table 3, which shows the trade-off between two weighted attributes, could be more intuitive if presented as a line graph instead of a table.

**Questions:**

1. Could you please clarify the mechanism for LoRA selection from the LoRA Hub? Figure 2 suggests a process where a user's request is parsed into attributes. Is this selection based on simple keyword matching from a user's structured input, or is there a more advanced mechanism for interpreting free-form text requests?
2. In Section 5.3, the attributes $a_1$ and $a_2$ are used as generic placeholders for the subtraction experiments. While the appendix provides specific examples, for clarity in the main text, could you specify which attribute pair was used to generate the results shown in Table 5?

---

> ### Author Response · Authors · 2025-11-21
> **Replies to Reviewer qk2A for Weaknesses 1&2**
>
> ## **Weakness 1: Reliability of AMOS Evaluation**
>
> Thank you for raising this important concern. Below, we clarify the evaluation protocol and provide additional statistical validation showing that our AMOS results are robust and trustworthy.
>
> 1. Actual Observation Count: 160 Ratings per Condition
>
> The “16 samples” mentioned in the paper refer only to distinct synthesized clips selected to ensure phonetic and stylistic diversity—not the total number of observations.
>
> Our evaluation adopts a Many-to-One MOS protocol:
>
> - Expert Pool: 30 trained evaluators
> - Per-Condition Evaluation: Each of the 16 clips is independently rated by 10 randomly assigned experts
> - Total Observations: $16 \text{ clips} \times 10 \text{ raters} = 160 \text{ ratings per AMOS score}$
>
> This observation volume is well above the commonly accepted threshold for MOS stability and ensures sufficient statistical power.
>
> 2. Strong Inter-Rater Reliability (IRR)
>
> To verify that the ratings reflect consistent expert judgment rather than subjective noise, we conducted a dedicated IRR study:
>
> - Validation Set: 100 audio clips
> - Raters: All 30 experts independently rated each clip
> - Metric: Krippendorff’s Alpha
>
> The result demonstrates strong agreement:
>
> | Metric | Evaluation Setup | Statistical Measure | Result | Interpretation |
> | --- | --- | --- | --- | --- |
> | AMOS | 100 samples rated by 30 experts (1–5 scale) | Krippendorff's Alpha ($\alpha$) | 0.8362 | High Consistency |
>
> This confirms that the human evaluation is both statistically reliable and methodologically sound.
>
> **Summary**: Although only 16 unique stimuli were used per setting, each AMOS score is backed by 160 independent expert ratings, and the IRR analysis shows high agreement ($\alpha$ = 0.8362). We will revise Section 5.1 to explicitly describe this protocol and avoid ambiguity.
>
> ## **Weakness 2: Validation of LLM-Based Metrics**
>
> Thank you for pointing out this important issue. We conducted a Ground-Truth Alignment Study comparing LLM judgments with expert consensus.
>
> 1. Experimental Setup
>
> - Data: 100 randomly sampled audio pairs (“Ours” vs. prompt baseline).
> - Human Ground Truth: 30 trained experts conducted pairwise evaluations (Win/Loss/Tie) for both *Attribute Expressiveness* and *Audio Quality*.
>
>     Majority vote was used to form a stable, high-quality ground truth.
>
> - Comparison: We compared GPT-4o-Audio-Preview’s decisions directly against this human ground truth.
>
> 2. Alignment Results
>
> We measured agreement using Fleiss’ Kappa, a standard metric for inter-rater reliability.
>
> | Metric | Evaluation Setup | Statistical Measure | Result | Interpretation |
> | --- | --- | --- | --- | --- |
> | Win-Attr | 100 pairs: Human Majority vs. GPT-4o-Audio-Preview | Fleiss' Kappa ($\kappa$) | 0.8135 | Strong Alignment |
> | Win-Qual | 100 pairs: Human Majority vs. GPT-4o-Audio-Preview | Fleiss' Kappa ($\kappa$) | 0.8417 | Strong Alignment |
>
> ---
>
> Both Win-Attr and Win-Qual exhibit strong alignment ($\kappa$ > 0.80) with expert judgments, confirming that GPT-4o-Audio-Preview is a reliable evaluator for this task.
>
> 3. We further include **Gemini-2.5 Pro** as an additional evaluation proxy.
>
> Gemini-2.5 Pro has demonstrated state-of-the-art audio understanding and reasoning on the MMAU audio-reasoning benchmark [3], making it a strong candidate for proxy-based evaluation. We further evaluated its agreement with human majority votes, obtaining Fleiss’ Kappa scores of 0.8445 (Win-Attr) and 0.8509 (Win-Qual). Notably, Gemini-2.5 Pro shows higher agreement with human judgments than GPT-4o-Audio-Preview. Together, these results confirm the robustness of our metric design.
>
> | Metric  | Validation Setup                | Statistical Measure     | Result | Interpretation  |
> |--------------|----------------------------------------|------------------------------|:----------:|----------------------|
> | Win-Attr | Human Majority vs. Gemini-2.5 Pro      | Fleiss' Kappa ($\kappa$)     | 0.8445 | Strong Alignment     |
> | Win-Qual | Human Majority vs. Gemini-2.5 Pro      | Fleiss' Kappa ($\kappa$)     | 0.8509 | Strong Alignment     |
>
>
> We also add references to recent work [1,2,3] that *adopts LLM-based metrics to evaluate speech quality* to better contextualize our use of such proxies within current practice in the community. We will include the above discussion in the revised version to strengthen the credibility of our LLM-based metrics.
>
> [1] Lou, Haowei, et al. "StyleSpeech: Parameter-efficient Fine Tuning for Pre-trained Controllable Text-to-Speech." Proceedings of the ACM International Conference on Multimedia in Asia. 2024.
>
> [2] Zhan, Jun, et al. "Vstyle: A benchmark for voice style adaptation with spoken instructions." arXiv:2509.09716. 2025.
>
> [3] Manku, Ruskin Raj, et al. "EmergentTTS-Eval: Evaluating TTS Models on Complex Prosodic, Expressiveness, and Linguistic Challenges Using Model-as-a-Judge." arXiv:2505.23009. 2025.

---

> ### Author Response · Authors · 2025-11-21
> **Replies to Reviewer qk2A for Weaknesses 3&4&Improving Visualization**
>
> ## **Weakness 3：Comparison with State-of-the-Art Controllable TTS Models**
>
> Thank you for highlighting this important issue. We agree that the original evaluation—focused on prompt-based baselines within the same backbone—did not fully situate TTS-Hub within the broader SOTA landscape. To address this, we conducted new experiments comparing TTS-Hub with two representative and competitive controllable TTS systems.
>
> 1. New Baselines Included
>
> We incorporated two widely used SOTA models that rely on structured control or reference signals:
>
> - CosyVoice 2 [1], high-quality reference-based cloning
> - StyleSpeech [2], style-vector-based controllable TTS
>
> To ensure fairness, we evaluate them under two realistic scenarios:
>
> - Matched Reference: the model receives a reference already containing the target attribute
> - Blank Reference + Prompt: the model generates the target attribute *without* a perfect reference. This setting simulates scenarios where no appropriate reference is available—one of the core motivations of our work.)
>
> Note: TTS-Hub requires no reference in any scenario.
>
> | Model | Condition | WER $\downarrow$ | AMOS $\uparrow$ | Win-Attr $\uparrow$ | Win-Qual $\uparrow$ |
> | --- | --- | --- | --- | --- | --- |
> | CosyVoice 2 | Matched Reference | 4.394 | 4.048 | 0.652 | 0.715 |
> | CosyVoice 2 | Blank Ref + Prompt | 4.322 | 3.314 | 0.387 | 0.464 |
> | StyleSpeech | Matched Reference | 4.713 | 3.623 | 0.609 | 0.651 |
> | StyleSpeech | Blank Ref + Prompt | 4.604 | 3.207 | 0.362 | 0.389 |
> | TTS-Hub (Ours) | No Reference Needed | 4.622 | 3.661 | 0.600 | 0.409 |
>
> [1] Du, Zhihao, et al. "Cosyvoice 2: Scalable streaming speech synthesis with large language models." *arXiv preprint arXiv:2412.10117* (2024).
>
> [2] Lou, Haowei, et al. "StyleSpeech: Parameter-efficient Fine Tuning for Pre-trained Controllable Text-to-Speech." *Proceedings of the 6th ACM International Conference on Multimedia in Asia*. 2024.
>
> 2. Analysis and Discussion
> The results reveal a critical trade-off in current SOTA methods compared to TTS-Hub:
>
> - Reference Dependency: SOTA models achieve high performance (high AMOS) only when provided with a matched reference. However, when the specific reference is unavailable (Neutral Ref condition), their ability to follow style instructions degrades significantly, as indicated by the sharp drop in AMOS and Win-Attr scores.
> - The "Black Box" vs. Explicit Control: Cloning models operate as "black boxes," entangling identity and prosody. They fail in composite or rare scenarios (e.g., *"A high-pitched, elderly Indian male"*) because finding a single reference audio containing all these traits is practically impossible.
> - TTS-Hub's Advantage: In contrast, TTS-Hub achieves precise, fine-grained control without needing any reference audio. By arithmetically composing modules (e.g., $\Delta_{Indian} + \Delta_{Sad}$), it allows for creative generation rather than just imitation.
>
> ## **Weakness 4：Correction of Writing and Citation Issues**
>
> Thank you for pointing out the citation formatting issue and other minor writing inconsistencies. We appreciate the reviewer’s careful reading.
>
> 1. Citation Fix: The incorrect citation key in Section 5.2 (`hu2022lora`) has been corrected to the proper reference format (Hu et al., 2022).
> 2. Full Proofreading: We performed a comprehensive proofreading of the manuscript, correcting typographical errors, unifying bibliography formats, and refining language for clarity and polish.
>
> We are committed to ensuring the final paper is clean, consistent, and professionally presented.
>
> ## **Comment: Improving Visualization of Table 3**
>
> We replaced Table 3 with a more intuitive line plot (Figure 9). The new figure visualizes AMOS scores of both attributes ($a_1$, $a_2$) across interpolation weights ($w_1 \in [0,1]$), clearly revealing the smooth **crossover behavior**—as $w_1$ increases, AMOS-$a_1$ rises while AMOS-$a_2$ declines accordingly. This visualization more effectively illustrates the fine-grained controllability enabled by arithmetic LoRA composition.

---

> ### Author Response · Authors · 2025-11-21
> **Replies to Reviewer qk2A for Questions 1&2**
>
> ## **Q1：Mechanism for LoRA Mapping in the LoRA Hub**
>
> We thank the reviewer for the question. In our implementation, the LoRA modules are explicitly selected by the user, rather than inferred from text description of user intent. This choice ensures that our experiments cleanly evaluate the attribute hubs and their composition, without introducing error propagation from intent-parsing mistakes.
>
> However, TTS-Hub is fully compatible with automated parsing. In internal tests, we found that both keyword-based and LLM-based parsing can map natural-language requests to the corresponding LoRA attributes. We did not adopt these mechanisms in the paper to avoid confounding the evaluation with NLP parsing errors, but the framework can readily support them.
>
> ## **Q2：Clarification of Attribute Pairs Used in Table 5**
>
> Thank you for pointing out this ambiguity. We apologize for the lack of clarity in Section 5.3.
>
> 1. Clarification of Experimental Setup
>
> The results in Table 5 are averaged across four representative attribute pairs, rather than based on a single $a_1,a_2$ combination. This aggregation provides a more stable and general evaluation of the subtraction behavior $\Delta_{a_1+a_2} - \Delta_{a_2} \approx \Delta_{a_1}$.
>
> 2. Attribute Pairs Used in Table 5
>
> For completeness, the four specific attribute pairs are:
>
> - Pair 1: US Accent ($a_1$) + Teens ($a_2$) → subtract *Teens*
> - Pair 2: UK Accent ($a_1$) + Age > 60 ($a_2$) → subtract *Over 60*
> - Pair 3: Happy ($a_1$) + Indian Accent ($a_2$) → subtract *Indian*
> - Pair 4: Sad ($a_1$) + Male ($a_2$) → subtract *Male*
>
> These four combinations cover Accent, Age, Emotion, and Gender, ensuring diversity in the subtraction evaluation.
>
> We have revised the caption of Table 5 to explicitly state that the results are averaged across these four attribute pairs, and we direct readers to Appendix D (Tables 19–22) for full per-pair breakdowns.

---

### Official Review · Reviewer_S5Dp · 2025-10-30

**Soundness:** 3
**Presentation:** 3
**Contribution:** 3
**Rating:** 4
**Confidence:** 4

**Summary:**

This paper introduces TTS-Hub, a novel framework for controllable text-to-speech (TTS) synthesis. The key idea is to leverage modular Low-Rank Adaptation (LoRA) adapters, each fine-tuned for a specific speech attribute (e.g., accent, age, emotion), and to enable flexible, fine-grained control by composing these adapters arithmetically at inference time. The authors construct a comprehensive data hub covering 32 fine-grained speech attributes and demonstrate the approach on two backbone TTS models (Parler-TTS and VoiceLDM). Extensive experiments show that TTS-Hub outperforms prompt-based baselines in both single- and multi-attribute control, as measured by human and automatic metrics.

**Strengths:**

1. The paper is the first to systematically explore modular LoRA composition for controllable TTS, moving beyond prompt-based or reference-based control.

2. The authors provide thorough experiments,  and is validated on two different TTS architectures, demonstrating robustness and generality.

**Weaknesses:**

1. While the method generally outperforms baselines, there are cases in the demo where the improvement is marginal or the quality drops. It's better to do some qualitative analysis or audio examples highlighting these failure modes and their possible causes.

2. The paper does not propose or analyze advanced fusion strategies to mitigate this, nor does it deeply investigate the causes of interference.

**Questions:**

1. Have you explored more advanced composition strategies (e.g., gating, attention, or learned fusion) to address the observed degradation when fusing many LoRA modules?

2. Can you provide more insight or analysis (e.g., ablations, visualizations) into why attribute interference occurs and how it might be mitigated?

---

> ### Author Response · Authors · 2025-11-21
> **Replies to Reviewer S5Dp for Weaknesses 1&2**
>
> ## **Weakness 1：Qualitative Assessment and Failure Mode Analysis**
>
> We appreciate this insightful observation. We acknowledge that while TTS-Hub excels in single- and dual-attribute control, synthesis quality can degrade when naively stacking numerous or conflicting attributes (e.g., $\ge$ 3), as indicated in our quantitative results (Tables 15 and 16).
>
> To rigorously address this, we have conducted a qualitative spectral analysis comparing successful samples against failure cases. We have included these Mel-spectrogram comparisons in Appendix D of the revised manuscript.
>
> 1. Qualitative Analysis Findings
> We examined the spectral evolution from a base sample (*US Accent*) to a complex composite (*US + Teens + Happy + High Pitch*):
>
> - Preservation in Low-Order Composition: In single and dual-attribute settings (*US* and *US + Teens*), the spectrograms exhibit sharp harmonic structures and clearly defined formants, confirming that the model preserves acoustic fidelity when handling compatible attributes.
> - Degradation in High-Order Composition: In the failure case (*US + Teens + Happy + High Pitch*), we observe distinct spectral smearing and a loss of harmonic definition in the high-frequency bands compared to the base model. This "blurring" effect correlates directly with the perceptual artifacts (e.g., metallic texture or reduced clarity) observed in the audio.
>
> 2. Diagnosis of Failure Modes
> Based on this analysis, we identify two primary causes for the quality drop, which we mechanistically define in Q2:
>
> - Manifold Shift (Linearity Limits): Our baseline method relies on linear arithmetic addition. When combining four distinct LoRA modules, the cumulative parameter shift pushes the backbone model too far from its pre-trained manifold, resulting in the observed spectral degradation.
> - Attribute Entanglement (Pitch Conflict): The specific combination of *Teens*, *Happy*, and *High Pitch* presents a semantic conflict, as all three strongly influence the Fundamental Frequency (F0). Naively stacking these modules creates an excessive shift in pitch-related features, disrupting natural prosody.
>
> **Action Taken:**
>
> 1. We have added the **Mel-spectrogram comparison Figure 6** to the revised paper to visually demonstrate these failure modes.
> 2. We have expanded the discussion in Section 5.4 to include mitigation strategies, such as applying attenuation factors (e.g., reducing the weight of *High Pitch* when *Happy* is present) to restore spectral clarity (supported by new quantitative data in the table under response Q2).
>
> ## **Weakness 2：Analyzing LoRA Conflicts and Exploring Mitigation Strategies**
>
> We thank the reviewer for this critical feedback. We have significantly expanded our analysis to include (1) a geometric investigation into interference and (2) an exploration of advanced fusion strategies.
>
> 1. Deep Investigation: The Mechanism of Interference
> To understand *why* interference occurs, we analyzed the geometry of the LoRA parameter space by calculating the Cosine Similarity between the weight update matrices ($\Delta W$) of all adapters. Our analysis (visualized as Figure 7 in the revised Appendix and detailed in Response Q2) identifies two distinct failure modes:
>
> - Type I: Vector Conflict (Destructive Interference): Mutually exclusive attributes exhibit negative cosine similarity, leading to cancellation.
> - Type II: Attribute Entanglement (Oversaturation): Closely related attributes exhibit high positive similarity, leading to parameter instability.
>
> 2. Advanced Fusion Strategies
> We explored two strategies to mitigate these issues:
>
> - Strategy A: Weighted Arithmetic Composition (Greedy Search). We found that applying non-uniform weighting (e.g., setting weights to $0.8$) significantly reduces perceptual degradation. We provide a new WER analysis (Table R3 in Q2) to validate this.
> - Strategy B: Learned Fusion via Mini-MoE. Addressing your call for "advanced fusion," we implemented a Learned Mini-Mixture-of-Experts (MoE) router. This learnable module dynamically predicts optimal mixing weights. Please refer to our detailed response in Q1 for the implementation and results of this strategy.
>
> Conclusion
> Our investigations confirm that interference is often a geometric vector conflict. While we demonstrate that a Mini-MoE (see Q1) can solve this automatically, our results (see Q2) suggest that Weighted Arithmetic Composition remains a highly efficient baseline when weights are appropriately tuned. We have added these analyses to the revised manuscript.

---

> ### Author Response · Authors · 2025-11-21
> **Replies to Reviewer S5Dp for Question 1**
>
> ## **Q1：Exploring Advanced Fusion Strategies for Multi-LoRA Composition**
>
> Thank you for this helpful suggestion. Following your comment, we conducted additional experiments to (1) analyze whether advanced fusion can mitigate interference, and (2) validate that TTS-Hub is compatible with learnable gating mechanisms. We will add these findings and tables to the revised manuscript.
>
> 1. Mini-MoE Router: A Learnable Fusion Mechanism
>
> We implemented a lightweight Mini-Mixture-of-Experts router to automatically predict LoRA mixing weights. The router (a 3-layer MLP) takes pooled text-encoder embeddings as input and outputs sparse weights over 7 LoRA experts. To test robustness, we intentionally included a “Sad” LoRA expert that was *not* seen during router training, enabling us to evaluate whether the router can correctly suppress redundant modules.
>
> 2. Results: Effective Conflict Resolution & Redundancy Handling
>
> The Mini-MoE router demonstrates two desirable behaviors:
>
> - Mutual exclusivity: Conflicting LoRAs (e.g., US vs. UK) are cleanly separated.
> - Redundancy robustness: The unseen *Sad* expert receives 0.00 weight even when prompted with emotional text.
>
> *(Active weights in bold)*
>
> | Input Text Prompt | US | UK | Canada | 20s | 30s | 50s | Sad (Redundant*) |
> | --- | --- | --- | --- | --- | --- | --- | --- |
> | *"A speaker in their twenties with a **British** accent."* | 0.00 | **1.00** | 0.00 | **1.00** | 0.00 | 0.00 | 0.00 |
> | *"A speaker in their thirties with an **American** accent."* | **1.00** | 0.00 | 0.00 | 0.00 | **1.00** | 0.00 | 0.00 |
> | *"A speaker in their fifties with a **Canadian** accent."* | 0.00 | 0.00 | **1.00** | 0.00 | 0.00 | **1.00** | 0.00 |
> | *"A very sad crying voice."* (OOD Prompt) | 0.00 | 0.00 | 0.00 | 0.00 | 0.00 | 0.00 | **0.00** |
> - *Note: The "Sad" expert was included in the pool but excluded from router training.*
> - “Sad” LoRA was included but never trained—0.00 confirms proper suppression.
>
> 3. Quality Improvement with Learned Fusion
>
> Compared with naive equal-weight addition ($w=1.0$), the Mini-MoE router provides measurable gains in multi-attribute synthesis quality.
>
> | Fusion Strategy | AMOS ($\uparrow$) | Win-Attr ($\uparrow$) | Win-Qual ($\uparrow$) |
> | --- | --- | --- | --- |
> | Naive Addition (Baseline) | 3.650 | 0.622 | 0.354 |
> | Learned Mini-MoE (Ours) | 3.679 | 0.700 | 0.371 |
> | *Improvement* | *+0.029* | *+0.078* | *+0.017* |
>
> 4. Discussion: The Trade-off
> The Mini-MoE experiment confirms that TTS-Hub supports advanced gating mechanisms. However, we emphasize that the primary contribution of TTS-Hub is its **zero-shot flexibility** via arithmetic composition. While MoE improves quality, it introduces training overhead. We view arithmetic composition as the fundamental "instruction set," while MoE serves as an optional enhancement for quality-critical applications.  These experiments show that **TTS-Hub naturally supports advanced fusion strategies**, and a learned Mini-MoE router can effectively alleviate LoRA interference and improve synthesis quality. However, we emphasize that our core contribution remains the **simplicity, zero-shot flexibility, and modularity** enabled by arithmetic composition. MoE-based methods serve as an optional enhancement for scenarios where users prefer higher quality over training-free flexibility.

---

> ### Author Response · Authors · 2025-11-21
> **Replies to Reviewer S5Dp for Question 2**
>
> ## **Q2：Analyzing and Mitigating Attribute Interference**
>
> We thank the reviewer for requesting this deeper analysis. To provide a mechanistic explanation, we visualized the LoRA parameter space by calculating the Cosine Similarity between the flattened weight update matrices ($\Delta W$) of all trained adapters.
>
> Our analysis (visualized as **Figure 7** in the revised Appendix) reveals that "interference" is not random, but stems from specific geometric relationships:
>
> 1. Type I: Vector Conflict (Destructive Interference): Mutually exclusive attributes (e.g., `Gender: Male` vs. `Gender: Female`) exhibit strong negative cosine similarity (ranging from -0.7 to -0.9).
>     - Mechanism: These update vectors point in opposing directions. Naive arithmetic addition results in mathematical cancellation.
>     - Consequence: This leads to a "neutralized" output where neither attribute is distinct.
> 2. Type II: Attribute Entanglement (Constructive Interference): Different emotion adapters exhibit high positive cosine similarity.
>     - Mechanism: This collinearity suggests different emotions share a common prosodic subspace. Direct addition causes the magnitude of the fused vector to become excessively large.
>     - Consequence: This pushes parameters off the stable manifold (oversaturation), causing the synthesized speech to become unstable or distorted.
>
> Mitigation Strategy & Verification:
> These geometric insights validate the necessity of the scaling factor ($w$) proposed in our framework. We mitigate interference by tuning $w$ based on the interference type (e.g., Norm-Constraint Scaling for Type II entanglement).
>
> To verify this empirically, we analyzed the impact of scaling on intelligibility (Word Error Rate) in a scenario prone to oversaturation (*US Accent + Age 30s*). As shown in Table R3, manually reducing the weights from $1.0$ to $0.8$ restores spectral clarity and significantly reduces WER.
>
> Table R3: Impact of Weight Scaling on Intelligibility (Type II Interference Mitigation)*Evaluation of WER using Whisper-Large-v3. Baseline is naive arithmetic addition ($\alpha=1.0$).*
>
> | Weight Config | WER ($\downarrow$) | Observation |
> | --- | --- | --- |
> | A=1.0, B=1.0 | 12.50% | Degraded: Audio artifacts and clipping due to parameter saturation. |
> | A=1.0, B=0.8 | 6.25% | Unstable: Minor distortions affect phonetic clarity. |
> | A=0.8, B=0.8 | **4.52%** | Optimal: High fidelity restored; clean articulation. |
> | A=0.6, B=0.6 | 5.12% | Conservative: Clear audio, though style intensity is milder. |
>
> *Configuration: "US Accent" and "Age 30s" LoRAs. Transcription Example (1.0/1.0): "The sunlight is bright and... [noise]... pleasant today." vs (0.8/0.8): "...very pleasant today."*

---

### Official Review · Reviewer_BCXY · 2025-10-31

**Soundness:** 3
**Presentation:** 3
**Contribution:** 2
**Rating:** 4
**Confidence:** 4

**Summary:**

The paper proposes TTS-Hub, a controllable TTS framework that fine-tunes attribute-specialist LoRA adapters (e.g., for accent, age, emotion, gender, energy, pitch) on curated, attribute-labeled subsets drawn from four public corpora. At inference, multiple adapters are algebraically composed (addition, subtraction, scalar weighting) and injected into a frozen backbone to realize multi-attribute control. The paper evaluates with human Attribute MOS (AMOS), LLM-judge win rates (Win-Attr/Win-Qual using GPT-4o-Audio-Preview), and objective metrics (MCD, F0-RMSE, SSIM), reporting consistent gains over prompt-only control and reasonable behavior under composition

**Strengths:**

- The modular attribute-specialist LoRA design is simple, extensible, and model-agnostic: train once per attribute, compose at inference without touching the backbone. The paper demonstrates this on Parler-TTS (main) and VoiceLDM (appendix), showing architectural robustness.

- Single-attribute control clearly improves over prompt-only baselines in AMOS and Win-Attr across many attributes; the tables quantify consistent, meaningful margins (e.g., ">60 age" gains). This supports the claim that LoRAs capture sharper acoustic factors than natural-language prompts alone.

**Weaknesses:**

- While the proposed method is promising, novelty seems to be incremental. Composing LoRAs arithmetically is well-trodden in NLP/vision (e.g., LoRAHub for cross-task generalization; Mixture-of-LoRA-Experts), and LoRA for TTS has been used for emotion/style/personalization (e.g., StyleSpeech/EELE/LorP-TTS). I would suggest the authors compare with dynamic composition (Huang et al., 2024a; Wu et al., 2024) and TTS-specific LoRA works (Lou et al., 2024; Qi et al., 2024; Bondaruk & Kubiak, 2025).

- Evaluation validity and reliance on LLM-as-judge: Win-Attr/Win-Qual depend on GPT-4o-Audio-Preview, which may encode hidden priors and can be sensitive to prompting and audio loudness/length. Stronger human listening (e.g., MUSHRA/CMOS with inter-rater reliability) and task-relevant automatic metrics (ASR WER, F0/energy controllability accuracy via independent estimators) would strengthen claims.

- Limited baselines/backbones and scaling limits: Comparisons are primarily to prompt-only control on Parler-TTS; many contemporary TTS systems expose structured control tokens/encoders (e.g., style tokens, prosody/vector controls) that would be stronger baselines.

**Questions:**

- What is the runtime/memory overhead of injecting multiple adapters, and how does latency scale with the number of composed LoRAs on Parler-TTS vs. VoiceLDM?

- Can you report fairness diagnostics (e.g., accent/gender/age bias) and failure cases for rare attribute combinations or cross-lingual accents?

---

> ### Author Response · Authors · 2025-11-21
> **Replies to Reviewer BCXY for Weakness 1**
>
> #### **Weakness 1：Methodological Positioning and Distinctions from Prior LoRA Work**
>
>
> Thank you for highlighting these related LoRA-based directions. While prior methods leverage LoRAs and their composition to efficiently adapt to new tasks (NLP/Vision) or holistic styles (TTS), our work addresses a fundamentally different research problem. TTS-Hub is designed to answer the following core question:
>
> >**Can speech attributes be disentangled into atomic LoRAs and composed flexibly and continuously via simple LoRA arithmetic?**
>
> This question has not been explored in prior LoRA literature. Existing LoRA systems do not investigate whether deeply entangled acoustic factors—such as pitch, prosody, emotion, timbre, and accent—can be factored into modular units and recombined to generate unseen multi-attribute behaviors. To clarify the conceptual distinctions, we provide a concise comparison below:
>
> | **Approach Category** | **Representative Works** | **Methodology** | **Limitation / Insight for TTS-Hub** |
> | --- | --- | --- | --- |
> | **Dynamic Composition (NLP / Vision)** | LoRAHub [1]; MoLE [2] | Combines LoRA modules via gradient-free search or routing for *cross-task transfer*. | **Limitation:** Focus on task metrics; lacks interpretable attribute-level control.  **Insight:** Motivates deterministic arithmetic for transparent attribute manipulation. |
> | **LoRA for TTS** | EELE [3]; LoRP-TTS [4]; StyleSpeech [5] | Models *holistic* styles (speaker/emotion) as indivisible units. | **Limitation:** Cannot decouple fine-grained factors; cannot generalize to unseen combinations.  **Insight:** Motivates attribute decomposition. |
> | **TTS-Hub (Ours)** | **TTS-Hub** | **Atomic Attribute LoRAs + Arithmetic Composition** | **Advantage:** Enables semantic disentanglement, additive/subtractive operations, and zero-shot multi-attribute creation. |
>
> We will incorporate these clarifications in the revised version to strengthen the methodological positioning of our contribution.
>
> [1] Huang, Chengsong, et al. "Lorahub: Efficient cross-task generalization via dynamic lora composition." *arXiv preprint arXiv:2307.13269* (2023).
>
> [2] Wu, Xun, Shaohan Huang, and Furu Wei. "Mixture of lora experts." *arXiv preprint arXiv:2404.13628* (2024).
>
> [3] Qi, Xin, et al. "EELE: Exploring Efficient and Extensible LoRA Integration in Emotional Text-to-Speech." *2024 IEEE 14th International Symposium on Chinese Spoken Language Processing (ISCSLP)*. IEEE, 2024.
>
> [4] Bondaruk, Łukasz, and Jakub Kubiak. "LoRP-TTS: Low-Rank Personalized Text-To-Speech." *arXiv preprint arXiv:2502.07562* (2025).
>
> [5] Lou, Haowei, et al. "StyleSpeech: Parameter-efficient Fine Tuning for Pre-trained Controllable Text-to-Speech." *Proceedings of the 6th ACM International Conference on Multimedia in Asia*. 2024.

---

> ### Author Response · Authors · 2025-11-21
> **Replies to Reviewer BCXY for Weaknesses 2&3**
>
> #### **Weakness 2: Validation of Subjective and LLM-Based Evaluation Metrics**
>
> We performed studies to ensure the statistical reliability of our evaluations, including assessing human annotator consistency and verifying two independent LLM proxies (GPT-4o-Audio-Preview and Gemini-2.5 Pro).
>
> - measuring inter-rater consistency for human AMOS ratings. We calculated Krippendorff’s Alpha across 100 samples rated by 30 experts.
>
> - verifying LLM-as-judge alignment with human perception. We used human-majority voting as ground truth, and measured alignment using Fleiss’ Kappa.
>
> | Metric   | Model                | Validation Setup             | Statistical Measure        | Result  | Interpretation    |
> |----------|-----------------------|-------------------------------|-----------------------------|-------|-------------------|
> | AMOS     | Human Experts         | 100 samples × 30 experts      | Krippendorff’s Alpha        | 0.8362  | High Consistency  |
> | Win-Attr | GPT-4o  | Human Majority vs. LLM Judge  | Fleiss’ Kappa ($\kappa$)    | 0.8135  | Strong Alignment  |
> | Win-Qual | GPT-4o  | Human Majority vs. LLM Judge  | Fleiss’ Kappa ($\kappa$)    | 0.8417  | Strong Alignment  |
> | Win-Attr | Gemini-2.5 Pro        | Human Majority vs. LLM Judge  | Fleiss’ Kappa ($\kappa$)    | 0.8445  | Strong Alignment  |
> | Win-Qual | Gemini-2.5 Pro        | Human Majority vs. LLM Judge  | Fleiss’ Kappa ($\kappa$)    | 0.8509  | Strong Alignment  |
>
> These results show that (1) human subjective ratings are stable and highly consistent, and (2) both GPT-4o-Audio-Preview and Gemini-2.5 Pro exhibit strong agreement with human judgments. The evidence strongly proves the reliability of our LLM-derived metrics. We also add references to recent work [6,7] that adopts *LLM-based metrics to evaluate speech quality* to better contextualize our use of such proxies within current practice in the community.
>
> Beyond subjective and proxy evaluations, we incorporated new and existing metrics:
> - Pitch & Timbre Control: Already reported in Tables 2 and 7–10, using F0-RMSE and MCD to measure controllability and acoustic fidelity.
> - Intelligibility (New): Following your suggestion, we added ASR Word Error Rate (WER) evaluation. The results will be updated in the final PDF.
>
> [6] Lou, Haowei, et al. "StyleSpeech: Parameter-efficient Fine Tuning for Pre-trained Controllable Text-to-Speech." Proceedings of the ACM International Conference on Multimedia in Asia. 2024.
>
> [7] Zhan, Jun, et al. "Vstyle: A benchmark for voice style adaptation with spoken instructions." arXiv:2509.09716. 2025.
>
> #### **Weakness 3: Comparison with Structured and Reference-Based Baselines**
>
> We conducted new comparisons with SOTA reference-based models, CosyVoice 2 [9] and StyleSpeech [10], under two settings.
>
> - Matched Reference: Using a reference audio that contains the target attribute.
> - Blank Reference+Prompt: requiring the model to express the target attribute based only on the prompt/style tag. This setting simulates scenarios where no appropriate reference is available—one of the core motivations of our work.
>
> The results are shown in the table below. (Note: WER measures intelligibility; AMOS/Win-Attr measures attribute control accuracy)
>
> | Model | Condition | WER $\downarrow$ | AMOS $\uparrow$ | Win-Attr $\uparrow$ | Win-Qual $\uparrow$ |
> | --- | --- | --- | --- | --- | --- |
> | CosyVoice 2 | Matched Reference | 4.394 | 4.048 | 0.652 | 0.715 |
> | CosyVoice 2 | Blank Ref + Prompt | 4.322 | 3.314 | 0.387 | 0.464 |
> | StyleSpeech | Matched Reference | 4.713 | 3.623 | 0.609 | 0.651 |
> | StyleSpeech | Blank Ref + Prompt | 4.604 | 3.207 | 0.362 | 0.389 |
> | TTS-Hub (Ours) | No Reference Needed | 4.622 | 3.861 | 0.600 | 0.409 |
>
> The results highlight why we argue reference-based methods have limitations:
>
> - TTS-Hub requires no reference yet matches matched-reference performance of SOTA reference-based models. This highlights a key practical advantage: Real-world users often cannot provide a perfect reference containing all desired traits. TTS-Hub can still generate the target voice through modular arithmetic composition (e.g., for *“a high-pitched, elderly Indian male”*, simply combine $\Delta_{\text{High}} + \Delta_{\text{Elderly}} + \Delta_{\text{Indian}} + \Delta_{\text{Male}}$), without relying on an exemplar that may hard to obtain.
> - Reference-based models degrade without a perfect match. CosyVoice 2 and StyleSpeech perform well only when given a matched reference. Under the *Blank Ref + Prompt* setting, their style-control ability drops substantially, showing that their controllability is tightly constrained by the availability of ideal references.
>
> [9] Du, Zhihao, et al. "Cosyvoice 2: Scalable streaming speech synthesis with large language models." arXiv:2412.10117. 2024.
>
> [10] Lou, Haowei, et al. "StyleSpeech: Parameter-efficient Fine Tuning for Pre-trained Controllable Text-to-Speech." *Proceedings of the 6th ACM International Conference on Multimedia in Asia*. 2024.

---

> ### Author Response · Authors · 2025-11-21
> **Replies to Reviewer BCXY for Questions 1&2**
>
> ## **Q1：Runtime and Memory Overhead of Multi-LoRA Composition**
>
> We conducted detailed experiments to quantify latency and memory overhead on Parler-TTS (Transformer) and VoiceLDM (Diffusion). Results are shown in the table below.
>
> - Parler-TTS
>
> | Configuration | Avg Latency (s) | Latency Scale | Inference Peak VRAM (MB) | Setup Storage Cost (MB) |
> | --- | --- | --- | --- | --- |
> | **Base Model** | **7.10** | **1.00x** | **5827.27** | **-** |
> | +1 LoRA | 7.16 | 1.01x | 5836.11 | +39.57 |
> | +2 LoRA | 7.14 | 1.01x | 5826.74 | +79.13 |
> | +3 LoRA | 7.14 | 1.01x | 5831.92 | +118.70 |
> | +4 LoRA | 7.15 | 1.01x | 5824.51 | +158.27 |
> | +5 LoRA | 7.17 | 1.01x | 5826.75 | +200.81 |
> | +6 LoRA | 7.15 | 1.01x | 5830.47 | +240.76 |
> - VoiceLDM
>
> | Configuration | Avg Latency (s) | Latency Scale | Inference Peak VRAM (MB) | Setup Storage Cost (MB) |
> | --- | --- | --- | --- | --- |
> | **Base Model** | **7.02** | **1.00x** | **3158.62** | **-** |
> | +1 LoRA | 7.09 | 1.01x | 3145.26 | +11.14 |
> | +2 LoRA | 7.08 | 1.01x | 3150.84 | +21.04 |
> | +3 LoRA | 7.09 | 1.01x | 3147.94 | +32.72 |
> | +4 LoRA | 7.07 | 1.01x | 3153.37 | +42.33 |
> | +5 LoRA | 7.06 | 1.01x | 3145.43 | +53.64 |
> | +6 LoRA | 7.07 | 1.01x | 3149.61 | +63.39 |
>
> Based on the empirical data above, we draw the following conclusions regarding the efficiency of TTS-Hub:
>
> - **Negligible Latency Overhead (Constant Time Complexity):**
> As observed in both tables, the latency remains effectively constant (~7.15s for Parler-TTS and ~7.07s for VoiceLDM) regardless of the number of LoRAs injected. The latency scaling factor is consistently **1.01x**.
> *Reasoning:* This is because our arithmetic composition is performed on the weight parameters *before* or *during* the loading phase (i.e., $W_{adapted} = W_{base} + \sum \Delta W_i$). Consequently, the computational graph depth during the forward pass remains identical to the base model, resulting in $O(1)$ inference latency scaling relative to the number of adapters.
> - **Stable Memory Footprint:**
> The Peak VRAM usage during inference does not increase significantly with the addition of LoRA modules (fluctuating marginally around ~5830 MB for Parler-TTS and ~3150 MB for VoiceLDM). This confirms that injecting multiple adapters does not require additional GPU memory for activation storage during generation.
> - **Lightweight Storage Cost:**
> The only linear increase is observed in the "Setup Storage Cost" (disk/RAM required to load the adapter weights). However, this cost is minimal. Even with 6 adapters, the total overhead is only ~240 MB for Parler-TTS and ~63 MB for VoiceLDM, which is negligible compared to the backbone model size.
>
> ## **Q2：Fairness Diagnostics and Failure Cases for Rare and Cross-Lingual Attributes**
>
> We performed additional diagnostics focusing on data-imbalance bias and systematic failure cases.
>
> 1. Fairness Diagnostics: Effects of Data Imbalance
>
> As shown in Figure 3 of the paper, our dataset follows a long-tailed distribution (e.g., US/Indian accents and 20–30 age group dominate, while African/German accents and Teens are under-represented). To quantify the impact of this imbalance, we compared system performance between high-resource and low-resource attribute combinations.
>
> *(Lower AMOS/Win scores indicate greater difficulty in modeling rare attributes)*
>
> | Resource Level | Attribute Combination | AMOS $\uparrow$ | Win-Attr $\uparrow$ | Win-Qual $\uparrow$ |
> | --- | --- | --- | --- | --- |
> | High-Resource | US Accent + Age 20-30 | 3.730 | 0.592 | 0.322 |
> | High-Resource | Indian Accent + Age 30-40 | 3.685 | 0.553 | 0.306 |
> | Low-Resource | African Accent + Age 20-30 | 3.237 | 0.359 | 0.228 |
> | Low-Resource | German Accent + Teens (<20) | 3.464 | 0.461 | 0.275 |
>
> *Finding:* Performance clearly degrades on low-resource combinations, confirming that fairness limitations stem primarily from attribute-specific data scarcity, not from the compositional mechanism itself.
>
> 2. Failure Cases Beyond Fairness
>
> We also observe two structural failure modes intrinsic to compositional control:
>
> - Complexity Overload (4-Attribute Limit).
>
>     When composing ≥4 attributes, quality drops substantially (Tables 4, 15, 16). The accumulated LoRA deltas push parameters away from the pre-trained manifold, leading to prosodic instability or artifacts.
>
> - Attribute Entanglement (Non-Orthogonality).
>
>     Subtraction experiments (Section 5.3) show that some attributes are acoustically coupled. Removing one (e.g., accent) may unintentionally affect others (e.g., emotion), revealing imperfect disentanglement when attributes co-occur frequently in data.
>
>
> Overall, these analyses show that (1) fairness gaps arise primarily from long-tailed data imbalance, and (2) rare or complex multi-attribute combinations expose structural failure limits in the current approach. We will incorporate these experiments and discussions into the revised version.

---

### Official Review · Reviewer_EnEN · 2025-11-06

**Soundness:** 3
**Presentation:** 3
**Contribution:** 3
**Rating:** 6
**Confidence:** 3

**Summary:**

This paper addresses the problem of controllable text-to-speech (TTS) and introduces a modular framework called TTS-Hub. The method builds on LoRA (Low-Rank Adaptations), where each adapter focuses on a specific speech attribute (e.g., accent, emotion, gender), and multiple attributes can be combined at inference time through simple arithmetic operations on LoRA weights.
A reconstructed multi-dataset Data Hub (covering 32 detailed attributes) is used to train the adapters. Experiments on two backbone models (Parler-TTS and VoiceLDM) demonstrate superior single- and multi-attribute control compared to prompt-only baselines, along with intuitive arithmetic effects such as subtraction and scaling.

**Strengths:**

* LoRA-based adapters have been used for TTS before, and arithmetic-based composition has appeared in other domains, but this paper is novel in explicitly exploring arithmetic composition of those adapters for controllable TTS.

* Keeping the backbone frozen while composing LoRAs arithmetically is lightweight, modular, and easily extensible. This is clearly the main highlight of this paper.

* The arithmetic weighting, subtraction, and scaling experiments make the controls intuitive and demonstrate that the model can effectively isolate targeted attributes.

* The paper provides public datasets, detailed descriptions, and code/configs in the supplementary materials, which improves reproducibility. The Data Hub itself is also a valuable contribution to the community.

**Weaknesses:**

* Some intuitive questions remain unanswered. It would be useful to analyze interference between conflicting attributes (e.g., “high pitch” and “low pitch” adapters) to understand the limits of arithmetic composition. It would also help to show how subtracting high pitch versus adding low pitch differs in practice, if at all. Similar logic applies to age attribute as well.

* While the evaluation is generally strong, it lacks several ablations. For example, it would be helpful to know how the choice of layers for LoRA insertion affects performance, or whether deeper layers influence attribute control more strongly.

* As with prompt-based or reference-based methods, this approach depends on the available data and adapters. If a desired attribute is not represented in the dataset or a corresponding LoRA is unavailable, it becomes impossible to synthesize that target attribute.

* The paper argues that existing approaches relying on natural-language prompts or reference audio cloning have limitations, but the experiments only compare against prompt-based control. If no such baseline for reference cloning is available, it might be better to slightly adjust the initial claim in the paper.

* Some reported metrics (e.g., Win-Attr, AMOS) appear subjective and prompt-dependent. It would strengthen the evaluation to report the true variance or ground-truth checks for these metrics.

Minor issues (did not affect score)

* Some citations (e.g., hu2022lora) did not compile correctly.
* Line 078: “flexibel” → “flexible.”
* Figure 1: “Unselcted” → “Unselected.”

**Questions:**

What are the most common examples where arithmetic composition fails? For instance, accent and pitch may be difficult to isolate — keeping one without affecting the other could introduce artifacts.


Can you show results for attribute combinations not seen together during training i.e., zero-shot composition?

---

> ### Author Response · Authors · 2025-11-21
> **Replies to Reviewer EnEN for Weakness 1**
>
> ## **Weakness 1：Ablations on Conflicting Attribute Interference**
>
> We thank the reviewer for these insightful questions regarding the geometry of the LoRA attribute space. You raise two fundamental questions: (1) Do conflicting attributes mathematically cancel each other out (interference)? and (2) Is the subtraction of an attribute equivalent to adding its opposite (e.g., is $\Delta _ {High} + \Delta _ {Low}$)?
>
> To address this, we conducted a new set of "Conflict & Inverse Analysis" experiments focusing on *Pitch* (High vs. Low) and *Age* (20s vs. 60s). The results suggest that while our framework supports flexible composition, the LoRA parameter space is not perfectly linear or orthogonal.
>
> **1. Interference of Conflicting Attributes (e.g., $\Delta _ {High} + \Delta _ {Low}$)**
>
> Intuitively, one might expect that adding "High Pitch" and "Low Pitch" ($\Delta _ {High} + \Delta _ {Low}$) would yield a "Neutral Pitch." However, our results in the Table below show that this results in destructive interference rather than a perfect neutral fusion.
>
> | Combination Strategy | Attribute Target | AMOS $\uparrow$ | Win-Qual $\uparrow$ | MCD $\downarrow$ | F0-RMSE $\downarrow$ | SSIM $\uparrow$ |
> | --- | --- | --- | --- | --- | --- | --- |
> | Base (No Adapter) | Neutral | 3.428 | — | 7.091 | 66.287 | 0.304 |
> | $\Delta _ {High\\_Pitch} + \Delta _ {Low\\_Pitch}$ | Neutral | 2.811 | 0.441 | 8.672 | 89.314 | 0.132 |
> | $\Delta _ {Age\\_20} + \Delta _ {Age\\_60}$ | Neutral | 2.739 | 0.512 | 9.349 | 80.737 | 0.155 |
> - Observation: The synthesized speech attempts to manifest both features simultaneously rather than averaging them. For *Pitch*, this results in unnatural structural artifacts. For *Age* (20s + 60s), we observed incoherence, such as a voice with the speaking rate of a young person but the vocal tremor/roughness of an older speaker.
> - Conclusion: This indicates that attribute-specific LoRA vectors are not strictly anti-parallel. They affect different feature subspaces, and forcing the model to attend to contradictory instructions impairs the naturalness of the backbone.
>
> **2. Subtraction vs. Addition of Opposites (e.g., $-High$ vs. $+Low$)**
>
> We investigated whether subtracting a specific attribute is mathematically or perceptually equivalent to adding its semantic opposite (i.e., does $Base - \Delta _ {High} \approx Base + \Delta _ {Low}$?). The results are summarized in the Table below.
>
> | Operation | Attribute Target | AMOS $\uparrow$ | Win-Attr $\uparrow$ | Win-Qual $\uparrow$ | MCD $\downarrow$ | F0-RMSE $\downarrow$ | SSIM $\uparrow$ |
> | --- | --- | --- | --- | --- | --- | --- | --- |
> | $+\Delta _ {Low\\_Pitch}$ | Low Pitch | 3.657 | 0.512 | 0.338 | 7.000 | 45.805 | 0.222 |
> | $-\Delta _ {High\\_Pitch}$ | Low Pitch | 2.395 | 0.410 | 0.375 | 9.191 | 78.834 | 0.121 |
> | $+\Delta _ {Age\\_60}$ | Elder | 3.273 | 0.582 | 0.504 | 7.126 | 52.973 | 0.364 |
> | $-\Delta _ {Age\\_20}$ | Elder | 1.972 | 0.363 | 0.438 | 8.773 | 90.194 | 0.136 |
> - Observation: while adding the target LoRA ($+\Delta _ {Low}$) successfully shifts the speech toward a deep, resonant vocal quality, subtracting the opposing attribute ($-\Delta _ {High}$) does not produce an equivalent "Low" voice. The same pattern holds for age control:$+\Delta _ {Age\\_60}$ induces clear elderly speech characteristics, whereas $-\Delta _ {Age\\_20}$ does not symmetrically replicate them.
> - Conclusion: Subtraction is not equivalent to adding the opposite. This confirms that the LoRA representations for "High" and "Low" are learned independently and are not simple mathematical inverses of one another ($Base - \Delta _ {a} \neq Base + \Delta _ {b}$).

---

> ### Author Response · Authors · 2025-11-21
> **Replies to Reviewer EnEN for Weaknesses 2&3**
>
> ## **Weakness 2：Ablations on LoRA Insertion Layer Selection**
>
> We thank the reviewer for this insightful suggestion. To systematically investigate this, we conducted a comprehensive Layer-wise Ablation Study using the Parler-TTS backbone.
>
> Specifically, we partitioned the model’s 24 decoder layers into three groups: Bottom (0–8), Mid (9–16), and Top (17–24). We also evaluated inserting LoRA modules exclusively into specific sub-modules: Feed-Forward Networks (FFN) and Cross-Attention. We selected *Accent* and *Age* as representative attributes for this evaluation. The quantitative results are summarized in the table below.
>
> | Attribute | Configuration | AMOS $\uparrow$ | Win-Attr $\uparrow$ | Win-Qual $\uparrow$ | MCD $\downarrow$ | F0-RMSE $\downarrow$ | SSIM $\uparrow$ |
> | --- | --- | --- | --- | --- | --- | --- | --- |
> | Age (20s) | Baseline (Full LoRA) | 3.390 | 0.658 | 0.438 | 7.389 | 73.073 | 0.338 |
> |  | Layers 0–8 (Bottom) | 3.582 | 0.712 | 0.489 | 7.056 | 68.442 | 0.355 |
> |  | Layers 9–16 (Mid) | 3.356 | 0.642 | 0.425 | 7.421 | 74.115 | 0.334 |
> |  | Layers 17–24 (Top) | 3.112 | 0.485 | 0.356 | 7.895 | 82.336 | 0.312 |
> |  | FFN Only | 2.856 | 0.324 | 0.245 | 8.442 | 89.112 | 0.298 |
> |  | Cross-Attention Only | 3.375 | 0.649 | 0.431 | 7.405 | 73.885 | 0.336 |
> | Accent (US) | Baseline (Full LoRA) | 3.736 | 0.664 | 0.446 | 7.282 | 51.677 | 0.362 |
> |  | Layers 0–8 (Bottom) | 3.412 | 0.512 | 0.388 | 7.654 | 55.123 | 0.335 |
> |  | Layers 9–16 (Mid) | 3.455 | 0.534 | 0.392 | 7.598 | 54.882 | 0.341 |
> |  | Layers 17–24 (Top) | 3.021 | 0.356 | 0.289 | 8.102 | 62.441 | 0.310 |
> |  | FFN Only | 3.752 | 0.678 | 0.458 | 7.215 | 50.992 | 0.366 |
> |  | Cross-Attention Only | 3.356 | 0.498 | 0.365 | 7.712 | 56.012 | 0.330 |
>
> Our experiments reveal that different speech attributes rely on distinct model components:
>
> - For the *Age* attribute, the Bottom Layers (0–8) achieved the best performance, outperforming other layer groups. We hypothesize that high-level speaker characteristics, such as timbre and identity, are primarily encoded in the earlier stages of the decoder (Bottom layers), which define the global style of the utterance. Consequently, intervening in these layers is most effective for controlling age.
> - For *Accent*, the FFN modules yielded the superior performance. Accents involve complex linguistic and prosodic patterns. Feed-Forward Networks (FFN) in Transformers are often associated with processing semantic content and storing linguistic knowledge, making them the optimal target for accent adaptation.
> - Across both attributes, the Top Layers (17–24) consistently demonstrated the weakest control capability. This suggests that the deepest layers are more focused on reconstructing fine-grained acoustic details and spectrogram generation based on features established earlier, rendering them less effective for high-level attribute manipulation.
>
> Conclusion: These ablations demonstrate that the optimal LoRA configuration is attribute-dependent. While our standard TTS-Hub approach (inserting into all layers) ensures robust performance across board, these findings suggest a promising direction for future work: implementing attribute-aware layer selection to further optimize efficiency and control precision.
>
>
>
>
> ## **Weakness 3: Clarifying Data Dependency and Modular Extensibility**
>
> We thank the reviewer for the insightful comment. We agree that, like all data-driven systems, TTS-Hub cannot synthesize an attribute that is entirely absent from the Data Hub. The key strength of TTS-Hub is not removing data dependence but maximizing reusability, combinatorial generalization, and extensibility.
>
> - Reusability of Atomic Attributes: Each LoRA adapter encodes an atomic attribute that can be reused across unlimited downstream combinations. Once trained, it becomes a permanent, plug-and-play module in the Hub.
> - Combinatorial Generalization: even if some composite attributes never appear in the data (e.g., “elderly, high-pitched, American female”), TTS-Hub can synthesize them by composing existing atomic adapters, greatly expanding the system’s expressive range.
> - Extensibility: when a missing attribute is needed, users only need to collect a small attribute-specific dataset and train a lightweight LoRA. The backbone remains frozen, and the new module is immediately usable, avoiding costly end-to-end retraining.
>
> In summary, while TTS-Hub necessarily relies on data for training its attribute adapters, it is a highly reusable, scalable, and easily extensible framework that substantially mitigates the practical limitations noted by the reviewer.

---

> ### Author Response · Authors · 2025-11-21
> **Replies to Reviewer EnEN for Weaknesses 4&5&Minor issues**
>
> ## **Weakness 4: Additional Clone-Synthesis Comparison**
>
> We conducted new comparisons with SOTA reference-based models, CosyVoice 2 [1] and StyleSpeech [2], under two settings.
>
> - Matched Reference: Using a reference audio that contains the target attribute.
> - Blank Reference+Prompt: requiring the model to express the target attribute based only on the prompt/style tag. This setting simulates scenarios where no appropriate reference is available—one of the core motivations of our work.
>
> The results are shown in the table below. (Note: WER measures intelligibility; AMOS/Win-Attr measures attribute control accuracy)
>
> | Model | Condition | WER $\downarrow$ | AMOS $\uparrow$ | Win-Attr $\uparrow$ | Win-Qual $\uparrow$ |
> | --- | --- | --- | --- | --- | --- |
> | CosyVoice 2 | Matched Reference | 4.394 | 4.048 | 0.652 | 0.715 |
> | CosyVoice 2 | Blank Ref + Prompt | 4.322 | 3.314 | 0.387 | 0.464 |
> | StyleSpeech | Matched Reference | 4.713 | 3.623 | 0.609 | 0.651 |
> | StyleSpeech | Blank Ref + Prompt | 4.604 | 3.207 | 0.362 | 0.389 |
> | TTS-Hub (Ours) | No Reference Needed | 4.622 | 3.661 | 0.600 | 0.409 |
>
> The results highlight why we argue reference-based methods have limitations:
>
> - TTS-Hub requires n*o* reference yet matches matched-reference performance of SOTA reference-based models. This highlights a key practical advantage: Real-world users often cannot provide a perfect reference containing all desired traits. TTS-Hub can still generate the target voice through modular arithmetic composition (e.g., for *“a high-pitched, elderly Indian male”*, simply combine $\Delta_{\text{High}} + \Delta_{\text{Elderly}} + \Delta_{\text{Indian}} + \Delta_{\text{Male}}$), without relying on an exemplar that may hard to obtain.
> - Reference-based models degrade without a perfect match. CosyVoice 2 and StyleSpeech perform well only when given a matched reference. Under the *Blank Ref + Prompt* setting, their style-control ability drops substantially , showing that their controllability is tightly constrained by the availability of ideal references.
>
> [1] Du, Zhihao, et al. "Cosyvoice 2: Scalable streaming speech synthesis with large language models." *arXiv preprint arXiv:2412.10117* (2024).
>
> [2] Lou, Haowei, et al. "StyleSpeech: Parameter-efficient Fine Tuning for Pre-trained Controllable Text-to-Speech." *Proceedings of the 6th ACM International Conference on Multimedia in Asia*. 2024.
>
> ## **Weakness 5: Statistical Reliability Analysis of Evaluation Metrics**
>
> To ensure that AMOS and LLM-based metrics (Win-Attr/Win-Qual) are reliable and not driven by noise or evaluator bias, we conducted two ground-truth validation studies. The results are summarized in the table below.
>
> | Metric | Validation Setup | Statistical Measure | Result | Interpretation |
> | --- | --- | --- | --- | --- |
> | AMOS | 100 samples rated by 30 experts (1-5 scale) | Krippendorff's Alpha ($\alpha$) | 0.8362 | High Consistency |
> | Win-Attr | 100 pairs evaluated by Human Majority vs. GPT-4o-Audio-Preview | Fleiss' Kappa ($\kappa$) | 0.8135 | Strong Alignment |
> | Win-Qual | 100 pairs evaluated by Human Majority vs. GPT-4o-Audio-Preview | Fleiss' Kappa ($\kappa$) | 0.8417 | Strong Alignment |
>
> 1. AMOS Inter-Rater Reliability
>
> We measured the agreement among 30 expert annotators on 100 sampled clips using Krippendorff’s Alpha (ordinal). The Alpha score of 0.8362 indicates high inter-rater consistency, confirming that AMOS reflects stable human judgment rather than subjective variance.
>
> 2. Proxy Validity of Win-Attr / Win-Qual
>
> To validate GPT-4o-Audio-Preview as an evaluation proxy, we compared its win/loss decisions with human majority votes on 100 paired comparisons. Fleiss’ Kappa scores of 0.8135 (Win-Attr) and 0.8417 (Win-Qual) fall within “substantial agreement,” demonstrating strong alignment with human ground truth.
>
> *Conclusion*: both validation studies confirm that our subjective and proxy-based metrics are statistically reliable and faithfully capture the performance differences between models.
>
> ## **Minor issues**
>
> Thank you for noting these minor issues. All citation and spelling mistakes have been corrected.

---

> ### Author Response · Authors · 2025-11-21
> **Replies to Reviewer EnEN for Questions 1&2**
>
> ##  **Q1： When Arithmetic Composition Breaks Down**
>
> We agree that perfectly disentangling coupled speech attributes remains fundamentally challenging. Our experiments reveal three major cases where arithmetic composition tends to fail:
>
> 1. Attribute Entanglement (Collateral Effects)
>
> Some attributes are intrinsically non-orthogonal (e.g., accent vs. pitch). In our subtraction experiments (Section 5.3; Appendix D), removing one attribute often unintentionally affects prosody or naturalness. This is because attributes like *Accent* inherently bundle pitch, rhythm, and articulatory cues—subtracting them removes shared information needed elsewhere.
>
> 2. Cumulative Interference (Too Many Modules)
>
> Arithmetic composition works reliably for 2–3 attributes but degrades with larger combinations. Quadruple-attribute tests (Appendix B) show increased artifacts (trembling, hoarseness). Summing multiple LoRAs pushes the model away from the natural-speech parameter manifold, producing destructive interference.
>
> 3. Semantic or Physiological Conflicts
>
> Contradictory attributes (e.g., high-pitch + low-pitch) cannot be linearly satisfied. Our conflict ablations (reply to Weakness 1) show such combinations produce unstable harmonics or inconsistent speaker identity. These attributes are not opposites on a linear axis but arise from incompatible physiological mechanisms.
>
> **Conclusion**：Arithmetic composition is effective in most settings but is fundamentally limited by attribute coupling, parameter accumulation, and intrinsic semantic conflicts. We will explicitly discuss these failure modes in the revised version.
>
> ##  **Q2: Zero-Shot Capability Through Modular LoRA Design**
>
> We clarify that nearly all multi-attribute results in our paper are zero-shot compositions by design.
>
> 1. Why our compositions are zero-shot
>
> As described in Section 3.1, each LoRA adapter is trained independently on a single atomic attribute (e.g., “Happy,” “Indian Accent,” “High Pitch”).
>
> We never train on paired or mixed-attribute data.
>
> Therefore, any combination applied during inference is unseen during training, and thus qualifies as zero-shot composition.
>
> 2. Evidence from our experiments
>
> The multi-attribute results in Table 4 (dual), Tables 13–14 (triple), and Tables 15–16 (quadruple) directly demonstrate this capability.
>
> For example, the combination *“Teens + Happy + Indian + High Pitch”* in Table 15 was never present in the dataset, yet the model successfully synthesizes it by composing four atomic LoRA modules.
>
> **Conclusion**: Zero-shot compositional control is an inherent property of TTS-Hub’s modular design. While attribute coupling can introduce challenges (as noted in Weakness 1), the system consistently generalizes to novel attribute combinations without requiring combinatorial training data.

---

> ### Comment · Reviewer_EnEN · 2025-11-23
> **Replies to Authors**
>
> I sincerely thank the authors for their detailed rebuttal. My question around interference and arithmetic composition were satisfactorily answered, although the results indicate that composition does not follow the linearity, additive inverse or related properties, even though those are fundamental to inference in the proposed method. The performance gains given the additional baselines are also not super convincing as many real world cases can do indeed fall into the "Matched Reference" category, although I do recognize the novelty and extensibility of the framework. Additionally, given the evidence, the amount of trust I am able to place in LLM derived metrics is fairly low. Thus, I would like to maintain my score.

---

> > ### Author Response · Authors · 2025-11-25
> >
> > Thank you for reading our reply and for sharing your detailed perspective. We fully appreciate the opportunity to clarify your remaining concerns. We hope these clarifications help address your concerns, and we would be very grateful for any further questions or suggestions you may have.
> >
> >
> > ### **Comparison with clone-based models**
> >
> > The fundamental limitation of reference-based models is their inability to perform **continuous, relative, and compositional control** to speech attributes.
> >
> > Let us illustrate with a straightforward example. In audiobooks and game dubbing, the same character appears in hundreds of scenes with subtly different mixtures of emotion, energy, and pitch. For instance:
> >
> > - “Slightly happier with moderately higher energy but slightly lower pitch”
> >
> > - “Calmer with reduced energy and a marginally higher pitch”
> >
> > The space of such continuous multi-attribute combinations is effectively unbounded, making it unrealistic to collect a perfectly matched reference sample for each target setting.
> >
> > In the above practical cases, the need shifts from cloning a static performance to exercising continuous and compositional control. This is the core problem our method is designed to solve. We believe this capability is fundamental for the next generation of expressive speech synthesis.
> >
> >
> >
> > ### **LLM derived metrics**
> >
> > We understand your concern regarding LLM-derived metrics. In our previous reply to *Weakness 5: Statistical reliability analysis of evaluation metrics*, we validated GPT-4o-Audio-Preview as an evaluation proxy. Building on this analysis, we further include **Gemini-2.5 Pro** as an additional evaluation proxy. Gemini-2.5 Pro has demonstrated state-of-the-art audio understanding and reasoning on the MMAU audio-reasoning benchmark [3], making it a strong candidate for proxy-based evaluation. We further evaluated its agreement with human majority votes, obtaining Fleiss’ Kappa scores of 0.8445 (Win-Attr) and 0.8509 (Win-Qual). Notably, Gemini-2.5 Pro shows higher agreement with human judgments than GPT-4o-Audio-Preview. Together, these results confirm the robustness of our metric design.
> >
> > | Metric  | Validation Setup                | Statistical Measure     | Result | Interpretation  |
> > |--------------|----------------------------------------|------------------------------|:----------:|----------------------|
> > | Win-Attr | Human Majority vs. Gemini-2.5 Pro      | Fleiss' Kappa ($\kappa$)     | 0.8445 | Strong Alignment     |
> > | Win-Qual | Human Majority vs. Gemini-2.5 Pro      | Fleiss' Kappa ($\kappa$)     | 0.8509 | Strong Alignment     |
> >
> >
> >
> >
> > Moreover, we would like to emphasize that LLM-based metrics are used as **auxiliary** diagnostic tools, rather than as the sole basis for our claims.
> >
> >
> >
> > We also add references to recent work [4,5,6] that *adopts LLM-based metrics to evaluate speech quality* to better contextualize our use of such proxies within current practice in the community.
> >
> >
> >
> > [3] Sakshi, S., et al. "Mmau: A massive multi-task audio understanding and reasoning benchmark." arXiv preprint arXiv:2410.19168 (2024).
> >
> > [4] Lou, Haowei, et al. "StyleSpeech: Parameter-efficient Fine Tuning for Pre-trained Controllable Text-to-Speech." Proceedings of the 6th ACM International Conference on Multimedia in Asia. 2024.
> >
> > [5] Zhan, Jun, et al. "Vstyle: A benchmark for voice style adaptation with spoken instructions." arXiv preprint arXiv:2509.09716 (2025).
> >
> > [6] Manku, Ruskin Raj, et al. "EmergentTTS-Eval: Evaluating TTS Models on Complex Prosodic, Expressiveness, and Linguistic Challenges Using Model-as-a-Judge." arXiv preprint arXiv:2505.23009 (2025).

---

### Author Response · Authors · 2025-11-27
**General Response to All Reviewers**

Dear Reviewers,

We sincerely thank all reviewers for their thoughtful and constructive feedback. Below, we summarize the key concerns raised during the first-round review and highlight the major revisions made in response.

**Novelty of our work**.
Our work aims to answer the following question:

>Can controllable TTS be realized by modeling attributes as **atomic LoRAs** and *freely* composing them with **arbitrary combinations** and **tunable strengths** using **simple LoRA arithmetic**?

Our contribution differs fundamentally from existing LoRA-based and reference-based TTS methods:

   -  **Fine-grained control**: Unlike prior methods that model holistic styles or identities, we factorize the speech into a set of *atomic* attributes, each parameterized by an independent LoRA module, allowing explicit and independent control over each one.
  - **Composable and continuous control**: Our arithmetic composition allows *multiple attributes to be combined with adjustable strengths*, achieving *zero-shot multi-attribute and continuous control*. This is impossible for reference-based methods, as they cannot provide infinite reference samples to cover every conceivable combination of attributes and their intensities.
  - **Backbone-agnostic design**: The modules are fully plug-and-play and do not depend on a particular model architecture, allowing controllable synthesis to seamlessly benefit from any *stronger future backbone*.


**Reliability of Human and LLM-Based Evaluations**.  We confirmed the reliability of our evaluation metrics through dedicated validation. Human AMOS exhibits strong inter-rater agreement (Krippendorff’s α = 0.8362), and both GPT-4o-Audio-Preview and Gemini-2.5 Pro show high alignment with human-majority judgments (Fleiss’ κ = 0.81–0.85). This demonstrates that both our human and LLM-based evaluations are statistically sound and dependable.

**Comparison Against Stronger Baselines**.  We compare our method with two SOTA reference-based TTS systems, CosyVoice 2 and StyleSpeech. The results show that reference-based systems perform well when a perfect attribute-matched reference is available, whereas ours achieves competitive attribute control without requiring any reference, confirming the practical advantage of our approach.

**Understanding the Limits of Arithmetic Composition**.  We conducted additional experiments to better understand the edge cases of arithmetic composition. We find that (i) conflicting attributes (e.g., high vs. low pitch) do not cancel out but may interfere, and (ii) subtracting an attribute is not equivalent to adding its opposite. These observations do not affect the overall effectiveness of our method, but they clarify the nonlinear structure of the attribute space and help define where arithmetic composition may behave less predictably.

**More Ablation Studies on Model Design and Efficiency**.  We conducted a series of ablation studies examining different aspects of TTS-Hub. (i) We introduced an advanced fusion strategy using a lightweight Mini-MoE router, showing that learned mixing weights can further improve multi-attribute synthesis. (ii) We performed layer-wise insertion ablations, revealing that different attributes rely on different parts of the backbone and validating our choice of using full-layer LoRA modules for robust generality. (iii) We analyzed the runtime and memory behavior of multi-LoRA composition and confirmed that our approach introduces negligible latency and VRAM overhead.


We thank all reviewers again for their valuable comments. We believe the revisions substantially strengthen the overall quality of the paper, and we would greatly appreciate any further comments or questions you may have. Please let us know if there are remaining concerns we can clarify. *We look forward to your interaction and feedback.*



Sincerely,

The Authors

---

### Meta-Review · Area_Chair_Wqv5 · 2025-12-25

**Summary:**

This paper introduces a modular framework for Controllable Text-to-Speech, termed TTS-Hub, which uses LoRA to model diverse speech attributes, and then use arithmetic composition for flexible and fine-grained TTS control. Reviewers' concerns can be summaried as:

- Incremental novelty: Reviewer **BCXY** points out that it is not novel to compose LoRA for solving tasks, and has been used in previous method. The proposed method mainly focus on how to decoupled fine-grained attributes from speech.
- Lacking Ablation: Reviewer **EnEN**, **S5Dp** points out that need to analyze the interference, fusion, how to select layers and so on.
- Evaluation issues: Reviewers (**EnEN**, **BCXY**, ** S5Dp**, **qk2A**) points that this paper mainly use some unverified and subjective metrics (e.g., LLM-as-judge, AMOS), may affect its trustworthiness.
- Insufficient Baselines: Reviewer **EnEN**, **BCXY**, **S5Dp**, **qk2A** points that this paper lack sufficient baselines for comparisons.
- Generalization on pre-defined attributes: Reviewer **EnEN**, **fo7r** points out the proposed method heavily relied on predefined attributes, and cannot be generalized to unknown attributes.

Therefore, this paper still remains some issues that need to address. Considering that ICLR is highly competitive, this paper is not selected for acceptance in this venue, and I suggest authors to solve the reviewer's comment and re-submit it in the next time.

**Reviewer Concerns:**

Authors have provided some results (e.g., ablation study, and additional baselines) to address some experimental issues.

**Reviewer Scores:**

Although authors have address some issues, Reviewer **EnEN** mentions the baseline used in this paper is still not convincing, and more confidenced metrics are required. Therefore, reviewers mostly will not change their scores.

---

### Decision · Program_Chairs · 2026-01-26

Reject